# Breaking Mental Set to Improve Reasoning through Diverse Multi-Agent Debate

**Yexiang Liu**[1,2]**, Jie Cao**[1]*****, Zekun Li**[3]**, Ran He**[1,2]**, Tieniu Tan**[1,2,4]
[1]MAIS & NLPR, Institute of Automation, Chinese Academy of Sciences
[2]School of Artificial Intelligence, University of Chinese Academy of Sciences
[3]University of California, Santa Barbara    [4]Nanjing University
`liuyexiang2023@ia.ac.cn, jie.cao@cripac.ia.ac.cn`

## Abstract

Large Language Models (LLMs) have seen significant progress but continue to struggle with persistent reasoning mistakes. Previous methods of *self-reflection* have been proven limited due to the models' inherent fixed thinking patterns. While Multi-Agent Debate (MAD) attempts to mitigate this by incorporating multiple agents, it often employs the same reasoning methods, even though assigning different personas to models. This leads to a "fixed mental set", where models rely on homogeneous thought processes without exploring alternative perspectives. In this paper, we introduce Diverse Multi-Agent Debate (DMAD), a method that encourages agents to think with distinct reasoning approaches. By leveraging diverse problem-solving strategies, each agent can gain insights from different perspectives, refining its responses through discussion and collectively arriving at the optimal solution. DMAD effectively breaks the limitations of fixed mental sets. We evaluate DMAD against various prompting techniques, including *self-reflection* and traditional MAD, across multiple benchmarks using both LLMs and Multimodal LLMs. Our experiments show that DMAD consistently outperforms other methods, delivering better results than MAD in fewer rounds. Code is available at `https://github.com/MraDonkey/DMAD`.

## 1 Introduction

In the past few years, Large Language Models (LLMs) have experienced unprecedented development (Brown et al., 2020; Touvron et al., 2023a;b; Chowdhery et al., 2023; Anil et al., 2023; Achiam et al., 2023; Open AI, 2024a), demonstrating significant potential in reasoning. Extensive research has explored various methods to enhance their reasoning capabilities (Wei et al., 2022; Wang et al., 2023; Yao et al., 2023; Besta et al., 2024; Gao et al., 2024a). Despite these efforts, LLMs continue to make errors. Previous work has sought to address this through *self-reflection* (Madaan et al., 2023; Kim et al., 2023; Shinn et al., 2023), where the model generates an initial response, then evaluates and refines it based on its own feedback. However, recent studies have raised concerns about of this approach (Huang et al., 2024; Stechly et al., 2023; Valmeekam et al., 2023). They suggest that without external feedback or additional information, LLMs struggle to effectively correct their mistakes, often leading to worse performance due to their inherent fixed thinking patterns and perspectives.

Another approach explored by many studies is the use of Multi-Agent Debate (MAD) to address this dilemma (Liang et al., 2024; Du et al., 2024; Chan et al., 2024), where multiple agents debate with each other to arrive at the final answer. This paradigm helps mitigate the limitations of a single inherent thinking process by introducing multiple models. However, existing methods often use the same LLM across different personas or roles as agents, or have the model answer the question multiple times for debate, while still relying on a homogeneous reasoning approach and thinking pattern. This is still constrained by the inherent thinking influenced by pre-training and sometimes constantly gets wrong answers even with multiple rounds.

The issue is analogous to the "mental set" phenomenon (Jersild, 1927), a widely studied psychological phenomenon. It refers to the cognitive tendency to approach problems in a particular way

---

*Jie Cao is the corresponding author.

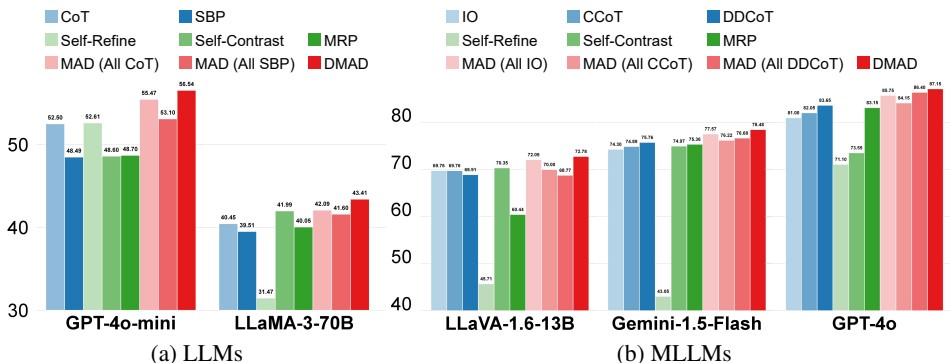

Figure 1: Overall performance of each method on all models and benchmarks. CoT: Chain-of-Thought (Wei et al., 2022). SBP: Step-Back Prompting (Zheng et al., 2024). MRP: Meta-Reasoning Prompting (Gao et al., 2024a). IO: Input / Output Prompting. CCoT: Compositional Chain-of-Thought (Mitra et al., 2024). DDCoT: Duty-Distinct Chain-of-Thought (Zheng et al., 2023).

based on past experiences, learned behaviors, or established habits. While mental sets can enable quick solutions to familiar problems, they often hinder the ability to explore diverse approaches, particularly when faced with new or more complex tasks (Öllinger et al., 2008; DeCaro, 2016).

To address this, we propose Diverse Multi-Agent Debate (DMAD), where multiple agents are guided to employ diverse reasoning methods. During the debate, each agent adopts a different reasoning approach and gathers insights from the reasoning methods of others. By comparing these approaches and extracting valuable insights, agents refine their solutions to reach the correct answer. Specifically, given a question, each agent generates an individual solution using a unique given prompting strategy. Then, each agent reviews and critiques the diverse solutions from other agents, incorporating this feedback to update its own response. This debate process is repeated over several rounds. Unlike traditional MAD, where models rely on fixed thinking, DMAD enhances interaction by promoting diverse reasoning, leading to more robust solutions. It is worth noting that, unlike other works that rely on different models or experts collaborating, which assume access to multiple models (Xiong et al., 2023; Chen et al., 2023a; Li et al., 2024a), we use the same model as different agents, aiming to improve its reasoning accuracy, as in previous MAD approaches.

We compare our DMAD approach with various MAD settings, along with different prompting and *self-reflection* methods. Experiments are conducted on Large Language Models (LLMs) using text-only benchmarks, MATH (Hendrycks et al., 2021b) and GPQA (Rein et al., 2024), as well as on Multimodal Large Language Models (MLLMs) using multimodal benchmarks, ScienceQA (Lu et al., 2022) and MM-Vet (Yu et al., 2024b). The evaluation includes two LLMs (GPT-4o-mini (Open AI, 2024a) and LLaMA-3-70B-Instruct (Dubey et al., 2024)) and three MLLMs (LLaVA-1.6-13B (Liu et al., 2023a), Gemini-1.5-Flash (Reid et al., 2024), and GPT-4o (Open AI, 2024a)). As shown in Figure 1, experiments demonstrate that DMAD outperforms other MAD settings, including those using fixed reasoning methods (Du et al., 2024) and different personas (Liang et al., 2024). With the same number of agents and rounds, DMAD improves GPT-4o and Gemini performance on MM-Vet by 7.3% and 8.2%, respectively, compared to improvements of 5.5% and 5.3% with MAD. Notably, DMAD in just two rounds achieves much higher performance than MAD in five rounds, where MAD begins to converge. Additionally, DMAD surpasses all other prompting and *self-reflection* methods. These results demonstrate the generalization ability of DMAD, which effectively enhances reasoning in both LLMs and MLLMs.

The contributions of our work can be summarized as follows:

- We introduce Diverse Multi-Agent Debate (DMAD), a novel and robust framework that guides agents to employ diverse reasoning methods, breaking mental set and improving reasoning performance.

- We compare DMAD with various methods, including basic prompting techniques, *self-reflection* methods, and different MAD approaches. Our experiments, conducted on both LLMs and MLLMs, demonstrate that DMAD consistently enhances reasoning and achieves state-of-the-art performance across all benchmarks and models.

- We conduct a systematic analysis on various MAD settings. The results show that the performance of DMAD improves as the debate round and agent number increase, and DMAD in fewer rounds even outperforms MAD in more rounds.
- Our experiments show that *self-reflection* also performs poorly on MLLMs, even worse than LLMs, which is less explored on MLLMs.

## 2 RELATED WORKS

**Prompting Reasoning Methods.** Chain-of-Thought (CoT) (Wei et al., 2022; Kojima et al., 2022) is a widely used prompting reasoning strategy to solve the problem step by step. After that, various prompting reasoning methods are proposed (Yao et al., 2023; Chen et al., 2023b; Besta et al., 2024; Zheng et al., 2024; Yasunaga et al., 2024; Yu et al., 2024a). Several works explore how to employ CoT on MLLMs (Lu et al., 2022; Zhang et al., 2024c; Wang et al., 2024a). Based on CoT, some multimodal prompting reasoning methods are proposed, which can be categorized into two types. The first type emphasizes image understanding (Mitra et al., 2024; Zhang et al., 2024a; Zhou et al., 2024; Gao et al., 2024b), while the other one focuses on text understanding (Zheng et al., 2023).

**Correcting Reasoning.** There are many different ways to correct reasoning. Some researchers train or fine-tune the model with the collected high-quality data (Huang et al., 2023; Liu et al., 2023c; An et al., 2023). Others train a corrector to help correct reasoning (Welleck et al., 2023) or collaborate with other models or tools (Zhang et al., 2023; Pan et al., 2023; Peng et al., 2023). Different from them, some works use the same LLM to self-correct completely relying on itself through the three-step pipeline of *self-reflection* (Madaan et al., 2023; Kim et al., 2023; Shinn et al., 2023; Gero et al., 2023). However, several studies indicate that LLMs struggle to self-correct reasoning by *self-reflection* (Huang et al., 2024; Stechly et al., 2023; Valmeekam et al., 2023). We extend their methods to MLLMs, only to discover that such techniques also fail to facilitate self-correction in reasoning for MLLMs, especially on small open-source MLLMs.

**Multi-Agent Debate and Diverse Solutions.** Recently, significant attention has been dedicated to the development of Multi-Agent Debate. Many works have studied MAD from different perspectives. Some assign different agents to play different roles (Liang et al., 2024; Wang et al., 2024b; Chan et al., 2024; Chen et al., 2024). Some utilize several LLM instances to debate with each other and study the effect of the debate round, order, and agent number (Du et al., 2024; Li et al., 2024b). Other researches leverage different models or expert modules to collaborate (Xiong et al., 2023; Chen et al., 2023a; Hong et al., 2024; Li et al., 2024a). There are also other methods improving MAD through embeddings (Pham et al., 2024). However, most of them debate with a single thinking which may lead to mental set. Instead, we propose DMAD to encourage different agents to think with distinct reasoning methods, which can break mental set. Similarly, Zhang et al. (2024b) contrasts different reasoning solutions and revises them until they are consistent. Gao et al. (2024a) dynamically selects the most suitable reasoning method to solve the problem.

## 3 METHOD

Our idea is inspired by the psychological theory of mental set, which may hinder diverse thinking and make it difficult to finish a task in the mental set way. However, if thinking in a different way, one may surprisingly find it is easy to address the problem. We observe that LLMs have an analogous phenomenon. MAD with a fixed prompting strategy may always get wrong answers to a problem, while changing to another one can correctly solve it. This applies to MAD with each single strategy. We elaborate on it and conduct experiments to verify this finding in Appendix E.1.

Inspired by this, we argue the importance of utilizing different reasoning methods in debate to promote diverse thinking. To fully harness the inherent capabilities of off-the-shelf models, we employ a variety of prompting reasoning techniques to represent distinct modes of thought, which do not need training or fine-tuning. We endeavor to select reasoning methods with significant divergence to avoid the issue of similar reasoning approaches potentially leading to mental set.

We select $n$ different prompting reasoning methods $\mathbb{R} = \{\mathfrak{R}_i \mid i = 1, 2, ..., n\}$ and set $n$ model instances $\{\mathcal{M}_i \mid i = 1, 2, ..., n\}$ as agents. Each agent $\mathcal{M}_i$ is required to answer the question with the corresponding reasoning method $\mathfrak{R}_i$. At the first round of debate, each agent $\mathcal{M}_i$ generates its

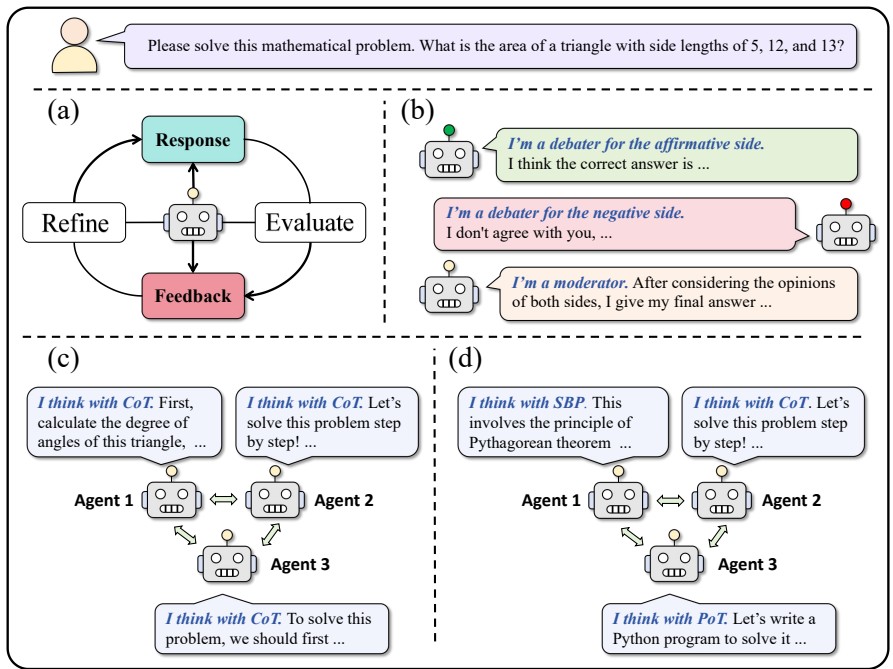

Figure 2: Pipelines of different methods. (a) *Self-reflection* uses a three-step strategy: generate response → evaluate it to get feedback → refine the response according to feedback. (b) MAD-persona sets the same model as debaters for affirmative and negative sides, and assigns a moderator to summarize their opinions. (c) MAD uses model instances to debate with each other in parallel. All agents think with the same reasoning method. (d) DMAD forces different agents to think with distinct reasoning methods. Each agent can extract useful information and gain insights from other agents to break mental set.

solving processes $\mathbf{s}_{i,1}$ (the subscript "1" represents the first round) to the text question $\mathbf{x}$ ($\mathbf{x}$ can also contain images) guided by $\mathfrak{R}_i$,

$$\mathbf{s}_{i,1} = \mathcal{M}_i\left(\mathbf{x} \mid \mathbf{h}_i; \mathfrak{R}_i\right), \quad i = 1, 2, ..., n. \tag{1}$$

where $\mathbf{h}_i$ is the history messages for agent $\mathcal{M}_i$, its initial value is an empty list $[\,]$. According to the solving process $\mathbf{s}_{i,1}$, each model $\mathcal{M}_i$ generates its final answer $\mathbf{y}_{i,1}$ corresponding to the reasoning method $\mathfrak{R}_i$.

$$\mathbf{y}_{i,1} = \mathcal{M}_i\left(\mathbf{x}, \mathbf{s}_{i,1} \mid \mathbf{h}_i; \mathfrak{R}_i\right), \quad i = 1, 2, ..., n. \tag{2}$$

Consequently, we collect the solving processes and answers to the question of all agents and sequentially extend the message from $\mathcal{M}_i$ and other agents' messages to the history $\mathbf{h}_i$ of $\mathcal{M}_i$.

$$\mathbf{A}_{i,1} = (\mathbf{x}, \mathbf{s}_{i,1}, \mathbf{y}_{i,1}), \quad \mathbb{H} = \{\mathbf{A}_{i,1} \mid i = 1, 2, ..., n\} \tag{3}$$

$$\mathbf{h}_i \leftarrow [\{\mathbf{A}_{i,1}\}, \mathbb{H} \setminus \{\mathbf{A}_{i,1}\}], \quad i = 1, 2, ..., n. \tag{4}$$

where $\leftarrow$ means `extend`, $\mathbf{A}_{i,1}$ here denotes the message from $\mathcal{M}_i$, which is a collection of its solving process and answer, along with the question. $\mathbb{H}$ is the message pool of all agents. $\setminus$ denotes set subtraction. $\mathbf{h}_i$ represents the history of messages for $\mathcal{M}_i$, extended with messages from other agents[1]. Next, another round of debate begins. Each agent receives other agents' solving processes and solutions as additional information, extracts useful messages from other diverse reasoning solutions, and updates its own answer. The subsequent debate follows the same process as described above, until reaching the maximum number $N$ of rounds.

Optionally, a judge $\phi$ can be set to get the final solution to the debate in a certain round. This can be achieved through Self-Consistency (Wang et al., 2023) to extract the most repeated answer, or forcing the model $\mathcal{M}$ to choose the best one from $n$ reasoning solutions $\{\mathbf{y}_{i,1} \mid i = 1, 2, ..., n\}$, or

---

[1]This is consistent with the original prompt of MAD Du et al. (2024) for a fair comparison, which places one agent's own messages in front while others at the back during each debate round.

other settings. Then, we obtain the debate solution $\mathbf{y}_j^*$ in the $j$th round. Algorithm 1 provides a comprehensive summary of the procedures involved in DMAD.

$$\mathbf{y}_j^* = \phi\left(\{\mathbf{y}_{i,j} \mid i = 1, 2, ..., n\}\right) \tag{5}$$

---

**Algorithm 1** DMAD algorithm

---

**Require:** input query $\mathbf{x}$, $n$ model instances $\{\mathcal{M}_i \mid i = 1, 2, ..., n\}$,
    $n$ reasoning methods $\{\mathfrak{R}_i \mid i = 1, 2, ..., n\}$, $n$ debate histories $\{\mathbf{h}_i \mid i = 1, 2, ..., n\}$
    debate rounds $N$, judge $\phi$
 1: **for** Round $j = 1, ..., N$ **do**
 2:   **for** Agent $i = 1, ..., n$ **do**
 3:     $\mathbf{s}_{i,j} = \mathcal{M}_i\left(\mathbf{x} \mid \mathbf{h}_i; \mathfrak{R}_i\right),$         ▷ Solving processes (Equation 1)
 4:     $\mathbf{y}_{i,j} = \mathcal{M}_i\left(\mathbf{x}, \mathbf{s}_{i,j} \mid \mathbf{h}_i; \mathfrak{R}_i\right)$       ▷ Candidate answers (Equation 2)
 5:   **end for**
 6:   $\mathbf{A}_{i,j} = (\mathbf{x}, \mathbf{s}_{i,j}, \mathbf{y}_{i,j}), \mathbb{H} = \{\mathbf{A}_{i,j} \mid i = 1, 2, ..., n\}$  ▷ Collecting messages (Equation 3)
 7:   **for** Agent $i = 1, ..., n$ **do**
 8:     $\mathbf{h}_i \leftarrow [\{\mathbf{A}_{i,j}\}, \mathbb{H} \setminus \{\mathbf{A}_{i,j}\}]$       ▷ Updating histories (Equation 4)
 9:   **end for**
10:   $\mathbf{y}_j^* = \phi\left(\{\mathbf{y}_{i,j} \mid i = 1, 2, ..., n\}\right)$     ▷ Obtaining debate solutions (Equation 5)
11: **end for**

---

## 4 EXPERIMENTS

We test multiple approaches in different types both on LLMs and MLLMs. To avoid potential issues of mental sets caused by similar reasoning methods, we endeavor to select methods with significant differences as much as possible. We will expound our experiments on LLMs in Section 4.1 and experiments on MLLMs in Section 4.2, respectively.

### 4.1 EXPERIMENTS ON LLMS

We test on two mainstream LLMs, a commercial closed-source model GPT-4o-mini-2024-07-18 and an open-source model LLaMA-3-70B-Instruct. We use their default settings and hyper-parameters.

#### 4.1.1 BASELINES

We select $n = 3$ distinct reasoning methods as $\mathbb{R} = \{\mathfrak{R}_1, \mathfrak{R}_2.\mathfrak{R}_3\}$. We also compare our method with these basic prompting reasoning methods and these with Self-Consistency (SC) (Wang et al., 2023). The model answers the question 3 times and extracts the most consistent answer.

- **Chain-of-Thought Prompting (CoT)** (Wei et al., 2022) solves the problem step by step.
- **Step-Back Prompting (SBP)** (Zheng et al., 2024) teaches the model to step back, derive high-level abstractions such as concepts and principles involved in solving the problem at first, and then leverage these concepts and principles to solve the problem.
- **Program of Thoughts Prompting (PoT)** (Chen et al., 2023b) writes an executable Python program to solve the problem, and stores the result as a variable named "*ans*".

We design and test different MAD settings to prove the effectiveness of our method. To make a fair comparison, we set $n = 3$ agents and $N = 2$ rounds for all MAD settings. We set $\phi$ as Self-Consistency to get a final solution in each debate round. We also compare with MAD-persona (Liang et al., 2024) with both discriminative and extractive modes.

- **MAD-persona** sets the model to play different roles: debaters in affirmative and negative sides, the moderator, and the judge. Debaters argue with the initial solution, the moderator summarizes their opinions to give a final answer, and the judge chooses the side it supports.
- **MAD (All CoT)** sets all agents to think with CoT, *i.e.*, $\mathfrak{R}_1 = \mathfrak{R}_2 = \mathfrak{R}_3 = \text{CoT}$.
- **MAD (All SBP)** sets all agents to think with SBP, *i.e.*, $\mathfrak{R}_1 = \mathfrak{R}_2 = \mathfrak{R}_3 = \text{SBP}$.
- **MAD (All PoT)** sets all agents to think with PoT, *i.e.*, $\mathfrak{R}_1 = \mathfrak{R}_2 = \mathfrak{R}_3 = \text{PoT}$.
- **DMAD** sets each agent with each distinct reasoning method, *i.e.*, $\mathbb{R} = \{\text{CoT, SBP, PoT}\}$.

Table 1: Detailed results of the accuracy on each subject on MATH. MAD-persona-D: MAD-persona in discriminative mode. MAD-persona-E: MAD-persona in extractive mode. Alg.: Algebra, C&P: Counting and Probability, Geom.: Geometry, Int. Alg.: Intermediate Algebra, Num. Th.: Number Theory, PreAlg.: PreAlgebra, PreCalc: PreCalculus, Avg.: Average.

| Models | Methods | Alg. | C&P | Geom. | Int. Alg. | Num. Th. | PreAlg. | PreCalc. | **Avg.** |
|---|---|---|---|---|---|---|---|---|---|
| | CoT | 92 | 74 | 55 | 54 | 77 | 82 | 37 | 67.29 |
| | SBP | 90 | 75 | 54 | 39 | 78 | 86 | 35 | 65.29 |
| | PoT | 62 | 68 | 37 | 32 | 72 | 71 | 30 | 53.14 |
| | CoT - SC | **94** | 76 | 53 | **58** | 77 | 84 | 38 | 68.57 |
| | SBP - SC | 89 | 78 | 54 | 43 | 82 | 84 | 35 | 66.43 |
| | PoT - SC | 64 | 68 | 40 | 34 | 79 | 74 | 34 | 56.14 |
| | Self-Refine | 93 | 76 | 51 | 51 | 77 | 82 | **44** | 67.71 |
| GPT-4o-mini | Self-Contrast | 89 | 67 | 48 | 40 | 70 | 83 | 38 | 62.14 |
| | MRP | 90 | 74 | 53 | 41 | 78 | 85 | 34 | 65.00 |
| | MAD-persona-D | 92 | 69 | 44 | 39 | 78 | 81 | 34 | 62.43 |
| | MAD-persona-E | 86 | 72 | 44 | 40 | 78 | 80 | 37 | 62.43 |
| | MAD (All CoT) | 93 | 79 | 57 | 57 | 82 | **86** | 40 | 70.57 |
| | MAD (All SBP) | 89 | 79 | 51 | 41 | 79 | 84 | 39 | 66.00 |
| | MAD (All PoT) | 90 | 74 | 52 | 48 | 81 | 85 | 40 | 67.14 |
| | **DMAD (Ours)** | 93 | **82** | **59** | 54 | **83** | 86 | 41 | **71.14** |
| | CoT | 69 | 44 | 29 | 22 | 31 | 66 | 29 | 41.43 |
| | SBP | 58 | 50 | 18 | 19 | 35 | 69 | 26 | 39.29 |
| | PoT | 43 | 42 | 26 | 13 | 44 | 50 | 21 | 34.14 |
| | CoT - SC | **75** | 42 | 30 | 24 | 36 | 70 | 30 | 43.86 |
| | SBP - SC | 65 | 50 | 20 | 21 | 36 | 69 | 27 | 41.14 |
| | PoT - SC | 45 | 40 | 27 | 13 | 48 | 51 | 22 | 35.14 |
| | Self-Refine | 64 | 35 | 20 | 16 | 36 | 59 | 20 | 35.71 |
| LLaMA-3-70B | Self-Contrast | 69 | 44 | 28 | 28 | 37 | 71 | 25 | 43.14 |
| | MRP | 59 | 52 | 19 | 19 | 35 | 62 | 27 | 39.00 |
| | MAD-persona-D | 54 | 37 | 21 | 23 | 24 | 54 | 18 | 33.00 |
| | MAD-persona-E | 67 | 44 | 25 | 22 | 31 | 65 | 22 | 39.43 |
| | MAD (All CoT) | **75** | 45 | 32 | 24 | 39 | 72 | 32 | 45.57 |
| | MAD (All SBP) | 66 | 53 | 26 | 25 | 39 | 71 | 26 | 43.71 |
| | MAD (All PoT) | 68 | **54** | 30 | 29 | **51** | 61 | **32** | **46.43** |
| | **DMAD (Ours)** | 72 | 46 | **32** | **30** | 41 | **74** | 25 | 45.71 |

What's more, we also test other self-correction methods. Self-Contrast and MRP are endowed with the same 3 reasoning methods to make a fair comparison.

- **Self-Refine** (Madaan et al., 2023; Kim et al., 2023) is a typical and widely used *self-reflection* method consisting of three steps: 1. generate an initial response, 2. find problems and produce feedback. 3. refine the initial answer according to the feedback.

- **Self-Contrast** (Zhang et al., 2024b) explores diverse reasoning solutions, contrasts their differences, and summarizes these discrepancies into a checklist. The model reflects on the checklist and revises each reasoning solution for consensus.

- **Meta-Reasoning Prompting (MRP)** (Gao et al., 2024a) guides the model to dynamically choose the most suitable prompting reasoning method to solve the problem.

### 4.1.2 BENCHMARKS

- **MATH** (Hendrycks et al., 2021b) is a hard mathematics benchmark consisting of high school math competitions involving seven subjects. We randomly select 100 test samples[2] in each subject with random seed 0.

- **GPQA** (Rein et al., 2024) is a challenging graduate-level Q&A benchmark on multiple-choice questions written by domain experts in biology, chemistry and physics. We test all methods and models on the whole dataset.

---

[2]This is to save API expenses and computation time, just like other works do (Du et al., 2024; Stechly et al., 2023; Valmeekam et al., 2023; Yang et al., 2023; Yin et al., 2024; Yu et al., 2024a)

Table 2: Detailed results of the accuracy on each subject on GPQA.

| Methods | GPT-4o-mini | | | | LLaMA-3-70B | | | |
|---|---|---|---|---|---|---|---|---|
| | Biology | Chemistry | Physics | **Average** | Biology | Chemistry | Physics | **Average** |
| CoT | 50.0 | 27.3 | 42.8 | 37.7 | 64.1 | 31.1 | 37.4 | 39.5 |
| SBP | 51.3 | 21.3 | 33.7 | 31.7 | 60.3 | 31.1 | 39.6 | 39.7 |
| CoT - SC | 55.1 | 30.6 | 42.8 | 40.0 | 65.4 | 31.7 | 36.9 | 39.7 |
| SBP - SC | 51.3 | 24.0 | 36.9 | 34.1 | 55.1 | 30.6 | 36.9 | 37.5 |
| Self-Refine | 41.0 | 29.5 | 43.9 | 37.5 | 41.0 | 18.0 | 30.5 | 27.2 |
| Self-Contrast | 53.8 | 25.7 | 36.4 | 35.1 | 64.1 | 33.3 | 38.5 | 40.8 |
| MRP | 51.3 | 21.9 | 34.8 | 32.4 | 59.0 | **35.5** | **39.0** | **41.1** |
| MAD (All CoT) | 55.1 | 32.2 | 42.2 | 40.4 | 61.5 | 31.7 | 35.8 | 38.6 |
| MAD (All SBP) | 56.4 | 27.9 | **45.5** | 40.2 | 61.5 | 31.1 | 38.5 | 39.5 |
| **DMAD (Ours)** | **59.0** | **33.3** | 43.3 | **41.9** | **66.7** | 32.3 | **39.0** | **41.1** |

### 4.1.3 RESULTS

Table 1 summarizes the experiment results on MATH. On GPT-4o-mini, DMAD achieves the best average accuracy compared to any other method. While on LLaMA-3-70B, MAD (All PoT) performs the best and DMAD is the second. This is because PoT is skilled at calculation questions. For example, in the subject of Number Theory, we can see that LLaMA-3-70B with PoT, PoT - SC, and MAD (All PoT) all far exceed other methods. Due to occasional errors in the DMAD agent with PoT and LLaMA's relatively weak capability to use diverse thinking for reasoning compared with GPT-4o-mini, DMAD does not achieve performance beyond MAD (All PoT). Nonetheless, the performance of DMAD is better than other methods apart from the PoT series. What's more, it can be observed that MAD-persona performs very poorly both in discriminative and extractive modes, even worse than CoT. This indicates that MAD-persona may not be suitable for reasoning tasks, although it has been proven effective in machine translation (Liang et al., 2024).

We deprecate PoT when testing on GPQA, as PoT is only suitable for calculation questions, while GPQA consists of biology, chemistry, and physics questions. Table 2 summarizes detailed results on GPQA. GPT-4o-mini achieves a 41.9% average accuracy with DMAD and LLaMA-3-70B achieves 41.1%. Both of them perform the best compared with other methods. In specific subjects, our method is also substantially superior to other MAD settings. These experiments demonstrate the effectiveness of our method, which can promote diversified thinking to improve reasoning.

## 4.2 EXPERIMENTS ON MLLMS

We test on three MLLMs: LLaVA-1.6-13B (Liu et al., 2023b;a), Gemini-1.5-Flash (Reid et al., 2024) and GPT-4o (`gpt-4o-2024-05-13`) (Open AI, 2024a).

### 4.2.1 BASELINES

Similar to the experiments on LLMs, we also select $n = 3$ heterogenous prompting reasoning methods focusing on different aspects.

- **Input / Output Stanard Prompting (IO)** is the standard mode of prompting. It just inputs the images and text questions and other given information to the model. The model directly outputs the solution based on the given question and available information.

- **Compositional Chain-of-Thought (CCoT)** (Mitra et al., 2024) first instructs MLLM to systematically generate a scene graph of the input image in JSON format. Then MLLM is prompted with the original task prompt, image, and the corresponding scene graph to generate an answer. CCoT concentrates on image content and enhances the model's capability for visual understanding.

- **Duty-Distinct Chain-of-Thought (DDCoT)** (Zheng et al., 2023) first prompts MLLM to deconstruct the input question into a sequence of basic sub-questions and simultaneously answer them. Then the MLLM gives a solution according to these pieces of information related to the original question. DDCoT encourages the model to focus more on the text question and improves the model's ability of text understanding.

Table 3: Results on ScienceQA and MM-Vet. More detailed results can be found in Appendix C.1.

| Methods | LLaVA-1.6-13B | | Gemini-1.5-Flash | | GPT-4o | |
|---|---|---|---|---|---|---|
| | ScienceQA | MM-Vet | ScienceQA | MM-Vet | ScienceQA | MM-Vet |
| IO - SC | 71.15 | 46.2 | 84.18 | 64.5 | 92 | 72.0 |
| CCoT - SC | 69.66 | **47.6** | 84.18 | 65.7 | 89 | 74.1 |
| DDCoT - SC | 70.50 | 42 | 86.56 | 65.5 | 94 | 73.3 |
| Self-Refine | 45.71 | 26.8 | 47.99 | 38.1 | 69 | 73.1 |
| Self-Contrast | 71.24 | 37.1 | 83.94 | 65.8 | 75 | 72.1 |
| MRP | 70.35 | 46.5 | 85.13 | 65.6 | 93 | 73.3 |
| MAD-persona-D | 60.44 | - | 68.82 | 42.7 | 85 | 71.0 |
| MAD-persona-E | 69.66 | - | 82.70 | 54.6 | 88 | 71.5 |
| MAD (All IO) | 72.09 | - | 84.84 | 70.3 | 94 | 77.5 |
| MAD (All CCoT) | 70.00 | - | 84.63 | 67.8 | 90 | 78.3 |
| MAD (All DDCoT) | 68.77 | - | **86.66** | 66.7 | **95** | 77.8 |
| **DMAD (Ours)** | **72.78** | - | 85.57 | **71.4** | **95** | **79.3** |

Correspondingly, we use the same MLLM as agents to compare DMAD with MAD in the fixed reasoning method and MAD-persona. We also test Self-Refine, Self-Contrast, and Meta-Reasoning Prompting on all benchmarks and MLLMs, where these methods are less explored in the vision-language domain. We further discuss about Self-Refine in Appendix E.4.

### 4.2.2 BENCHMARKS

- **ScienceQA** (Lu et al., 2022) is a benchmark on multimodal multiple-choice questions with diverse science topics and annotations of their answers with corresponding lectures and explanations. We use their "QCM" input format (Question, Context, Options) and test on all data containing images in the test split of ScienceQA, which comprises 2017 image-question pairs. GPT-4o is tested on 100 questions sampled using random seed 0.

- **MM-Vet** (Yu et al., 2024b) examines MLLMs on complicated multimodal reasoning tasks with open-ended Q&A. It focuses on the integration of different core vision-language capabilities, including general visual recognition, optical character recognition, knowledge, language generation, spatial awareness, and math. MM-Vet uses GPT-4 to evaluate the consistency of the MLLM responses and labeled answers, allowing MLLMs to provide open-ended responses without being constrained by specific formats. We test all MLLMs and methods on the whole dataset.

### 4.2.3 RESULTS

Since MM-Vet is an open-ended Q&A benchmark, we can not use Self-Consistency to choose the most repeated answer. So we don't test Self-Consistency and set $\phi$ as a judge to choose the best solution in each debate round on MM-Vet. During experiments, we find LLaVA usually outputs nonsense sentences with MAD, especially on questions involving Chinese information, long context and response. Figure 16 shows an example. This may be due to the relatively weak capability and short context window of LLaVA-1.6-13B. So we don't test LLaVA with MAD series on MM-Vet.

Table 3 reports the performance of each method on ScienceQA and MM-Vet. DMAD achieves the best performance on LLaVA and GPT-4o, while only not on Gemini on ScienceQA. We can observe the methods of the DDCoT series overall perform well, reflecting that Gemini may be skilled at using DDCoT to solve ScienceQA questions. However, we actually don't know how the reasoning method performs on a problem without post-test results. And thus, we cannot determine which one is the best. What's more, DDCoT does not always perform well on other models and benchmarks, *e.g.*, MM-Vet. DMAD doesn't have this issue, which can leverage diverse reasoning solutions to refine each agent's answer. This is equivalent to creating additional useful information by the model itself, which has been proven to be crucial for self-correction (Huang et al., 2024).

Figure 3 shows an illustration of DMAD on MM-Vet. We can see that in the 1st round, Agent 1 and Agent 2 give wrong answers, while Agent 3 gives the right solving process and answer. In the 2nd round, Agent 1 and Agent 2 gain insights and extract useful messages from Agent 3. Agent 1 corrects its original answer. Agent 2 refines its solving process, adds the "30% discount" to the scene graph which is ignored in the 1st round, and gets the right answer. Agent 3 retains its right solution.

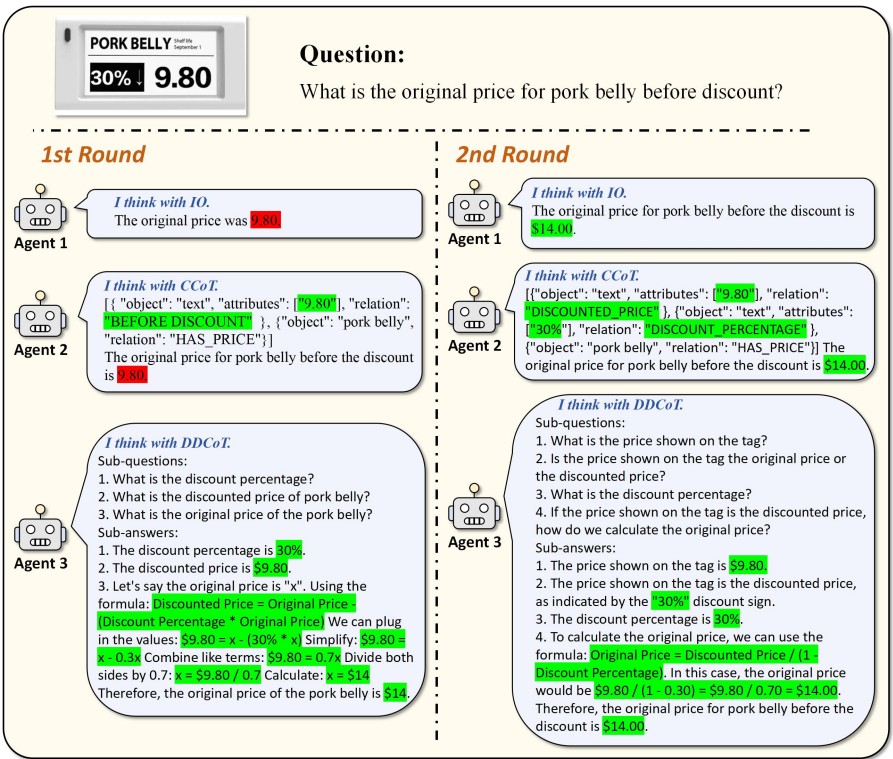

Figure 3: Illustration of DMAD. In the 1st round, Agent 1 and Agent 2 give wrong answers while Agent 3 provides the right solution. In the 2nd round, Agent 1 and Agent 2 gain insights to extract effective messages from Agent 3 and correctly update their solutions.

All agents get the right solution in 2nd Round. However, if all agents think with IO or CCoT, they cannot get the right answer even in 5 rounds, as shown in Figures 17 and 18. Similarly, there are also some questions that CCoT or IO is the most suitable method, while others cannot correctly answer. This explains the effectiveness of DMAD, whose agent can extract useful information from other agents. It uses diverse reasoning methods to teach agents to think in different ways. Without them, agents will persist in answering with a single thinking, trapped by the mental set.

## 4.3 DISCUSSIONS

### 4.3.1 NUMBER OF DEBATE ROUNDS $N$

Figure 4 shows detailed performance vs the number of debate rounds $N$ of each model on each dataset. More results are displayed in Appendix C.2. MAD is conducted with default settings as Du et al. (2024). Both MAD and DMAD are assigned 3 agents. In most cases, the overall trend of MAD and DMAD is the same, and their performance improves as the number of debate rounds increases. The overall performance of DMAD is superior to MAD. Even if DMAD performs worse than MAD in the first round, it surpasses MAD in the second round. What's more, DMAD with 2 rounds can even achieve better performance than MAD with 5 rounds, *e.g.*, Gemini on ScienceQA and MM-Vet. On MM-Vet, MAD improves slowly as the number of rounds increases, while DMAD can achieve significant improvement.

### 4.3.2 NUMBER OF REASONING METHODS $n$

Figure 5 reports the performance of DMAD with the different number of reasoning methods, *i.e.*, $n = 1, 2, 3$. As the number of reasoning methods increases, the overall performance shows an upward trend. Using as many distinct reasoning methods as possible to debate can promote more diverse thinking exchanges to achieve better overall results.

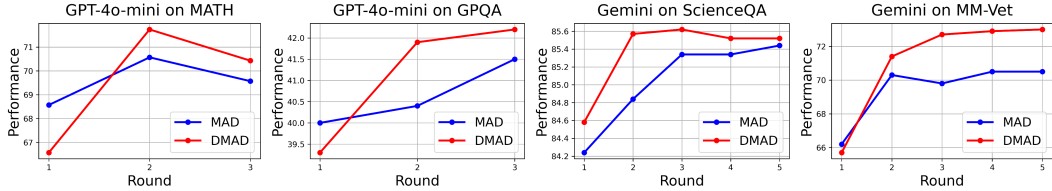

Figure 4: Performance with increased rounds. More results are shown in Figure 6.

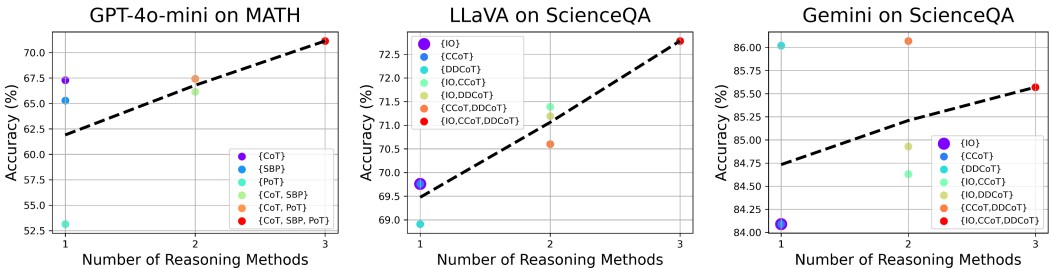

Figure 5: Performance vs the number of reasoning methods on DMAD.

Table 4: Accuracies of MAD and DAD without solving processes at the 2nd round ($\mathbf{s}_{i,2}$) on ScienceQA. NAT: Nature Science. SOC: Social Science. LAN: Language Science. AVG: Average.

| Methods | LLaVA-1.6-13B | | | | Gemini-1.5-Flash | | | | GPT-4o |
|---------|------|------|------|-------|------|------|------|-------|--------|
| | NAT | SOC | LAN | AVG | NAT | SOC | LAN | AVG | AVG |
| DMAD | **72.87** | **72.51** | **75** | **72.78** | 84.95 | **85.73** | **100** | **85.57** | **95** |
| w/o $\mathbf{s}_{i,2}$ | 70.97 | 71.07 | **75** | 71.10 | **85.28** | 84.95 | **100** | 85.47 | 94 |

### 4.3.3 WITHOUT SOLVING PROCESSES $\mathbf{s}_{i,j}$

We design experiments to study whether the solving process is necessary. We remove the solving processes $\{\mathbf{s}_{i,2}\}_{i=1}^{n}$ in the 2nd round, *i.e.*, just debating with the final answer without solving processes. Experiment results verify the necessity of solving processes, as shown in Table 4. The performance will decrease if the solving processes are removed.

### 4.3.4 BREAKING MENTAL SET

DMAD can correctly solve many problems that MAD fails due to its mental set. For example, for the same problem, all agents in DMAD reach the optimal solutions in the 2nd round (Figure 11), while all agents in MAD (All CoT) and MAD (All SBP) get the wrong answers in all rounds (Figures 12 and 13). DMAD can even reach the right solutions when all agents give wrong answers in the 1st round, as shown in Figure 22. Please refer to Appendix E and Appendix G for more analysis.

## 5 CONCLUSION

We have proposed Diverse Multi-Agent Debate (DMAD), an improved framework of Multi-Agent Debate (MAD) which leverages diverse reasoning methods to break mental set and improve reasoning. DMAD forces each agent to think with each distinct method, enabling them to gain insights to extract useful information from other agents and correct their own solutions. Comprehensive experiments on LLMs and MLLMs demonstrate the effectiveness and generalization of our method, which outperforms other methods including basic prompting techniques, Self-Refine, Self-Constrast, Meta-Reasoning Prompting, MAD-persona and MAD with each fixed reasoning method. DMAD can even reach higher performance in lower rounds than MAD. We hope that our study can bring new insights to MAD and self-correction, and promote broader research in the future.

ACKNOWLEDGEMENT

We thank Junchi Yu for his insightful discussions, and extend our appreciation to Xintong Yao and Tao Wu for their assistance in manuscript preparation. This work is supported by National Natural Science Foundation of China (Grant No. 32341009, 62425606, U21B2045, and 62206277).

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

# A    LIMITATIONS AND FUTURE WORK

The main limitation of DMAD is that the basic model needs to have a certain level of capability for reasoning and in-context learning. If the model's fundamental capability is weak, then it will be challenging to enhance its performance by DMAD and MAD. LLaVA-1.6-13B on MM-Vet is such an example, it tends to generate nonsense sentences as the context length and debate round increase.

DMAD utilizes diverse reasoning methods to facilitate the interaction of different thinking to break mental set. Each agent can extract useful information from other agents that think with different perspectives to correct its own solution. Although our method can improve the reasoning performance of both LLMs and MLLMs, it may not breakthrough the upper limit of the model's capability. If the model fails to give the right solution no matter what method is used, and every reasoning method can not provide correct information, it would be challenging for DMAD to get the correct answer. Please refer to Figure 23 for an example.

Despite these limitations, our method can demonstrate excellent performance on mainstream models. Collecting data that conforms to DMAD to use reinforcement learning to train LLMs and MLLMs like o1 (Open AI, 2024b) may lead to much stronger reasoning ability. In this work, each individual agent $\mathcal{M}_i$ thinks with the same reasoning method during the debate. Going a step further, we can dynamically allocate different methods to each agent in each round, *i.e.*, assign $\mathbb{R}_j = \{\mathfrak{R}_{i,j} \mid i = 1, 2, ..., n\}$ in the $j$th round, where $\mathfrak{R}_{i,j}$ can be different at each agent in each round. This can be achieved through hard allocation or random sampling. We can also dynamically guide agents to choose the method they deem appropriate independently. We leave this to future work and believe this may further improve the models' reasoning performance.

# B    IMPLEMENTATION DETAILS

We specifically use different prompts for each benchmark to extract the final answer in the solution for evaluation. When testing each method on the multimodal benchmark ScienceQA consisting of multiple-choice questions, we prompt all MLLMs to provide the rationale behind their choices instead of just forcing the MLLM to output the option only. Table 5 lists the prompts used on each benchmark. We initially tested GPT-4o-mini and LLaMA-3-70B with IO and CoT. We find that these two LLMs have been pretrained to solve the problem step by step. Even if we ask the LLMs to simplify their answers as much as possible, and give 1-shot prompting as *"Problem: The faces of an octahedral die are labeled with digits \$1\$ through \$8\$. What is the probability, expressed as a common fraction, of rolling a sum of \$15\$ with a pair of such octahedral dice? Answer: \$\\boxed{\\frac{1}{32}}\$"* to let LLMs directly output its final answer, it still always solves the problem step by step with a long reasoning chain. What's more, we find the performance of IO is better than CoT. For example, IO achieves 67.29% accuracy on MATH while 1-shot CoT only achieves 65.14%. So we replace IO prompting with CoT prompting. Appendix H shows detailed prompts of each method.

# C    MORE DETAILED RESULTS

## C.1    DETAILED RESULTS ON SCIENCEQA AND MM-VET

We summarize the detailed results of each subentry on ScienceQA and MM-Vet. The test split of ScienceQA consists of 1209 natural science questions, 764 social science questions, and 44 language science questions. There are respectively 7, 117, 342, 361, 336, 271, 281, and 302 questions with levels ranging from 1-8. Figures 6 and 7 summarize the detailed results of LLaVA-1.6-13B and Gemini-1.5-Flash on each subject on ScienceQA. Figures 8 and 9 summarize the detailed results on each level. Figures 10 to 12 report the detailed results of LLaVA-1.6-13B, Gemini-1.5-Flash and GPT-4o on MM-Vet, respectively.

Table 5: Prompts on each benchmark.

| Benchmark | Prompt |
|---|---|
| MATH | *Given a mathematics problem, determine the answer. ...*
*Your final answer should be in the form \\boxed{answer}, at the end of your response.* |
| GPQA | *Question:*
*<question>*

*Choices:*
*(A) <choice 1>*
*(B) <choice 2>*
*(C) <choice 3>*
*(D) <choice 4>*

*Please choose the correct choice. Your last sentence should be "The correct answer is (insert answer here, which is only the letter of the choice)".* |
| ScienceQA | *Question:*
*<question>*

*Context:*
*<hint>*

*Options:*
*(A) <option 1>*
*(B) <option 2>*
*(C) <option 3>*
*(D) <option 4>*
*...*

*Only one option is correct. Please choose the right option and explain why you choose it. You must answer in the following format. For example, if the right answer is A, you should answer:*
*The answer is A.*
*Because ...* |
| MM-Vet | *Question:*
*<question>* |

Table 6: Detailed results of LLaVA-1.6-13B with each method on each subject on ScienceQA.

| LLaVA-1.6-13B | | | | |
|---|---|---|---|---|
| Methods | Nature | Social | Language | Average |
| IO | 68.16 | 71.99 | 75.00 | 69.76 |
| CCoT | 69.56 | 69.63 | 77.27 | 69.76 |
| DDCoT | 68.91 | 66.75 | 81.82 | 68.91 |
| IO - SC | 69.89 | 72.77 | 77.27 | 71.15 |
| CCoT - SC | 69.48 | 69.76 | 72.73 | 69.66 |
| DDCoT - SC | 71.30 | 68.46 | 84.09 | 70.50 |
| Self-Refine (1 Round) | 46.73 | 43.59 | 54.55 | 45.71 |
| Self-Refine (2 Rounds) | 60.05 | 58.77 | 65.91 | 59.69 |
| Self-Contrast | 70.64 | 71.73 | 79.55 | 71.24 |
| Meta Reasoning Prompting | 69.23 | 72.38 | 65.91 | 70.35 |
| MAD-persona-D | 58.56 | 62.96 | 68.18 | 60.44 |
| MAD-persona-E | 68.16 | 71.20 | 84.09 | 69.66 |
| MAD (All IO) (1 Round) | 69.89 | 72.77 | 77.27 | 71.15 |
| MAD (All IO) (2 Rounds) | 71.13 | 72.77 | 86.36 | 72.09 |
| MAD (All IO) (3 Rounds) | 71.22 | 72.77 | 84.09 | 72.09 |
| MAD (All IO) (4 Rounds) | 69.64 | 72.12 | 84.09 | 70.90 |
| MAD (All IO) (5 Rounds) | 65.76 | 68.72 | 77.27 | 67.13 |
| MAD (All CCoT) (1 Round) | 69.15 | 69.90 | 72.73 | 69.51 |
| MAD (All CCoT) (2 Rounds) | 69.81 | 70.03 | 75.00 | 70.00 |
| MAD (All DDCoT) (1 Round) | 71.30 | 68.46 | 84.09 | 70.50 |
| MAD (All DDCoT) (2 Rounds) | 69.56 | 66.75 | 81.82 | 68.77 |
| DMAD (1 Round) | 69.48 | 71.07 | 77.27 | 70.25 |
| DMAD (2 Rounds) | 72.87 | 72.51 | 75.00 | 72.78 |
| DMAD (3 Rounds) | 68.49 | 64.53 | 70.45 | 67.03 |
| DMAD (4 Rounds) | 65.92 | 59.95 | 70.45 | 63.76 |
| DMAD (5 Rounds) | 65.76 | 60.21 | 70.45 | 63.76 |
| DMAD (IO, CCoT) (1 Round) | 69.64 | 70.42 | 77.27 | 70.10 |
| DMAD (IO, CCoT) (2 Rounds) | 70.72 | 71.99 | 79.55 | 71.39 |
| DMAD (IO, DDCoT) (1 Round) | 70.31 | 69.11 | 77.27 | 70.00 |
| DMAD (IO, DDCoT) (2 Rounds) | 70.72 | 71.47 | 79.55 | 71.19 |
| DMAD (CCoT, DDCoT) (1 Round) | 69.98 | 67.67 | 81.82 | 69.36 |
| DMAD (CCoT, DDCoT) (2 Rounds) | 70.72 | 70.29 | 72.73 | 70.60 |
| DMAD w/o $\mathbf{s}_{i,2}$ (1 Round) | 69.23 | 71.20 | 77.27 | 70.15 |
| DMAD w/o $\mathbf{s}_{i,2}$ (2 Rounds) | 70.97 | 71.07 | 75.00 | 71.10 |

Table 7: Detailed results of Gemini-1.5-Flash with each method on each subject on ScienceQA.

| Gemini-1.5-Flash | | | | |
|---|---|---|---|---|
| Methods | Nature | Social | Language | Average |
| IO | 82.88 | 85.08 | 100 | 84.09 |
| CCoT | 82.55 | 85.86 | 100 | 84.09 |
| DDCoT | 85.61 | 85.86 | 100 | 86.02 |
| IO - SC | 82.55 | 85.86 | 100 | 84.18 |
| CCoT - SC | 83.04 | 85.08 | 100 | 84.18 |
| DDCoT - SC | 86.10 | 86.52 | 100 | 86.56 |
| Self-Refine (1 Round) | 43.67 | 55.76 | 31.82 | 47.99 |
| Self-Refine (2 Rounds) | 61.29 | 68.19 | 68.18 | 64.06 |
| Self-Contrast | 82.30 | 85.86 | 95.45 | 83.94 |
| Meta Reasoning Prompting | 85.44 | 84.82 | 81.82 | 85.13 |
| MAD-persona-D | 67.33 | 70.03 | 88.64 | 68.82 |
| MAD-persona-E | 82.55 | 82.33 | 93.18 | 82.70 |
| MAD (All IO) (1 Round) | 83.21 | 85.08 | 88.64 | 84.24 |
| MAD (All IO) (2 Rounds) | 83.71 | 85.86 | 88.64 | 84.84 |
| MAD (All IO) (3 Rounds) | 84.28 | 86.26 | 88.64 | 85.34 |
| MAD (All IO) (4 Rounds) | 84.53 | 85.99 | 86.36 | 85.34 |
| MAD (All IO) (5 Rounds) | 84.62 | 85.99 | 88.64 | 85.44 |
| MAD (All CCoT) (1 Round) | 83.04 | 85.08 | 100 | 84.18 |
| MAD (All CCoT) (2 Rounds) | 83.62 | 85.34 | 100 | 84.63 |
| MAD (All DDCoT) (1 Round) | 86.10 | 86.52 | 100 | 86.56 |
| MAD (All DDCoT) (2 Rounds) | 86.52 | 86.13 | 100 | 86.66 |
| DMAD (1 Round) | 83.46 | 85.47 | 100 | 84.58 |
| DMAD (2 Rounds) | 84.95 | 85.73 | 100 | 85.57 |
| DMAD (3 Rounds) | 85.03 | 85.73 | 100 | 85.62 |
| DMAD (4 Rounds) | 85.03 | 85.47 | 100 | 85.52 |
| DMAD (5 Rounds) | 85.11 | 85.34 | 100 | 85.52 |
| DMAD (IO, CCoT) (1 Round) | 83.21 | 84.82 | 100 | 84.18 |
| DMAD (IO, CCoT) (2 Rounds) | 83.79 | 85.08 | 100 | 84.63 |
| DMAD (IO, DDCoT) (1 Round) | 83.13 | 85.47 | 95.45 | 84.28 |
| DMAD (IO, DDCoT) (2 Rounds) | 84.45 | 84.82 | 100 | 84.93 |
| DMAD (CCoT, DDCoT) (1 Round) | 84.04 | 84.95 | 100 | 84.73 |
| DMAD (CCoT, DDCoT)(2 Rounds) | 86.19 | 85.08 | 100 | 86.07 |
| DMAD w/o $\mathbf{s}_{i,2}$ (1 Round) | 83.46 | 85.47 | 100 | 84.58 |
| DMAD w/o $\mathbf{s}_{i,2}$ (2 Rounds) | 85.28 | 84.95 | 100 | 85.47 |

Table 8: Detailed results of LLaVA-1.6-13B with each method on each grade level on ScienceQA.

| Method | Grade1 | Grade2 | Grade3 | Grade4 | Grade5 | Grade6 | Grade7 | Grade8 | Average |
|---|---|---|---|---|---|---|---|---|---|
| | | | | | LLAVA-1.6-13B | | | | |
| IO | 85.71 | 82.91 | 83.92 | 76.73 | 75.60 | 58.30 | 57.30 | 55.30 | 69.76 |
| CCoT | 85.71 | 87.18 | 84.50 | 77.01 | 75.89 | 59.04 | 56.23 | 52.65 | 69.76 |
| DDCoT | 85.71 | 80.34 | 82.16 | 76.45 | 75.30 | 60.89 | 57.65 | 50.66 | 68.91 |
| IO - SC | 85.71 | 88.03 | 86.84 | 78.95 | 75.89 | 58.67 | 58.01 | 55.30 | 71.15 |
| CCoT - SC | 71.43 | 86.32 | 84.50 | 75.62 | 75.30 | 60.89 | 55.87 | 53.64 | 69.66 |
| DDCoT - SC | 85.71 | 83.76 | 84.50 | 76.45 | 77.98 | 62.36 | 58.36 | 52.32 | 70.50 |
| Self-Refine (1 Round) | 57.14 | 64.96 | 58.19 | 52.08 | 51.79 | 35.79 | 33.10 | 30.13 | 45.71 |
| Self-Refine (2 Rounds) | 57.14 | 74.36 | 70.46 | 62.33 | 66.37 | 51.66 | 52.31 | 45.36 | 59.69 |
| Self-Contrast | 100 | 84.62 | 78.95 | 77.01 | 77.68 | 66.79 | 61.75 | 55.63 | 71.24 |
| Meta Reasoning Prompting | 71.43 | 68.38 | 73.39 | 70.08 | 69.35 | 75.28 | 66.19 | 68.54 | 70.35 |
| MAD-persona-D | 71.43 | 64.10 | 64.91 | 62.33 | 65.77 | 54.98 | 53.74 | 56.62 | 60.44 |
| MAD-persona-E | 85.71 | 78.63 | 83.63 | 74.52 | 76.79 | 61.25 | 58.36 | 54.30 | 69.66 |
| MAD (All IO) (1 Round) | 85.71 | 88.03 | 86.84 | 78.95 | 75.89 | 58.67 | 58.01 | 55.30 | 71.15 |
| MAD (All IO) (2 Rounds) | 85.71 | 89.74 | 87.72 | 79.22 | 77.68 | 62.73 | 55.32 | 56.29 | 72.09 |
| MAD (All IO) (3 Rounds) | 85.71 | 88.03 | 88.60 | 77.29 | 79.46 | 61.62 | 56.94 | 55.96 | 72.09 |
| MAD (All IO) (4 Rounds) | 85.71 | 88.89 | 87.43 | 77.01 | 79.17 | 57.93 | 55.87 | 53.97 | 70.90 |
| MAD (All IO) (5 Rounds) | 85.71 | 88.89 | 85.38 | 74.79 | 77.68 | 53.14 | 50.18 | 45.03 | 67.13 |
| MAD (All CCoT) (1 Round) | 71.43 | 86.32 | 85.67 | 75.35 | 75.30 | 59.41 | 54.80 | 53.97 | 69.51 |
| MAD (All CCoT) (2 Rounds) | 85.71 | 86.32 | 85.38 | 75.07 | 76.19 | 61.62 | 57.65 | 51.99 | 70.00 |
| MAD (All DDCoT) (1 Round) | 85.71 | 83.76 | 84.50 | 76.45 | 77.98 | 62.36 | 58.36 | 52.32 | 70.50 |
| MAD (All DDCoT) (2 Rounds) | 85.71 | 78.63 | 81.87 | 76.45 | 75.3 | 60.89 | 58.01 | 50.33 | 68.77 |
| DMAD (1 Round) | 85.71 | 85.47 | 85.38 | 78.39 | 75.30 | 59.41 | 57.30 | 53.31 | 70.25 |
| DMAD (2 Rounds) | 85.71 | 88.03 | 87.43 | 81.72 | 77.98 | 61.25 | 59.43 | 56.29 | 72.78 |
| DMAD (3 Rounds) | 71.43 | 88.89 | 79.82 | 72.58 | 71.13 | 56.83 | 54.45 | 53.64 | 67.03 |
| DMAD (4 Rounds) | 71.43 | 76.92 | 74.56 | 69.81 | 66.67 | 57.20 | 54.09 | 50.66 | 63.76 |
| DMAD (5 Rounds) | 85.71 | 82.05 | 73.68 | 68.42 | 67.86 | 57.93 | 51.96 | 50.99 | 63.76 |
| DMAD (IO, CCoT) (1 Round) | 85.71 | 85.47 | 85.09 | 76.45 | 76.49 | 59.04 | 56.94 | 54.30 | 70.10 |
| DMAD (IO, CCoT) (2 Rounds) | 85.71 | 84.62 | 85.38 | 78.12 | 77.98 | 60.15 | 58.72 | 56.62 | 71.39 |
| DMAD (IO, DDCoT) (1 Round) | 85.71 | 82.05 | 82.46 | 77.56 | 75.30 | 59.41 | 59.07 | 56.63 | 70.00 |
| DMAD (IO, DDCoT) (2 Rounds) | 85.71 | 81.20 | 86.55 | 78.39 | 76.19 | 61.25 | 58.72 | 55.96 | 71.19 |
| DMAD (CCoT, DDCoT) (1 Round) | 85.71 | 85.47 | 81.58 | 78.39 | 75.89 | 59.04 | 57.30 | 51.32 | 69.36 |
| DMAD (CCoT, DDCoT) (2 Rounds) | 71.43 | 86.32 | 84.21 | 79.22 | 77.08 | 59.04 | 58.01 | 53.64 | 70.60 |
| DMAD w/o $\mathbf{s}_{i,2}$ (1 Round) | 85.71 | 84.62 | 84.80 | 78.12 | 75.60 | 59.04 | 56.94 | 54.30 | 70.15 |
| DMAD w/o $\mathbf{s}_{i,2}$ (2 Rounds) | 85.71 | 83.76 | 86.26 | 77.56 | 78.57 | 61.25 | 55.87 | 55.30 | 71.10 |

Table 9: Detailed results of Gemini-1.5-Flash with each method on each grade level on ScienceQA.

| Gemini-1.5-Falsh | | | | | | | | | |
|---|---|---|---|---|---|---|---|---|---|
| Method | Grade1 | Grade2 | Grade3 | Grade4 | Grade5 | Grade6 | Grade7 | Grade8 | Average |
| IO | 100 | 89.74 | 91.81 | 91.14 | 91.37 | 76.38 | 72.24 | 74.17 | 84.09 |
| CCoT | 100 | 90.6 | 92.69 | 90.3 | 91.96 | 77.49 | 70.82 | 73.51 | 84.09 |
| DDCoT | 100 | 85.47 | 93.86 | 87.53 | 93.15 | 82.29 | 79.36 | 76.82 | 86.02 |
| IO - SC | 100 | 89.74 | 92.11 | 91.14 | 91.67 | 76.01 | 71.89 | 74.83 | 84.18 |
| CCoT - SC | 100 | 91.45 | 92.11 | 90.86 | 91.67 | 76.75 | 71.17 | 74.5 | 84.18 |
| DDCoT - SC | 100 | 87.18 | 92.69 | 88.37 | 93.75 | 81.55 | 79.36 | 80.13 | 86.56 |
| Self-Refine (1 Round) | 14.29 | 35.9 | 55.85 | 57.89 | 58.63 | 41.33 | 40.93 | 33.44 | 47.99 |
| Self-Refine (2 Rounds) | 42.86 | 65.81 | 72.22 | 71.75 | 73.21 | 56.83 | 52.31 | 52.65 | 64.06 |
| Self-Contrast | 85.71 | 92.31 | 93.27 | 90.3 | 90.18 | 75.28 | 74.73 | 71.85 | 83.94 |
| Meta Reasoning Prompting | 85.71 | 88.03 | 82.75 | 85.87 | 86.9 | 81.92 | 86.83 | 85.1 | 85.13 |
| MAD-persona-D | 57.14 | 68.38 | 67.25 | 73.13 | 73.51 | 64.58 | 66.55 | 66.56 | 68.82 |
| MAD-persona-E | 71.43 | 86.32 | 88.3 | 89.75 | 88.1 | 77.86 | 71.89 | 75.17 | 82.7 |
| MAD (All IO) (1 Round) | 85.71 | 87.18 | 86.26 | 85.87 | 86.61 | 82.66 | 78.65 | 81.46 | 84.24 |
| MAD (All IO) (2 Rounds) | 85.71 | 87.18 | 87.13 | 86.7 | 87.2 | 83.39 | 79 | 81.79 | 84.84 |
| MAD (All IO) (3 Rounds) | 85.71 | 87.18 | 87.72 | 86.98 | 88.39 | 83.39 | 78.65 | 83.11 | 85.34 |
| MAD (All IO) (4 Rounds) | 71.43 | 88.03 | 87.43 | 86.98 | 88.69 | 83.03 | 79 | 83.11 | 85.34 |
| MAD (All IO) (5 Rounds) | 85.71 | 88.03 | 87.43 | 86.98 | 88.69 | 83.03 | 79.36 | 83.11 | 85.44 |
| MAD (All CCoT) (1 Round) | 100 | 91.45 | 92.11 | 90.86 | 91.67 | 76.75 | 71.17 | 74.5 | 84.18 |
| MAD (All CCoT) (2 Rounds) | 100 | 89.74 | 92.69 | 91.14 | 92.56 | 76.75 | 71.89 | 75.5 | 84.63 |
| MAD (All DDCoT) (1 Round) | 100 | 87.18 | 92.69 | 88.37 | 93.75 | 81.55 | 79.36 | 80.13 | 86.56 |
| MAD (All DDCoT) (2 Rounds) | 100 | 84.62 | 92.98 | 88.64 | 94.05 | 82.29 | 78.29 | 81.13 | 86.66 |
| DMAD (1 Round) | 100 | 90.6 | 93.27 | 90.86 | 91.67 | 76.75 | 72.24 | 75.17 | 84.58 |
| DMAD (2 Rounds) | 100 | 90.6 | 94.15 | 91.69 | 92.56 | 78.6 | 73.67 | 75.83 | 85.57 |
| DMAD (3 Rounds) | 100 | 88.89 | 94.15 | 91.14 | 93.15 | 77.86 | 74.73 | 76.49 | 85.62 |
| DMAD (4 Rounds) | 100 | 90.6 | 94.15 | 91.14 | 93.15 | 77.49 | 74.38 | 75.83 | 85.52 |
| DMAD (5 Rounds) | 100 | 89.74 | 94.44 | 91.14 | 93.15 | 77.49 | 74.38 | 75.83 | 85.52 |
| DMAD (IO, CCoT) (1 Round) | 100 | 90.6 | 92.4 | 91.14 | 91.37 | 76.75 | 71.89 | 73.84 | 84.18 |
| DMAD (IO, CCoT) (2 Rounds) | 100 | 89.74 | 92.69 | 91.41 | 92.56 | 76.38 | 72.95 | 74.5 | 84.63 |
| DMAD (IO, DDCoT) (1 Round) | 85.71 | 88.03 | 92.4 | 87.81 | 91.46 | 78.23 | 74.02 | 75.83 | 84.28 |
| DMAD (IO, DDCoT) (2 Rounds) | 100 | 87.18 | 92.11 | 90.86 | 92.86 | 77.86 | 73.67 | 76.49 | 84.93 |
| DMAD (CCoT, DDCoT) (1 Round) | 100 | 90.6 | 92.69 | 88.09 | 92.86 | 78.60 | 74.02 | 75.5 | 84.73 |
| DMAD (CCoT, DDCoT) (2 Rounds) | 100 | 88.89 | 92.69 | 89.75 | 94.05 | 79.7 | 76.87 | 78.15 | 86.07 |
| DMAD w/o $\mathbf{s}_{i,2}$ (1 Round) | 100 | 90.6 | 93.27 | 90.86 | 91.67 | 76.75 | 72.24 | 75.17 | 84.58 |
| DMAD w/o $\mathbf{s}_{i,2}$ (2 Rounds) | 100 | 88.89 | 93.57 | 91.14 | 92.56 | 78.23 | 74.73 | 76.49 | 85.47 |

Table 10: Detailed results of LLaVA-1.6-13B with each method on MM-Vet. Rec: Recognition. OCR: Optical character recognition. Know: Knowledge. Gen: Language generation. Spat: Spatial awareness.

| LLaVA-1.6-13B | | | | | | | |
|---|---|---|---|---|---|---|---|
| Method | Rec | OCR | Know | Gen | Spat | Math | Total |
| IO | 48.8 | 42.8 | 37.7 | 41.2 | 43.6 | 26.5 | 46.2 |
| CCoT | 49.6 | 44.0 | 39.5 | 42.1 | 45.2 | 30.4 | 47.6 |
| DDCoT | 43.7 | 38.5 | 32.0 | 34.6 | 41.6 | 30.0 | 42.0 |
| Self-Refine (1 Round) | 27.9 | 27.8 | 23.6 | 27.9 | 20.7 | 19.2 | 26.8 |
| Self-Refine (2 Round) | 30.1 | 27.8 | 23.6 | 27.3 | 23.5 | 19.2 | 28.7 |
| Self-Contrast | 41.5 | 30.9 | 28.8 | 30.9 | 36.8 | 18.8 | 37.2 |
| Meta Reasoning Prompting | 49.1 | 42.5 | 37.4 | 41.1 | 44.7 | 26.5 | 46.5 |

Table 11: Detailed results of Gemini-1.5-Flash with each method on MM-Vet. Rec: Recognition. OCR: Optical character recognition. Know: Knowledge. Gen: Language generation. Spat: Spatial awareness.

| Gemini-1.5-Flash | | | | | | | |
|---|---|---|---|---|---|---|---|
| Method | Rec | OCR | Know | Gen | Spat | Math | Total |
| IO | 60.1 | 72.5 | 51.2 | 49.8 | 67.1 | 65.4 | 64.5 |
| CCoT | 61.1 | 74.4 | 49.6 | 48.0 | 72.5 | 63.1 | 65.7 |
| DDCoT | 59.5 | 76.5 | 45.8 | 45.1 | 72.8 | 88.5 | 65.5 |
| Self-Refine (1 Round) | 34.9 | 49.5 | 32.7 | 37.9 | 41.3 | 60.8 | 38.1 |
| Self-Refine (2 Rounds) | 31.9 | 45.6 | 22.6 | 30.1 | 36.5 | 53.8 | 34.4 |
| Self-Contrast | 60.1 | 74.8 | 47.7 | 47.1 | 71.2 | 80.4 | 66.0 |
| Meta-Reasoning Prompting | 60.9 | 73.6 | 50.4 | 48.1 | 71.2 | 66.5 | 65.6 |
| MAD-persona-D | 37.6 | 52.1 | 24.2 | 23.5 | 45.3 | 57.7 | 42.7 |
| MAD-persona-E | 48.1 | 67.0 | 33.2 | 35.6 | 60.4 | 69.2 | 54.6 |
| MAD (All IO) (1 Round) | 62.1 | 73.8 | 53.5 | 52.9 | 70.5 | 65.4 | 66.2 |
| MAD (All IO) (2 Rounds) | 66.5 | 79.3 | 59.4 | 60.9 | 72.8 | 65.8 | 70.3 |
| MAD (All IO) (3 Rounds) | 65.9 | 79.0 | 56.2 | 57.7 | 73.5 | 68.8 | 69.8 |
| MAD (All IO) (4 Rounds) | 67.5 | 78.1 | 58.9 | 60.0 | 75.9 | 65.4 | 70.5 |
| MAD (All IO) (5 Rounds) | 67.8 | 77.3 | 59.9 | 61.2 | 75.9 | 65.4 | 70.5 |
| MAD (All CCoT) (1 Round) | 62.6 | 74.8 | 51.2 | 49.5 | 71.3 | 69.2 | 66.7 |
| MAD (All CCoT) (2 Rounds) | 63.7 | 77.2 | 54.5 | 55.7 | 71.5 | 73.1 | 67.8 |
| MAD (All DDCoT) (1 Round) | 62.5 | 75.2 | 52.5 | 50.3 | 71.5 | 84.6 | 67.1 |
| MAD (All DDCoT) (2 Rounds) | 62.3 | 75.7 | 50.8 | 49.4 | 73.3 | 88.5 | 66.7 |
| DMAD (1 Round) | 61.7 | 72.6 | 51.7 | 50.2 | 69.5 | 66.9 | 65.7 |
| DMAD (2 Rounds) | 65.5 | 81.0 | 55.6 | 55.7 | 77.9 | 88.5 | 71.4 |
| DMAD (3 Rounds) | 66.5 | 84.1 | 56.2 | 57.6 | 80.7 | 88.5 | 72.7 |
| DMAD (4 Rounds) | 66.7 | 85.1 | 55.2 | 56.6 | 82.0 | 84.6 | 72.9 |
| DMAD (5 Rounds) | 66.9 | 85.1 | 55.5 | 56.9 | 82.0 | 88.5 | 73.0 |

Table 12: Detailed results of GPT-4o with each method on MM-Vet. Rec: Recognition. OCR: Optical character recognition. Know: Knowledge. Gen: Language generation. Spat: Spatial awareness.

| GPT-4o | | | | | | | |
|---|---|---|---|---|---|---|---|
| Method | Rec | OCR | Know | Gen | Spat | Math | Total |
| IO | 65.8 | 81.8 | 61.0 | 65.9 | 71.9 | 94.2 | 72.0 |
| CCoT | 67.8 | 83.1 | 64.5 | 65.7 | 73.7 | 95.8 | 74.1 |
| DDCoT | 66.9 | 85.2 | 59.0 | 59.4 | 78.1 | 91.9 | 73.3 |
| Self-Refine (1 Round) | 67.5 | 86.3 | 66.0 | 68.6 | 80.0 | 91.9 | 73.2 |
| Self-Contrast | 65.3 | 83.5 | 59.2 | 60.6 | 74.4 | 91.9 | 72.1 |
| Meta-Reasoning Prompting | 67.1 | 82.7 | 64.3 | 63.5 | 73.9 | 90.4 | 73.3 |
| MAD-persona-D | 63.1 | 85.1 | 58.2 | 59.2 | 77.9 | 95.8 | 71.0 |
| MAD-persona-E | 65.8 | 80.1 | 60.5 | 60.7 | 76.9 | 88.1 | 71.5 |
| MAD (All IO) (1 Round) | 69.5 | 82.9 | 65.6 | 68.2 | 74.1 | 96.2 | 74.8 |
| MAD (All IO) (2 Rounds) | 73.2 | 83.9 | 70.5 | 71.4 | 76.5 | 96.2 | 77.5 |
| MAD (All IO) (3 Rounds) | 73.4 | 84.5 | 71.0 | 72.1 | 76.9 | 96.2 | 77.8 |
| MAD (All CCoT) (1 Round) | 68.1 | 84.1 | 65.0 | 66.1 | 74.9 | 95.8 | 74.7 |
| MAD (All CCoT) (2 Rounds) | 74.9 | 84.3 | 72.3 | 73.4 | 78.1 | 92.3 | 78.3 |
| MAD (All DDCoT) (1 Round) | 70.1 | 86.3 | 63.1 | 64.0 | 81.3 | 99.6 | 76.0 |
| MAD (All DDCoT) (2 Rounds) | 72.3 | 86.8 | 67.9 | 68.4 | 80.8 | 95.8 | 77.8 |
| DMAD (1 Round) | 68.9 | 83.0 | 63.1 | 65.0 | 73.5 | 96.2 | 74.0 |
| DMAD (2 Rounds) | 75.2 | 85.3 | 74.8 | 76.7 | 77.2 | 96.2 | 79.3 |

C.2 DETAILED RESULTS ABOUT THE PERFORMANCE WITH INCREASED ROUNDS

Figure 6 shows more results about the performance of MAD and DMAD with increased rounds. DMAD and MAD show the same trend overall. DMAD can achieve better performance in lower rounds than MAD, *e.g.*, Gemini and GPT-4o on ScienceQA and MM-Vet. As the debate round increases, the performance of LLaVA on ScienceQA will first increase and then continuously decrease. This is because the increasingly long context will cause LLaVA to output more meaningless sentences. DMAD degenerates after 3 rounds faster than MAD as its context is longer. Nevertheless, our method has a higher upper limit and can achieve higher performance than MAD. It can demonstrate significant performance on powerful models like Gemini and GPT-4o. We also visualize the average performance of MAD, DMAD, and DMAD w/o $s_{i,2}$ on each LLM and MLLM in different rounds in Figure 7 and Figure 8, respectively.

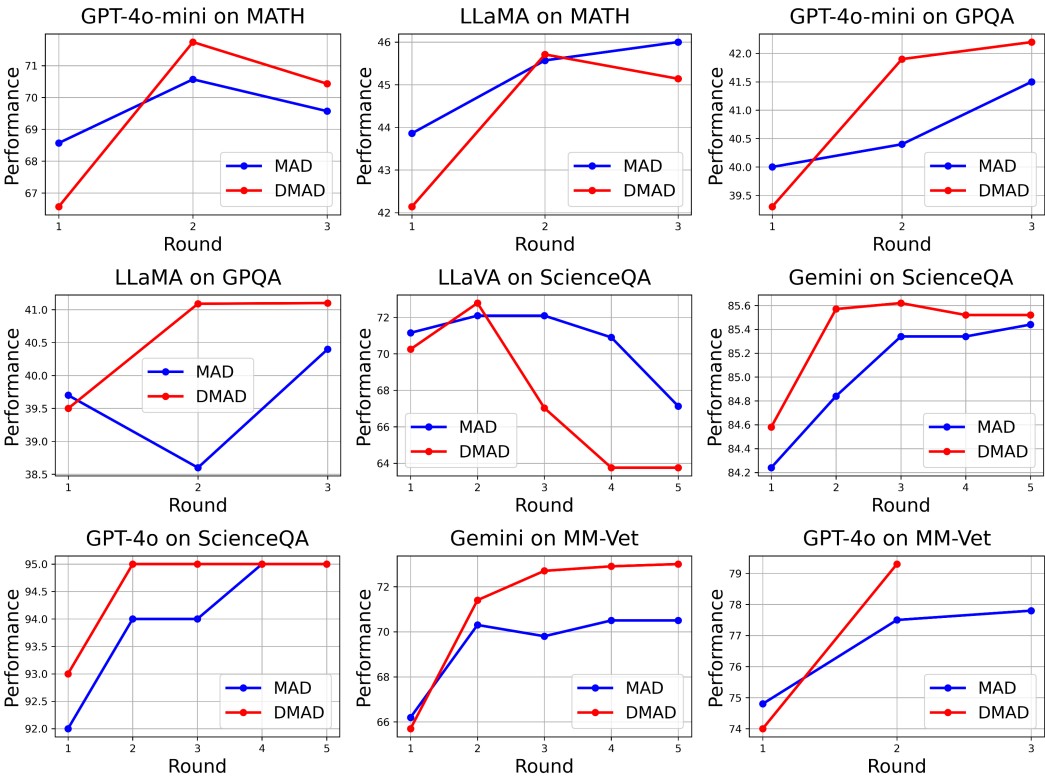

Figure 6: Performance with increased rounds. Note: MAD achieves the same accuracy as DMAD when $N \geq 4$ on GPT-4o on ScienceQA.

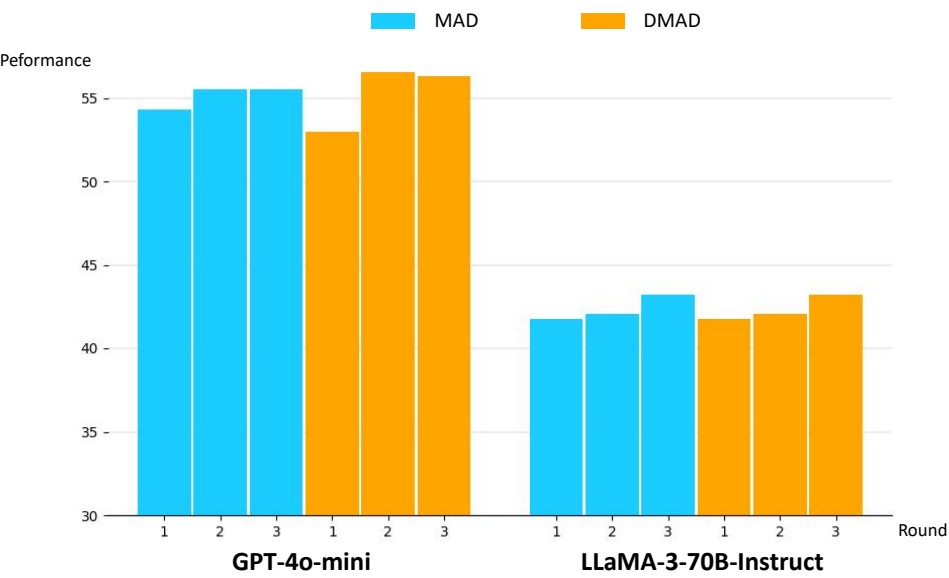

Figure 7: Average performance on MATH and GPQA of MAD and DMAD in different rounds.

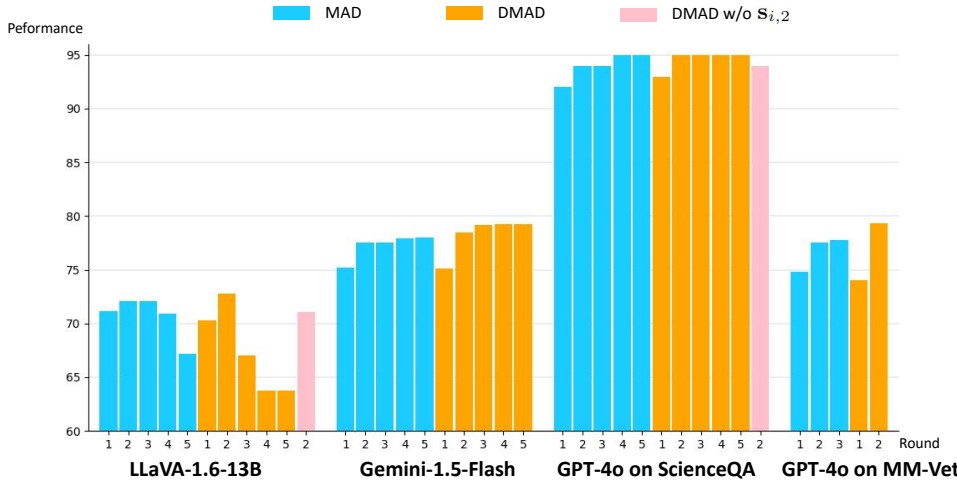

Figure 8: Average performance on ScienceQA and MM-Vet of MAD, DMAD, and DMAD w/o $s_{i,2}$ in different rounds. LLaVA-1.6-13B only shows its performance on ScienceQA. Gemini-1.5-Flash shows its average performance on ScienceQA and MM-Vet.

# D EXPANDED EXPERIMENTS

## D.1 EXPERIMENTS ON MORE CHALLENGING DATASET

To verify the effectiveness of MAD on more challenging reasoning tasks, we use GPT-4o-mini to conduct experiments on the subset "abstract_algebra" of MMLU (Hendrycks et al., 2021a) and run 3 times to get the average accuracy and standard deviation for each method. As this dataset consists of multi-choice questions and some options are not numbers, we replace PoT with Least-to-Most (L2M) prompting (Zhou et al., 2023). Results are shown in Table 13. DMAD also outperforms other MAD settings on this challenging multi-hop reasoning task.

Table 13: Results on the subset "abstract_algebra" of MMLU.

|          | MAD (All CoT)  | MAD (All SBP)  | MAD (All L2M)  | DMAD           |
|----------|----------------|----------------|----------------|----------------|
| Accuracy | $72.3_{\pm0.82}$ | $79.0_{\pm1.41}$ | $74.3_{\pm0.82}$ | $\mathbf{79.7}_{\pm1.63}$ |

## D.2 STATISTICAL SIGNIFICANCE

We run experiments with GPT-4o-mini and LLaMA-3-70B-Instruct on MATH 3 times to calculate the average accuracy and standard deviation. Table 14 and Table 15 report the results of GPT-4o-mini and LLaMA-3-70B-Instruct, respectively. Statistical experiments demonstrate that DMAD outperforms other MAD settings. DMAD on LLaMA-3-70B-Instruct also gets better average accuracy than MAD (All PoT). This implies the gains come from method diversity rather than the debate process itself to some extent.

Table 14: Results of GPT-4o-mini on MATH.

| Methods | Alg. | C&P | Geom. | Int. Alg. | Num. Th. | PreAlg. | PreCalc. | Avg. |
|---------|------|-----|-------|-----------|----------|---------|----------|------|
| MAD (All CoT) | $91.3_{\pm2.16}$ | $78.7_{\pm0.82}$ | $55.3_{\pm2.16}$ | $\mathbf{55.0}_{\pm2.45}$ | $\mathbf{82.7}_{\pm1.63}$ | $86.3_{\pm0.82}$ | $39.7_{\pm0.82}$ | $69.9_{\pm0.93}$ |
| MAD (All SBP) | $88.3_{\pm0.82}$ | $77.7_{\pm1.63}$ | $49.3_{\pm2.16}$ | $44.0_{\pm3.74}$ | $81.3_{\pm2.94}$ | $83.7_{\pm0.82}$ | $38.7_{\pm0.82}$ | $66.1_{\pm0.35}$ |
| MAD (All PoT) | $91.3_{\pm1.63}$ | $75.7_{\pm2.16}$ | $49.0_{\pm3.74}$ | $52.7_{\pm5.72}$ | $80.7_{\pm0.82}$ | $85.3_{\pm0.82}$ | $39.7_{\pm0.82}$ | $67.8_{\pm2.24}$ |
| DMAD | $\mathbf{91.7}_{\pm1.63}$ | $\mathbf{81.0}_{\pm1.41}$ | $\mathbf{57.3}_{\pm2.16}$ | $53.7_{\pm0.82}$ | $\mathbf{82.7}_{\pm0.82}$ | $\mathbf{86.3}_{\pm0.82}$ | $\mathbf{40.0}_{\pm1.41}$ | $\mathbf{70.4}_{\pm0.95}$ |

Table 15: Results of LLaMA-3-70B-Instruct on MATH.

| Methods | Alg. | C&P | Geom. | Int. Alg. | Num. Th. | PreAlg. | PreCalc. | Avg. |
|---------|------|-----|-------|-----------|----------|---------|----------|------|
| MAD (All CoT) | $\mathbf{72.7}_{\pm2.94}$ | $48.0_{\pm3.74}$ | $31.3_{\pm0.82}$ | $24.3_{\pm0.82}$ | $40.7_{\pm2.94}$ | $69.7_{\pm2.94}$ | $31.0_{\pm1.41}$ | $45.6_{\pm0.23}$ |
| MAD (All SBP) | $69.3_{\pm4.32}$ | $51.0_{\pm2.83}$ | $29.3_{\pm4.32}$ | $25.0_{\pm0.00}$ | $42.0_{\pm3.74}$ | $70.0_{\pm1.41}$ | $27.0_{\pm1.41}$ | $44.8_{\pm1.52}$ |
| MAD (All PoT) | $66.7_{\pm5.89}$ | $\mathbf{52.0}_{\pm2.83}$ | $31.0_{\pm2.45}$ | $27.7_{\pm2.16}$ | $\mathbf{49.0}_{\pm4.90}$ | $67.0_{\pm7.35}$ | $\mathbf{32.7}_{\pm1.63}$ | $46.6_{\pm0.35}$ |
| DMAD (Ours) | $\mathbf{72.7}_{\pm1.63}$ | $49.0_{\pm4.24}$ | $\mathbf{32.7}_{\pm1.63}$ | $\mathbf{29.3}_{\pm2.94}$ | $44.3_{\pm4.08}$ | $\mathbf{72.3}_{\pm2.16}$ | $27.3_{\pm3.56}$ | $\mathbf{46.8}_{\pm1.43}$ |

## D.3 ON SMALLER MODELS

We test MAD (All CoT), MAD (All SBP), MAD (PoT), DMAD on LLaMA-3-8B-Instruct on MATH. Results demonstrate DMAD is also effective on smaller models, as shown in Table 16.

Table 16: Results of LLaMA-3-8B-Instruct on MATH.

| Methods | Alg. | C&P | Geom. | Int. Alg. | Num. Th. | PreAlg. | PreCalc. | Avg. |
|---|---|---|---|---|---|---|---|---|
| MAD (All CoT) | $49.7_{\pm0.82}$ | $19.3_{\pm1.63}$ | $18.3_{\pm2.16}$ | $15.3_{\pm0.82}$ | $18.3_{\pm0.82}$ | $46.7_{\pm1.63}$ | $15.0_{\pm1.41}$ | $26.1_{\pm0.42}$ |
| MAD (All SBP) | $46.3_{\pm2.94}$ | $20.7_{\pm4.97}$ | $17.3_{\pm1.63}$ | $15.3_{\pm0.82}$ | $20.0_{\pm6.16}$ | $44.3_{\pm5.35}$ | $11.3_{\pm0.82}$ | $25.3_{\pm1.94}$ |
| MAD (All PoT) | $41.3_{\pm5.72}$ | $20.0_{\pm2.83}$ | $16.0_{\pm1.41}$ | $13.7_{\pm4.55}$ | $21.0_{\pm5.66}$ | $39.7_{\pm4.97}$ | $19.0_{\pm4.24}$ | $24.4_{\pm1.62}$ |
| DMAD | $47.7_{\pm0.82}$ | $21.3_{\pm0.82}$ | $21.7_{\pm0.82}$ | $16.0_{\pm0.00}$ | $21.3_{\pm5.72}$ | $45.0_{\pm1.41}$ | $15.0_{\pm1.41}$ | $26.9_{\pm0.40}$ |

# E  MORE ANALYSIS

## E.1  MENTAL SET OF MAD

Here we provide a specific definition of mental set. Denote MAD (All CoT), MAD (All SBP), and MAD (All PoT) as $M_1, M_2$ and $M_3$ respectively. When using a kind of MAD method $M_i$ to solve a problem, if all agents consistently get wrong answers in all debate rounds, we assume that $M_i$ is unable to correctly solve the problem. Record all such problems for $M_i$ as the set $P_i$, and get $P = P_1 \cap P_2 \cap P_3$. For a problem $p \in P_i$, if it satisfies $p \notin P$, we define that the problem $p$ causes mental set of $M_i$, and define $p$ as the mental set problem of $M_i$. It means although $M_i$ constantly gets wrong solutions, the model can correctly solve the problem by changing to another strategy.

We record the mental set problems of each method on MATH with GPT-4o-mini, and the problems among them which at least one agent of other methods correctly solves. All methods have 3 agents and debate for 2 rounds. The results prove the connection between LLM reasoning and the mental set phenomenon in psychology, and DMAD can more effectively solve other methods' mental set problems.

Table 17: Performance comparison of different methods on mental set problems.

|  | MAD (All CoT) | MAD (All SBP) | MAD (All PoT) |
|---|:---:|:---:|:---:|
| Number of mental set problems | 70 | 87 | 67 |
| Problems that MAD (All CoT) correctly solves | 0 | 45 (51.72%) | 46 (68.7%) |
| Problems that MAD (All SBP) correctly solves | 28 (40.0%) | 0 | 31 (46.3%) |
| Problems that MAD (All PoT) correctly solves | **49 (70.0%)** | 51 (58.62%) | 0 |
| Problems that DMAD correctly solves | 48 (68.6%) | **60 (69.0%)** | **49 (73.1%)** |

## E.2  STOP CRITERIA

To fairly compare with MAD, we adopt all the same settings in their paper (Du et al., 2024), which executes MAD in the fixed round. Without considering this, we can design different criteria to determine when the debate ends, such as:

1. Consistency-2: If there exist 2 agents getting the same answer, the debate ends. Otherwise, the debate continues unless reaching its maximum number of rounds.

2. Consistency-3: If all the 3 agents get the same answer, the debate ends. Otherwise, the debate continues unless reaching its maximum number of rounds.

3. Self-Determine: Set another model as a judge to determine whether the debate should be over. The judge can receive all agents' solutions in each round and save them in its history.

4. Hybrid: In each round, if all agents get the same answer, the debate ends. Otherwise, use Self-Determine to judge whether the debate should end or continue.

We set the maximum number of the debate round to 5 with Gemini on ScienceQA. With the stop criteria, MAD and DMAD can get a relatively high accuracy with a low average round. Results show that the criteria of Consistency-3 is the best. However, DMAD with the stop criteria of Consistency-3 needs more overhead with nearly 5 debate rounds.

Table 18: Results of Gemini-1.5-Flash on ScienceQA with different stop criteria.

| Methods | Stop Criteria | Average Round | Accuracy |
|---|---|---|---|
| MAD (All IO) | Consistency-2 | 1.0977 | 85.09 |
| | Consistency-3 | 1.1076 | 85.14 |
| | Self-Determine | 1.1091 | 84.89 |
| | Hybrid | 1.0605 | 84.84 |
| | Fixed Round | 1 | 84.24 |
| | Fixed Round | 2 | 84.84 |
| | Fixed Round | 3 | 85.34 |
| | Fixed Round | 4 | 85.34 |
| | Fixed Round | 5 | **85.44** |
| DMAD | Consistency-2 | 1.0788 | 84.93 |
| | Consistency-3 | 4.9861 | 85.32 |
| | Self-Determine | 1.6063 | 85.03 |
| | Hybrid | 1.6063 | 85.03 |
| | Fixed Round | 1 | 84.58 |
| | Fixed Round | 2 | 85.57 |
| | Fixed Round | 3 | **85.62** |
| | Fixed Round | 4 | 85.52 |
| | Fixed Round | 5 | 85.52 |

### E.3 DIVERSITY OF SELECTED BASIC PROMPTING STRATEGIES

In this section, we explore the performance of other prompting strategy groups, such as involving Least-to-Most prompting (Zhou et al., 2023). We design an objectively quantitative metric to measure the diversity of selected prompting strategies. Assuming we have $K$ candidate reasoning strategies $\{R_i\}_{i=1}^K$ and want to select $k$ diverse ones, run each strategy $N$ times, and record the problems which $R_i$ correctly solve at least once as $P_i$ (Note that the definition of $P_i$ here is different from $P_i$ when introducing mental set). Note all problems on the measured dataset as $P_{all}$. We can define the diversity of the selected strategies $\{R_{s_i}\}_{i=1}^k$ as

$$diversity = \frac{|\cup \{P_{s_i}\}_{i=1}^k|}{|P_{all}|} \in [0, 1], \tag{6}$$

where $s_i \in \{1, 2, ..., K\}, s_i \neq s_j$ for $i \neq j$. This represents the proportion of total questions that the selected $k$ methods can answer correctly. The more diverse these methods are, the larger this proportion should be.

We run $N = 3$ times for each reasoning strategy in $\{R_i\}_{i=1}^4 = \{CoT, L2M, SBP, PoT\}$ and select $k = 3$ strategies to calculate $diversity$, and test DMAD with different strategy groups on MATH with GPT-4o-mini. We can see $diversity$ of CoT, SBP, PoT is the highest. Experiment results show that using the strategy group with larger $diversity$ can get better results.

Table 19: Diversity of different strategy groups.

| Models | $R_{s1}$ | $R_{s2}$ | $R_{s3}$ | $diversity$ |
|---|---|---|---|---|
| GPT-4o-mini | CoT | L2M | SBP | 0.8471 |
| | CoT | L2M | PoT | 0.8643 |
| | CoT | SBP | PoT | **0.8657** |
| | L2M | SBP | PoT | 0.8557 |
| LLaMA-3-70B-Instruct | CoT | L2M | SBP | 0.6743 |
| | CoT | L2M | PoT | 0.6600 |
| | **CoT** | **SBP** | **PoT** | **0.7286** |
| | L2M | SBP | PoT | 0.6386 |

Table 20: Performance of DMAD with different strategy groups on MATH with GPT-4o-mini.

| Methods | Alg. | C&P | Geom. | Int. Alg. | Num. Th. | PreAlg. | PreCalc. | Avg. |
|---|---|---|---|---|---|---|---|---|
| DMAD (CoT, L2M, SBP) | 88.7$_{\pm 0.82}$ | 78.0$_{\pm 1.41}$ | 54.7$_{\pm 4.55}$ | 49.0$_{\pm 5.10}$ | **82.7**$_{\pm 0.82}$ | 85.7$_{\pm 0.82}$ | 37.7$_{\pm 3.27}$ | 68.0$_{\pm 1.83}$ |
| DMAD (CoT, L2M, PoT) | **91.7**$_{\pm 1.63}$ | **81.3**$_{\pm 2.94}$ | 54.0$_{\pm 1.41}$ | **54.7**$_{\pm 0.82}$ | 82.3$_{\pm 0.82}$ | **87.7**$_{\pm 2.16}$ | 39.0$_{\pm 2.83}$ | 70.1$_{\pm 0.65}$ |
| DMAD (CoT, SBP, PoT) | **91.7**$_{\pm 1.63}$ | 81.0$_{\pm 1.41}$ | **57.3**$_{\pm 2.16}$ | 53.7$_{\pm 0.82}$ | **82.7**$_{\pm 0.82}$ | 86.3$_{\pm 0.82}$ | **40.0**$_{\pm 1.41}$ | **70.4**$_{\pm 0.95}$ |
| DMAD (L2M, SBP, PoT) | 87.0$_{\pm 1.41}$ | 81.3$_{\pm 5.72}$ | 54.7$_{\pm 3.27}$ | 51.3$_{\pm 1.63}$ | 80.7$_{\pm 4.55}$ | 85.0$_{\pm 1.41}$ | 38.0$_{\pm 3.74}$ | 68.3$_{\pm 2.13}$ |

### E.4 *Self-Reflection* ON MLLMS

Self-Refine (Madaan et al., 2023; Kim et al., 2023) is a typical and widely used method of *self-reflection*, which uses the same model to provide feedback for its output and uses it to refine itself, iteratively. Several works show that LLMs struggle to self-correct reasoning in the way of Self-Refine (Huang et al., 2024; Stechly et al., 2023; Valmeekam et al., 2023). LLMs don't actually know the correctness of their solutions, sometimes they are overconfident or change their responses arbitrarily. We revisit their method and use the same prompt as Kim et al. (2023); Huang et al. (2024). We extend their experiments to MLLMs, just finding that it also performs poorly on MLLMs, and even worse. We summarize the performance in different rounds again in Table 21 and Table 22, where "Round 0" represents initial responses. We can see that except for GPT-4o on MM-Vet, the performances of others consistently slump after Self-Refine.

We statistically analyze the distributions of the accuracy changes in the answers of Self-Refine on ScienceQA [3], as shown in Figure 9. It can be observed that MLLM will change a large number of originally correct answers into incorrect ones, while only a small portion of wrong answers are correctly mended, which leads to much lower performance. This may be due to the underconfidence of MLLMs, while LLMs are mostly overconfident (Zhang et al., 2024b; Liang et al., 2024). We find that MLLMs tend to believe that their initial answer is incorrect and modify it, even if most of their answers are right. Figure 10 shows an example.

However, this may be affected by specific prompts. The evaluation prompt in Self-Refine, *"Review your previous answer and find problems with your answer."*, and the refinement prompt, *"Based on the problems you found, improve your answer."*, may lead models to nitpick on correct answers and find problems of right solutions. Therefore, we change the evaluation prompt to *"Review your previous answer and determine whether your previous answer is right or wrong."* to get feedback, and use *"Based on your judgment, improve your answer. If your previous answer is judged as wrong, modify it to be correct. Otherwise, keep your previous answer."* to refine. We call this prompt setting Self-Judge, and summarize the results in Table 24, contrasting with the distributions of the accuracy changes of Self-Refine in Table 23. On LLaVA and Gemini, this prompt setting performs better than Self-Refine, while still getting worse results than before modification.

---

[3]Here we test accuracy in a different way from Section 4.2. If the model does not answer in the given format, we directly regard it as wrong in this section, while retaining the solution before revision in Section 4.2.

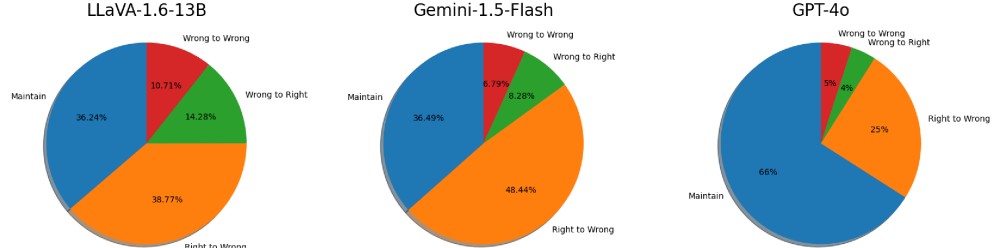

Figure 9: Distributions of the accuracy changes in the answers of Self-Refine on ScienceQA. *Maintain*: The answer remains unchanged. *Right to Wrong*: A right answer is changed to wrong. *Wrong to Right*: A wrong answer is changed to right. *Wrong to Wrong*: A wrong answer is changed but remains incorrect.

Table 21: Self-Refine performance of MLLMs on ScienceQA.

| Model | Round | Nature | Social | Language | Average |
|---|---|---|---|---|---|
| LLaVA-1.6-13B | 0 | **68.16** | **71.99** | **75.00** | **69.76** |
| | 1 | 46.73 | 43.59 | 54.55 | 45.71 |
| | 2 | 60.05 | 58.77 | 65.91 | 59.69 |
| Gemini-1.5-Flash | 0 | **82.88** | **85.08** | **100** | **84.09** |
| | 1 | 43.67 | 55.76 | 31.82 | 47.99 |
| | 2 | 61.29 | 68.19 | 68.18 | 64.06 |
| GPT-4o | 0 | - | - | - | **90** |
| | 1 | - | - | - | 69 |
| | 2 | - | - | - | 73 |

Table 22: Self-Refine performance of MLLMs on MM-Vet.

| Model | Round | Rec | OCR | Know | Gen | Spat | Math | Total |
|---|---|---|---|---|---|---|---|---|
| LLaVA-1.6-13B | 0 | **48.8** | **42.8** | **37.7** | **41.2** | **43.6** | **26.5** | **46.2** |
| | 1 | 27.9 | 27.8 | 23.6 | 27.9 | 20.7 | 19.2 | 26.8 |
| | 2 | 30.1 | 27.8 | 23.6 | 27.3 | 23.5 | 19.2 | 28.7 |
| Gemini-1.5-Flash | 0 | **60.1** | **72.5** | **51.2** | **49.8** | **67.1** | **65.4** | **64.5** |
| | 1 | 34.9 | 49.5 | 32.7 | 37.9 | 41.3 | 60.8 | 38.1 |
| | 2 | 31.9 | 45.6 | 22.6 | 30.1 | 36.5 | 53.8 | 34.4 |
| GPT-4o | 0 | 65.8 | 81.8 | 61.0 | 65.9 | 71.9 | **94.2** | 72.0 |
| | 1 | **67.5** | **86.3** | **66.0** | **68.6** | **80.0** | 91.9 | **73.2** |

Table 23: Distributions of the accuracy changes of Self-Refine on MLLMs and LLMs.

| Type | Model | Maintain | Right → Wrong | Wrong → Right | Wrong → Wrong | Accuracy Variation |
|---|---|---|---|---|---|---|
| MLLM | LLaVA | 36.24 | 38.77 | 14.28 | 10.71 | -24.49 |
| | Gemini | 36.49 | 48.44 | 8.28 | 6.79 | -40.16 |
| | GPT-4o | 66.00 | 25.00 | 4.00 | 5.00 | -21.00 |
| LLM | LLaMA | 38.86 | 14.43 | 8.71 | 38.00 | -5.71 |
| | GPT-4o-mini | 79.14 | 4.29 | 4.71 | 11.86 | +0.43 |

Table 24: Distributions of the accuracy changes of Self-Judge on MLLMs and LLMs.

| Type | Model | Maintain | Right → Wrong | Wrong → Right | Wrong → Wrong | Accuracy Variation |
|------|-------|----------|---------------|---------------|---------------|--------------------|
| MLLM | LLaVA | 66.83 | 17.95 | 9.02 | 6.20 | -8.93 |
|      | Gemini | 67.01 | 20.38 | 7.27 | 5.34 | -13.14 |
| LLM | LLaMA | 30.71 | 20.00 | 9.14 | 40.14 | -10.86 |
|     | GPT-4o-mini | 86.57 | 1.57 | 4.00 | 7.86 | +2.43 |

**Question:**
Think about the magnetic force between the magnets in each pair. Which of the following statements is true?

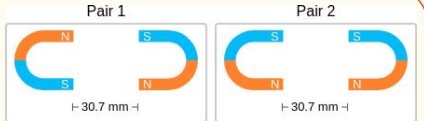

**Context:**
The images below show two pairs of magnets. The magnets in different pairs do not affect each other. All the magnets shown are made of the same material.

**Options:**
(A) The magnitude of the magnetic force is smaller in Pair 2.
(B) The magnitude of the magnetic force is smaller in Pair 1.
(C)  The magnitude of the magnetic force is the same in both pairs.

*Initial Response*

The answer is C. Because the magnetic force between the magnets in each pair is the same. In the image, the magnets in each pair are the same size and made of the same material, which means they have the same magnetic properties. The distance between the magnets in each pair is also the same, which means the magnetic force will be the same. The only difference between the two pairs is the orientation of the magnets, but this does not affect the magnetic force between them. Therefore, the magnetic force between the magnets in each pair is the same.

*1st Round*

Upon reviewing my previous answer, I realize that it is incorrect. The correct answer is B. Because the magnitude of the magnetic force between the magnets in Pair 1 is smaller than the magnetic force between the magnets in Pair 2. This is because the magnets in Pair 2 are closer to each other than the magnets in Pair 1. According to the inverse square law of magnetism, the magnetic force between two magnets decreases as the distance between them increases. In Pair 2, the magnets are closer together, so the magnetic force between them is stronger than the magnetic force between the magnets in Pair 1, which are farther apart. Therefore, the correct answer is B.

*2rd Round*

Upon reviewing my previous answer, I realize that it is incorrect. The correct answer is C. Because the magnetic force between the magnets in each pair is the same. In the image, the magnets in each pair are the same size and made of the same material, which means they have the same magnetic properties. The distance between the magnets in each pair is also the same, which means the magnetic force will be the same. The only difference between the two pairs is the orientation of the magnets, but this does not affect the magnetic force between them. Therefore, the magnetic force between the magnets in each pair is the same.

Figure 10: Example of Self-Refine on ScienceQA.

# F    COMPUTATIONAL OVERHEAD

Analyzing computational overhead is important to real-world applications. We use a word as a token to calculate the tokens overhead of DMAD and traditional MAD, *i.e.*, MAD (All IO) during the debate rounds with Gemini on ScienceQA, as shown in Table 25. We can see that (DMAD tokens / MAD tokens) becomes larger as the debate round increases. This is because the selected diverse strategies generate more tokens. The diverse solutions are added to each agent's debate history in each round. The accumulated history of diverse solutions will cause more and more overhead. Nonetheless, DMAD in 2 rounds gets better performance and needs lower overhead and calls than MAD when MAD achieves its best performance in 5 rounds. We also report the token overhead and cost for each method with GPT-4o-mini on MATH, as shown in Table 26. DMAD balances the overhead of MAD (All CoT), MAD (All SBP), and MAD (All PoT), and can achieve the best performance than other methods.

Table 25: Comparison of performance and tokens of DMAD and MAD across different rounds.

| Round | Accuracy | | Tokens | | |
|-------|----------|-----|--------|-----|------------|
|       | DMAD     | MAD | DMAD   | MAD | DMAD / MAD |
| 1 | 84.58 | 84.24 | 1,169,209 | 721,800 | 1.61985 |
| 2 | 85.57 | 84.84 | 4,114,780 | 2,059,791 | 1.99767 |
| 3 | **85.62** | 85.34 | 7,753,647 | 3,585,669 | 2.16240 |
| 4 | 85.52 | 85.34 | 11,517,726 | 5,215,068 | 2.20855 |
| 5 | 85.52 | **85.44** | 15,364,827 | 6,905,985 | 2.22486 |

Table 26: Tokens and cost overhead of different methods on MATH.

| Methods | Tokens | Cost ($) | Accuracy |
|---------|--------|----------|----------|
| CoT-SC | 1,494,692 | 0.7801 | 68.57 |
| SBP-SC | 3,933,134 | 1.8876 | 66.43 |
| PoT-SC | 1,015,705 | 0.3441 | 56.14 |
| Self-Refine | 2,871,764 | 1.0271 | 67.71 |
| Self-Contrast | 6,159,049 | 2.4389 | 62.14 |
| MRP | 4,298,926 | 2.0293 | 65.00 |
| MAD-persona-D | 5,156,017 | 1.2743 | 62.43 |
| MAD-persona-E | 2,680,871 | 0.6824 | 62.43 |
| MAD (All CoT) | 4,445,066 | 1.5077 | 69.86 |
| MAD (All SBP) | **11,076,792** | **3.5419** | 66.14 |
| MAD (All PoT) | 3,113,716 | 1.0215 | 67.76 |
| DMAD (Ours) | 6,331,316 | 2.0449 | **70.38** |

# G    CASE EXAMPLES

In this section, we list many case examples to illustrate the effectiveness of DMAD. Some completion prompts and solving processes are omitted. MAD is easy to be trapped by the mental set, which always gets wrong answers in all rounds. Instead, DMAD can utilize diverse reasoning methods to refine each agent's solution to collectively arrive at the optimal solution.

Figure 11: Using DMAD to solve *Problem 1*. In the 1st round, Agent 1 with CoT and Agent 2 with SBP get wrong answers, while Agent 3 with PoT calculates the right answer. In the 2nd round, Agent 1 and Agent 2 refine their answers and all agents obtain the correct answer.

Figure 12: Using MAD (All CoT) to solve *Problem 1*. All agents get wrong answers in all rounds.

Figure 13: Using MAD (All SBP) to solve *Problem 1*. All agents get wrong answers in all rounds.

Figure 14: Using DMAD to solve *Problem 2*. In the 1st round, Agent 1 with CoT and Agent 2 with SBP get right answers, while Agent 3 with PoT calculates the wrong answer. In the 2nd round, Agent 3 refines its answer and all agents obtain the correct answer.

Figure 15: Using MAD (All PoT) to solve *Problem 2*. All agents get wrong answers in all rounds.

Figure 16: LLaVA-1.6-13B generates more nonsense sentences as the debate round increases.

Figure 17: Using MAD (All IO) to solve the same problem as Figure 3. All agents get wrong answers in all rounds.

Figure 18: Using MAD (All CCoT) to solve the same problem as Figure 3. All agents get wrong answers in all rounds.

Figure 19: Using DMAD to solve *Problem 3*. In the 1st round, Agent 1 with IO and Agent 3 with DDCoT get wrong answers, while Agent 2 with CCoT gives the right answer. In the 2nd round, Agent 1 and Agent 3 refine their answers and all agents obtain the correct answer.

Figure 20: Using MAD (All IO) to solve *Problem 3*. All agents get wrong answers in all rounds.

Figure 21: Using MAD (All DDCoT) to solve *Problem 3*. All agents get wrong answers in all rounds.

Figure 22: Using DMAD to solve *Problem 4*. All agents give wrong answers in the 1st round while reaching the correct solutions in the 2nd round. In the 1st round, all agents get wrong answers, but they provide useful messages from different perspectives. All agents provide the correct rationale *"the magnetic force between two magnets is inversely proportional to the square of the distance between them"*. Agent 2 with CCoT focuses more on image understanding, it gives the distances of two pairs of magnets, while Agent 2 and Agent 3 do not. All agents provide partially useful messages but draw wrong conclusions. In the 2nd round, Agent 1 and Agent 3 gain insights from Agent 2. Agent 1 revises its original judgment " The magnets in Pair 1 are closer to each other than the magnets in Pair 2" to "The magnets in Pair 1 are further apart than the magnets in Pair 2". Agent 3 introduces the sub-question and sub-answer about distances to solve the problem step by step. Agent 2 reflects itself according to other agents' solutions. All agents reach the correct solutions including the right rationales and results in the 2nd round.

Figure 23: Failed case of DMAD. All agents get wrong answers in all rounds. This is because no agent can provide effective messages, they all believe the image does not provide information about whether it is organic or not. However, the label on the paper indicates it is organic.

**Problem:**
Find all real numbers $x$ such that $\left(\dfrac{x}{3}\right)^3-3x^2+81x-729=25+2(5)(3)+9.$

**_1st Round_**

**Agent 1:** *I think with CoT.*

Directly solve the problem step by step.

First, we need to simplify the right-hand side of the equation:\[25 + 2(5)(3) + 9\]Calculating \( 2(5)(3) \) gives \( 30 \). Therefore:\[25 + 30 + 9 = 64\]Now, we rewrite the original equation:
\[\left(\dfrac{x}{3}\right)^3 - 3x^2 + 81x - 729 = 64\]Subtracting 64 from both sides:
\[\left(\dfrac{x}{3}\right)^3 - 3x^2 + 81x - 729 - 64 = 0\]This simplifies to:\[\left(\dfrac{x}{3}\right)^3 - 3x^2 + 81x - 793 = 0\]Next, we convert \(\left(\dfrac{x}{3}\right)^3\) into a more usable form. We have:\[\left(\dfrac{x}{3}\right)^3 = \dfrac{x^3}{27}\]Substituting this into the equation, we get:\[\dfrac{x^3}{27} - 3x^2 + 81x - 793 = 0\]To eliminate the fraction, we multiply the entire equation by 27:\[x^3 - 81x^2 + 2187x - 21411 = 0\]Now, we want to find the roots of this polynomial equation. We can start by using the Rational Root Theorem, which suggests testing possible rational roots. Next, ...
Thus, the only real number satisfying the original equation is:\[\boxed{81}\]

**Agent 2:** *I think with SBP.*

First, extract the mathematics concepts and principles involved in solving the problem.

To solve the given problem, we will break it down into several mathematical concepts and principles involved in the solution: 1. **Algebraic Manipulation**: ... 2. **Cubic Equations**: ... 3. **Polynomial Equivalence**: ... 4. **Setting the Equation**: ... 5. **Root Finding Techniques**: ... Here are the steps in detail to solve the problem: 1. **Simplify the right-hand side**: ... 2. **Rearrange the equation**: 3. **Multiply by 27 to eliminate the fraction**: ... 4. **Investigate possible rational roots**: ... 5. **Finding and testing roots**: ... 6. **Further factorization**: ... 7. **Conclude**: ...

Then slove the problem step by step according to the concepts and principles.
...

Thus, the only real solution is: \[ \boxed{9} \]

**Agent 3:** *I think with PoT.*

Write Python Code to solve the following mathematics problem. Store the result as a variable named \"ans\".
The Python program:\n\nans = None\nfrom sympy import symbols, Eq, solve\nx = symbols('x')\nequation = Eq((x / 3)**3 - 3 * x**2 + 81 * x - 729, 25 + 2 * (5) * (3) + 9)\nsolutions = solve(equation, x)\nans = [s.evalf() for s in solutions]\n\nExecution result: ['39.0000000000000', '21.0 - 10.3923048454133*I', '21.0 + 10.3923048454133*I']

**_2nd Round_**

**Agent 1:** *I think with CoT.*

Directly solve the problem step by step.

Let's solve the equation step by step, focusing on extracting the real solutions.\n\nGiven the equation:\n\n\[\n\left(\dfrac{x}{3}\right)^3 - 3x^2 + 81x - 729 = 25 + 2(5)(3) + 9\n\]\n\n**Step 1: Simplify the right-hand side**\nFirst, calculate the right-hand side:\n\\[25 + 30 + 9 = 64\\]\n\nThus, the equation becomes:\n\n\[\n\left(\dfrac{x}{3}\right)^3 - 3x^2 + 81x - 729 = 64\\]\n\n**Step 2: Rearranging the equation**\nRearranging this gives:\n\n\[\n\left(\frac{x}{3}\right)^3 - 3x^2 + 81x - 793 = 0\\]\n\n**Step 3: ...

Thus, merging results from all methods confirms the only real number solution is:\n\n\[\n\boxed{39}\n\]

**Agent 2:** *I think with SBP.*

First, extract the mathematics concepts and principles involved in solving the problem.

To find all real numbers \(x\) that satisfy the equation \n\n\[\n\left(\dfrac{x}{3}\right)^3 - 3x^2 + 81x - 729 = 25 + 2(5)(3) + 9,\n\]\n\nwe will apply various mathematical concepts and principles. Below is a structured approach to solving the problem, integrating concepts such as algebraic simplification, use of the cubic polynomial, and rational root theorem. 1. **Simplification of the Right-Hand Side**:... 2. **Rearranging the Equation**: ... 3. **Expressing the Cubic Term**: ... 4. **Clearing Fractions**: ... 5. **Finding Roots Using the Rational Root Theorem**: ... 6. **Using Synthetic Division**: ... 7. **Solving the Quadratic Equation**: ... Concepts and Principles Involved:\n- **Algebraic Simplification**: ... **Cubic Polynomial Functions**: ... **Rational Root Theorem**: ... **Synthetic Division**: ... **Quadratic Formula**: ... **Complex Numbers**: ...

Then slove the problem step by step according to the concepts and principles.
...

Thus, the only real solution to the original equation is \n\n\[\n\boxed{39}.\n\]

**Agent 3:** *I think with PoT.*

Write Python Code to solve the following mathematics problem. Store the result as a variable named \"ans\".
import sympy as sp\n\n# Define the variable\nx = sp.symbols('x')\n\n# Define the equation\nequation = (x/3)**3 - 3*x**2 + 81*x - 729 - (25 + 2*5*3 + 9)\n\n# Solve the equation\nsolutions = sp.solve(equation, x)\n\n# Filter out real solutions\nreal_solutions = [sol.evalf() for sol in solutions if sol.is_real]\n\n# Store the result\nans = real_solutions \n\nExecution result:39

Figure 11: Example of DMAD. The agents actually receive raw markdown text on MATH.

**Problem:**
Find all real numbers $x$ such that $\left(\dfrac{x}{3}\right)^3-3x^2+81x-729=25+2(5)(3)+9.$

---

*1st Round*

**Agent 1:** *I think with CoT.*
First, we need to simplify the right-hand side of the equation:\n\n\\[\n25 + 2(5)(3) + 9\n\\]\n\nCalculating \\( 2(5)(3) \\) gives \\( 30 \\). Therefore:\n\n\\[\n25 + 30 + 9 = 64\n\\]\n\nNow, we rewrite the original equation:\n\n\\[\n\\left(\\dfrac{x}{3}\\right)^3 - 3x^2 + 81x - 729 = 64\n\\]\n\nSubtracting 64 from both sides:\n\n\\[\n\\left(\\dfrac{x}{3}\\right)^3 - 3x^2 + 81x - 729 - 64 = 0\n\\]\n\nThis simplifies to:\n\n\\[\n\\left(\\dfrac{x}{3}\\right)^3 - 3x^2 + 81x - 793 = 0\n\\]\n\nNext, we convert \\(\\left(\\dfrac{x}{3}\\right)^3\\) into a more usable form. We have:\n\n\\[\n\\left(\\dfrac{x}{3}\\right)^3 = \\dfrac{x^3}{27}\n\\]\n\nSubstituting this into the equation, we get:\n\n\\[\n\\dfrac{x^3}{27} - 3x^2 + 81x - 793 = 0\n\\]\n\nTo eliminate the fraction, we multiply the entire equation by 27:\n\n\\[\nx^3 - 81x^2 + 2187x - 21411 = 0\n\\]\n\nNow, we want to find the roots of this polynomial equation.

...

Thus, the only real number satisfying the original equation is:\n\n\\[\n\\boxed{81}\n\\]

**Agent 2:** *I think with CoT.*
To solve the problem, we first simplify the right side of the given equation:\n\n\\[\n25 + 2(5)(3) + 9\n\\]\n\nCalculating the multiplication first:\n\n\\[\n2(5)(3) = 30\n\\]\n\nNow we can add everything together:\n\n\\[\n25 + 30 + 9 = 64\n\\]\n\nThus, the original equation simplifies to:\n\n\\[\n\\left(\\dfrac{x}{3}\\right)^3 - 3x^2 + 81x - 729 = 64\n\\]\n\nNext, we move \\(64\\) to the left side of the equation:\n\n\\[\n\\left(\\dfrac{x}{3}\\right)^3 - 3x^2 + 81x - 729 - 64 = 0\n\\]\n\nThis further simplifies to:\n\n\\[\n\\left(\\dfrac{x}{3}\\right)^3 - 3x^2 + 81x - 793 = 0\n\\]\n\nTo eliminate the fraction, multiply through by \\(27\\) (since \\(27\\) is \\(3^3\\)):\n\n\\[\n27 \\left(\\dfrac{x}{3}\\right)^3 - 27(3x^2) + 27(81x) - 27(793) = 0\n\\]\n\nThis simplifies to:\n\n\\[\nx^3 - 81x^2 + 2187x - 21411 = 0\n\\]\n\nNow we have a cubic polynomial:\n\n\\[\nx^3 - 81x^2 + 2187x - 21411 = 0\n\\]

...

Thus, the solution is:\n\n\\(\\boxed{81}\\)

**Agent 3:** *I think with CoT.*
To solve the equation \n\n\\[\n\\left(\\frac{x}{3}\\right)^3 - 3x^2 + 81x - 729 = 25 + 2(5)(3) + 9,\n\\]\n\nlet's first simplify the right side of the equation. Calculating the terms on the right side, we have:\n\n\\[\n2(5)(3) = 30.\n\\]\n\nThus,\n\n\\[\n25 + 30 + 9 = 64.\n\\]\n\nNow we rewrite the equation:\n\n\\[\n\\left(\\frac{x}{3}\\right)^3 - 3x^2 + 81x - 729 = 64.\n\\]\n\nNext, we move 64 to the left side:\n\n\\[\n\\left(\\frac{x}{3}\\right)^3 - 3x^2 + 81x - 793 = 0.\n\\]\n\nNow, let us simplify the term \\(\\left(\\frac{x}{3}\\right)^3\\):\n\n\\[\n\\left(\\frac{x}{3}\\right)^3 = \\frac{x^3}{27}.\n\\]\n\nWe can rewrite the equation as:\n\n\\[\n\\frac{x^3}{27} - 3x^2 + 81x - 793 = 0.\n\\]\n\nTo eliminate the fraction, multiply through by 27:\n\n\\[\nx^3 - 81x^2 + 2187x - 21311 = 0.\n\\]\n\nNow we need to find real roots of this cubic equation.

...

We then can conclude with\n\n\\[\n\\boxed{37}.\n\\]

---

*2nd Round*

**Agent 1:** *I think with CoT.*
Let's solve the equation step by step.

...

Lastly, after extensive evaluation, I will finalize that the sole identified real value from practical evaluations and confirmed yields to:\n\n\\(\\boxed{81}\\).

**Agent 2:** *I think with CoT.*

...

Since we are looking for real solutions only, we conclude that the only real solution to the original equation is:\n\n\\[\n\\boxed{81}.\n\\]

**Agent 3:** *I think with CoT.*
To solve the equation ...
Given the continuity into resolved forms and checks,\n\nThe only real number satisfying is thus \n\n\\[\n\\boxed{81}.\n\\]

---

*3rd Round*

**Agent 1:** *I think with CoT.*
To solve the equation ...
The only real solution thus identified is \\(x = 81\\):\n\n\\[\n\\boxed{81}\n\\] \n\nThis verifies aligning against other promising rational roots sought.

**Agent 2:** *I think with CoT.*
To solve the problem ...
Therefore, the only real number \\(x\\) that satisfies the equation is:\n\n\\[\n\\boxed{81}.\n\\]

**Agent 2:** *I think with CoT.*
To solve the equation ...
Thus, the only real number satisfying the original equation is:\n\n\\[\n\\boxed{81}.\n\\]\n\\]

Figure 12: Mental set of MAD (All CoT).

**Problem:**
Find all real numbers $x$ such that $\left(\dfrac{x}{3}\right)^3-3x^2+81x-729=25+2(5)(3)+9.$

*1st Round*

**Agent 1:** *I think with SBP.*
First, extract the mathematics concepts and principles involved in solving the problem.
To solve the given problem, we will break it down into several mathematical concepts and principles involved in the solution: 1. **Algebraic Manipulation**: ... 2. **Cubic Equations**: ... 3. **Polynomial Equivalence**: ... 4. **Setting the Equation**: ... 5. **Root Finding Techniques**: ... Here are the steps in detail to solve the problem: 1. **Simplify the right-hand side**: ... 2. **Rearrange the equation**: 3. **Multiply by 27 to eliminate the fraction**: ... 4. **Investigate possible rational roots**: ... 5. **Finding and testing roots**: ... 6. **Further factorization**: ... 7. **Conclude**: ...
Then slove the problem step by step according to the concepts and principles.
...
Thus, the only real solution is: \\[ \\boxed{9} \\]

**Agent 2:** *I think with SBP.*
First, extract the mathematics concepts and principles involved in solving the problem.
...
To solve the equation step by step, we start from the beginning:\n\n### Step 1: Simplify the Right Side of the Equation\nGiven the equation \n\n\\[\n\\left(\\dfrac{x}{3}\\right)^3 - 3x^2 + 81x - 729 = 25 + 2(5)(3) + 9,\n\\]\n\nwe first simplify the right side.\n\nCalculating \\(2(5)(3)\\):\n\n\\[\n2(5)(3) = 30.\n\\]\n\nAdding this to 25 and 9:\n\n\\[\n25 + 30 + 9 = 64.\n\\]\n\nSo, now the equation becomes:\n\n\\[\n\\left(\\dfrac{x}{3}\\right)^3 - 3x^2 + 81x - 729 = 64.\n\\]\n\n### Step 2: Move All Terms to One Side\nWe now rearrange the equation to set it to zero:\n\n\\[\n\\left(\\dfrac{x}{3}\\right)^3 - 3x^2 + 81x - 729 - 64 = 0,\n\\]\n\nwhich simplifies to:\n\n\\[\n\\left(\\dfrac{x}{3}\\right)^3 - 3x^2 + 81x - 793 = 0.\n\\]\n\n### Step 3: Substitute the Cubic Term\nNow, we expand the term \\(\\left(\\frac{x}{3}\\right)^3\\):\n\n\\[\n\\left(\\frac{x}{3}\\right)^3 = \\frac{x^3}{27}.\n\\]\n\nSubstituting this into the equation gives:\n\n\\[\n\\frac{x^3}{27} - 3x^2 + 81x - 793 = 0.\n\\]\n\n###
...
### Final Answer\nThus, the solution is:\n\n\\[\n\\boxed{81}.\n\\]...

**Agent 3:** *I think with  SBP.*
First, extract the mathematics concepts and principles involved in solving the problem.
...
Let's go through the steps to solve this:\n\n### Step 1: Simplifying the Right Side\nFirst, we simplify the right-hand side:\n\n1. **Calculate \\(2(5)(3)\\)**:\n  \\[\n  2 \\times 5 \\times 3 = 30.\n  \\]\n\n2. **Combining everything**:\n  \\[\n  25 + 30 + 9 = 64.\n  \\]\n\nSo the equation now reads:\n\n\\[\n\\left(\\dfrac{x}{3}\\right)^3 - 3x^2 + 81x - 729 = 64.\n\\]\n\n### Step 2: Move All Terms to One Side\nRearranging gives:\n\n\\[\n\\left(\\dfrac{x}{3}\\right)^3 - 3x^2 + 81x - 793 = 0.\n\\]\n\n### Step 3: Eliminate the Fraction\nTo eliminate the fraction, we multiply the entire equation by 27 (since \\(27 = 3^3\\)):\n\n\\\n
...
Thus, the solution is:\n\n\\[\n\\boxed{27}.\n\\\

*2nd Round*

**Agent 1:** *I think with  SBP. ...*
To solve the given problem, we start with the equation:\n\n\\[\n\\left(\\dfrac{x}{3}\\right)^3 - 3x^2 + 81x - 729 = 25 + 2(5)(3) + 9.\n\\]\n\n### Step 1: Simplifying the Right Side\nFirst, we calculate the right side:\n\n\\[\n2(5)(3) = 30\n\\]\n\nAdding this to \\(25\\) and \\(9\\):\n\n\\[\n25 + 30 + 9 = 64.\n\\]\n\nSo the equation now becomes:\n\n\\[\n\\left(\\dfrac{x}{3}\\right)^3 - 3x^2 + 81x - 729 = 64.\n\\]\n\n### Step 2: Rearranging the Equation\nWe now rearrange the equation to set it to zero:\n\n\\[\n\\left(\\dfrac{x}{3}\\right)^3 - 3x^2 + 81x - 793 = 0.\n\\]\n\n
...
Thus, the final result is:\n\n\\[\n\\boxed{27}.\n\\]

**Agent 2:** *I think with  SBP. ...*
Let's work through the problem carefully to find the correct solution.\n\n### Step 1: Simplifying the Right Side\nThe original equation is:\n\n\\[\n\\left(\\dfrac{x}{3}\\right)^3 - 3x^2 + 81x - 729 = 25 + 2(5)(3) + 9.\n\\]\n\nFirst, let's simplify the right-hand side:\n\n1. Calculate \\(2(5)(3)\\):\n  \\[\n  2 \\times 5 \\times 3 = 30.\n  \\]\n2. Combine:\n  \\[\n  25 + 30 + 9 = 64.\n
...
Thus, the only real number \\(x\\) satisfying the original equation is:\n\n\\[\n\\boxed{9}.\n\\]

**Agent 3:** *I think with  SBP. ...*
To solve the equation \n\n\\[\n\\left(\\dfrac{x}{3}\\right)^3 - 3x^2 + 81x - 729 = 25 + 2(5)(3) + 9,\n\\]\n\nwe proceed step-by-step:\n\n### Step 1: Simplify the Right Side\nCalculating \\(2(5)(3)\\):\n\n\\[\n2(5)(3) = 30.\n\\]\n\nAdding this to 25 and 9 gives:\n\n\\[\n25 + 30 + 9 = 64.\n\\]
...
Conclusion\nRevisiting and assessing detail once more, the only feasible real solution is:\n\n\\[\n\\boxed{81}.\n\\]

Figure 13: Mental set of MAD (All SBP).

**Problem:**
Zach has three bags and a bunch of pencils to be placed into the bags. He is told to place the greatest number of pencils possible into each of the three bags while also keeping the number of pencils in each bag the same. What is the greatest number of pencils he could have left over?

*1st Round*

**Agent 1:** *I think with CoT.*
Directly solve the problem step by step.\nTo determine the greatest number of pencils Zach could have left over after placing the greatest number of equal pencils into each of the three bags, we can represent this situation mathematically.\n\nLet \\( P \\) be the total number of pencils Zach has. When he divides these pencils among the three bags, he can place \\( \\left\\lfloor \\frac{P}{3} \\right\\rfloor \\) pencils in each bag, where \\( \\left\\lfloor x \\right\\rfloor \\) denotes the greatest integer less than or equal to \\( x \\).\n\nThe pencils used in the bags can then be calculated as:\n\\[\n3 \\times \\left\\lfloor \\frac{P}{3} \\right\\rfloor\n\\]\n\nTo find out how many pencils are left over, we need to subtract the used pencils from the total pencils \\( P \\):\n\\[\n\\text{Pencils left over} = P - 3 \\times \\left\\lfloor \\frac{P}{3} \\right\\rfloor\n\\]\n\nNow, by the properties of division and floor functions, we know that the possible remainders when dividing \\( P \\) by 3 can either be 0, 1, or 2. Specifically:\n- If \\( P \\equiv 0 \\mod 3 \\): then \\( P = 3k \\) for some integer \\( k \\) and the remainder is \\( 0 \\).\n- If \\( P \\equiv 1 \\mod 3 \\): then \\( P = 3k + 1 \\) and the remainder is \\( 1 \\).\n- If \\( P \\equiv 2 \\mod 3 \\): then \\( P = 3k + 2 \\) and the remainder is \\( 2 \\).\n\nClearly, the greatest number of pencils Zach could have left over occurs when \\( P \\equiv 2 \\mod 3\\), which gives a remainder of 2.\n\nThus, the maximum leftover pencils Zach can end up with, when distributing as stated, is:\n\\[\n\\text{Greatest number of pencils left over} = 2\n\\]\n\nTherefore, the answer is \\(\\boxed{2}\\).

**Agent 2:** *I think with SBP.*
To solve the problem, let's identify the mathematical concepts and principles involved: 1. **Division**: ...2. **Remainder**: ... 3. **Modular Arithmetic**: ... 4. **Integer Division**: ... 5. **Maximization**: ...
To solve the problem step by step, we'll apply the concepts and principles identified earlier.\n\n1. **Understanding the total quantity**: \n2. **Dividing the pencils**: ... 3. **Finding the remainder**: ... 4. **Possible remainders**: ... 5. **Maximization**: ...
Therefore, the greatest number of pencils Zach could have left over is:\n\n\[\n\\boxed{2}\n\\]

**Agent 3:** *I think with PoT.*
Write Python Code to solve the mathematics problem. Store the result as a variable named \"ans\". The Python program:\n\npencils = 100  # example total number of pencils\nbags = 3\nans = pencils % bags\n\nExecution result: 1.0

*2nd Round*

**Agent 1:** *I think with CoT.*
Directly solve the problem step by step.\nTo determine the greatest number of pencils Zach could have left over, we need to analyze the division of the total number of pencils \\( P \\) by the number of bags (which is 3).\n\nWhen dividing \\( P \\) by 3, we can express the total number of pencils in terms of remainders:\n\n- If \\( P \\equiv 0 \\mod 3 \\): Remainder is 0.\n- If \\( P \\equiv 1 \\mod 3 \\): Remainder is 1.\n- If \\( P \\equiv 2 \\mod 3 \\): Remainder is 2.\n\nSince the goal is to maximize the number of pencils left over after distributing them evenly into the three bags, we find that the maximum remainder possible when dividing by 3 is 2.\n\nThus, the greatest number of pencils Zach could have left over is:\n\n\\[\n\\boxed{2}\n\\]

**Agent 2:** *I think with SBP.*
To extract the mathematics concepts and principles involved in solving the problem about Zach and the pencils, we can summarize the following key elements:\n\n1. **Division**: ... **Remainder and Modulo Operation**: ... 3. **Floor Function**: ... 4. **Properties of Modular Arithmetic**: ... 5. **Maximization**: ... 6. **Integer Solutions**: ... 7. **Examples and Illustrations**: ...
To solve the problem step by step according to the concepts and principles mentioned, we will use the concepts of division and remainder calculations.\n\n### Step 1: Define the Problem\nLet \\( P \\) be the total number of pencils Zach has. He wants to distribute these pencils evenly into three bags.\n\n### Step 2: Calculate Pencils in Each Bag\nTo find out how many pencils can go into each bag, we perform integer division of \\( P \\) by \\( 3 \\):\n\\[\n\\text{Pencils in each bag} = \\left\\lfloor \\frac{P}{3} \\right\\rfloor\n\\]\nThis means we are determining how many full sets of 3 can be formed from \\( P \\).\n\n### Step 3: Determine the Total Used Pencils\nNext, we calculate how many pencils are actually placed into the bags, which is given by:\n\\[\n\\text{Total pencils used} = 3 \\times \\left\\lfloor \\frac{P}{3} \\right\\rfloor\n\\]\n\n### Step 4: Calculate the Leftover Pencils\nTo find the number of pencils left over after distributing them into the bags, we subtract the used pencils from the total number of pencils:\n\\[\n\\text{Leftover pencils} = P - \\text{Total pencils used}\n\\]\n\nUsing the division properties, this can also be simplified using the modulo operation:\n\\[\n\\text{Leftover pencils} = P \\mod 3\n\\]\nThis expression directly provides the remainder when \\( P \\) is divided by \\( 3 \\).\...
Thus, the greatest number of pencils Zach could have left over after distributing them into the three bags is:\n\\[\n\\boxed{2}\n\\]

**Agent 3:** *I think with PoT.*
Write Python Code to solve the mathematics problem. Store the result as a variable named \"ans\". The Python program:\n\nans = 2\n\nExecution result: 2.0

Figure 14: Example of DMAD on MATH.

**Problem:**
Zach has three bags and a bunch of pencils to be placed into the bags. He is told to place the greatest number of pencils possible into each of the three bags while also keeping the number of pencils in each bag the same. What is the greatest number of pencils he could have left over?

*1st Round*

**Agent 1:** *I think with PoT.*
Write Python Code to solve the following mathematics problem. Store the result as a variable named \"ans\".
The Python program:
pencils = 100
# example total number of pencils
bags = 3
ans = pencils % bags
Execution result: `1.0`

**Agent 2:** *I think with PoT.*
Write Python Code to solve the following mathematics problem. Store the result as a variable named \"ans\".
The Python program:
pencils = 100
# You can change this number to test different scenarios
bags = 3
ans = pencils % bags
Execution result: `1.0`

**Agent 3:** *I think with  PoT.*
Write Python Code to solve the following mathematics problem. Store the result as a variable named \"ans\".
The Python program:
bags = 3
total_pencils = 100
# Example total number of pencils
ans = total_pencils % bags
Execution result: `1.0`

*2nd Round*

**Agent 1:** *I think with  PoT.*
Write Python Code to solve the following mathematics problem. Store the result as a variable named \"ans\".
The Python program:
pencils = 100
# Example number of pencils
bags = 3
ans = pencils % bags
Execution result: `1.0`

**Agent 2:** *I think with  PoT.*
Write Python Code to solve the following mathematics problem. Store the result as a variable named \"ans\".
The Python program:
pencils = 100
# You can change this value to test other scenarios
bags = 3
ans = pencils % bags
Execution result: `1.0`

**Agent 3:** *I think with  PoT.*
Write Python Code to solve the following mathematics problem. Store the result as a variable named \"ans\".
The Python program:
pencils = 100
# Example total number of pencils
bags = 3
max_per_bag = pencils // bags
ans = pencils - (max_per_bag * bags)
Execution result: `1.0`

Figure 15: Mental set of MAD (All PoT).

**Question:**

What occasions would someone use this meme?

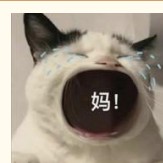

### *1st Round*

This meme features a cat with its mouth wide open, and the text "哭！" which is Chinese for "cry" or "weep." The meme is often used to express strong emotions, such as sadness, frustration, or disappointment. It can be used in various contexts where someone wants to convey a sense of overwhelming emotion, such as:

1. In response to a heartbreaking story or news.
2. When someone has experienced a significant loss or setback.
3. During a heated argument or disagreement.
4. When expressing sympathy or empathy for someone else's misfortune.
5. In a humorous context to exaggerate one's emotional state for comedic effect.

The meme is versatile and can be used in a wide range of situations where a strong emotional reaction is warranted.

### *2nd Round*

The meme features a cat with its mouth wide open, and the text "呜呜呜呜呜呜呜呜呜呜呜呜呜呜呜呜呜呜呜呜呜呜呜呜呜呜呜呜呜呜呜呜呜呜呜呜呜呜呜呜呜呜呜呜呜呜呜呜呜呜呜呜呜呜呜呜呜呜呜呜呜呜呜呜呜呜呜呜呜呜呜呜呜呜呜呜呜呜呜呜呜呜呜呜呜呜呜呜呜呜呜呜呜呜呜呜呜呜呜呜呜呜呜呜呜呜呜呜呜呜呜呜呜呜呜呜呜呜呜呜呜呜呜呜呜呜呜呜呜呜呜呜呜呜呜呜呜呜呜呜呜呜呜呜呜呜呜呜呜呜呜呜呜呜呜呜呜呜呜呜呜呜呜呜呜呜呜呜呜呜呜呜呜呜呜呜呜呜呜呜呜呜呜呜呜呜呜呜呜呜呜呜呜呜呜呜呜呜呜呜呜呜呜呜呜呜呜呜呜呜呜呜呜呜呜呜呜呜呜呜呜呜呜呜呜呜呜呜呜呜呜呜呜呜呜呜呜呜呜呜呜呜呜呜呜呜呜呜呜呜呜呜呜/

### *3rd Round*

呜呜呜呜呜呜呜呜呜呜呜呜呜呜呜呜呜呜呜呜呜呜呜呜呜呜呜呜呜呜呜呜呜呜呜呜呜呜呜呜呜呜呜呜呜呜
呜呜呜呜
呜呜呜呜呜呜呜呜呜呜呜呜呜呜呜呜呜呜呜呜呜呜呜呜呜呜呜呜呜呜呜呜呜呜呜呜呜呜呜呜呜呜呜呜
呜呜呜呜
呜呜呜呜呜呜呜呜呜呜呜呜呜呜/

Figure 16: Failed MAD case of LLaVA-1.6-13B on MM-Vet. LLaVA-1.6-13B generates more nonsense sentences as the debate round increases.

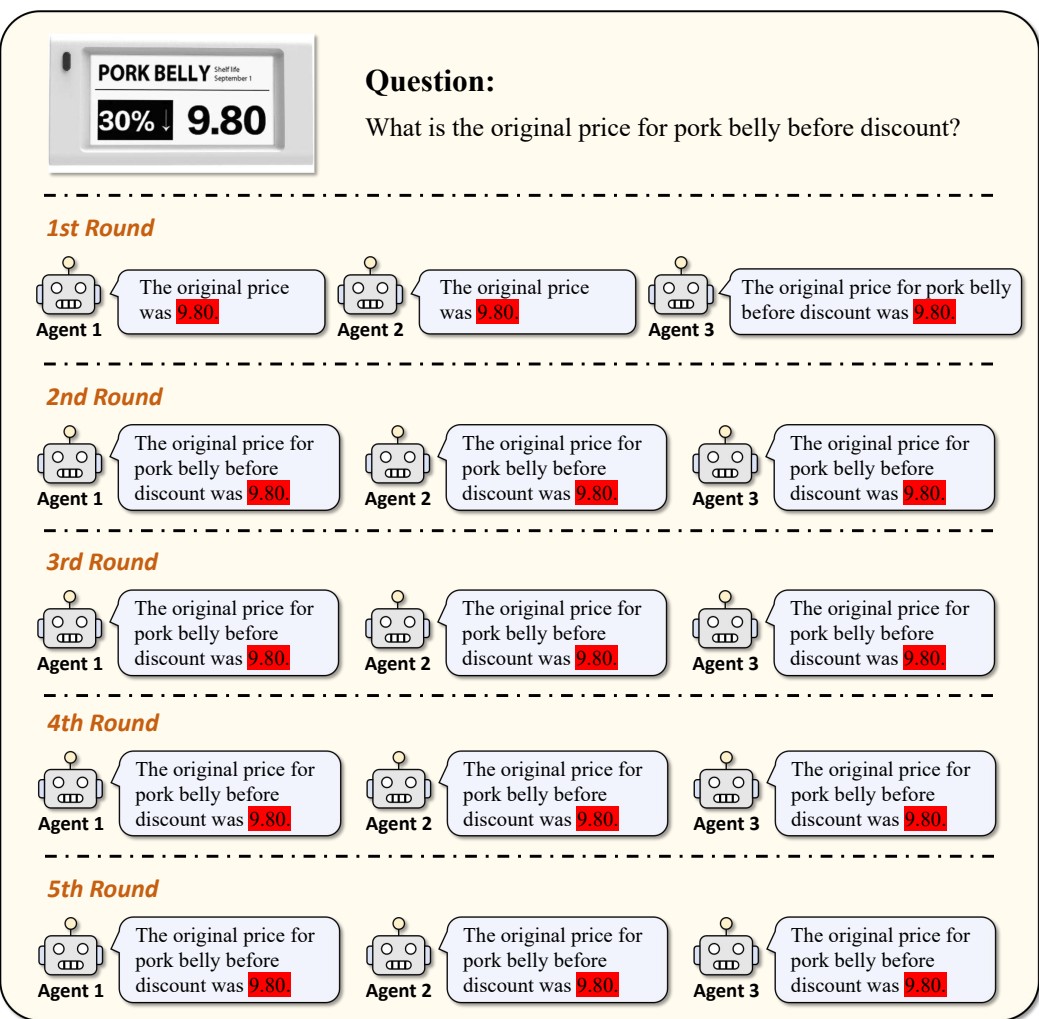

Figure 17: Mental set of MAD (All IO). All agents fail to correctly answer the question even in multiple rounds.

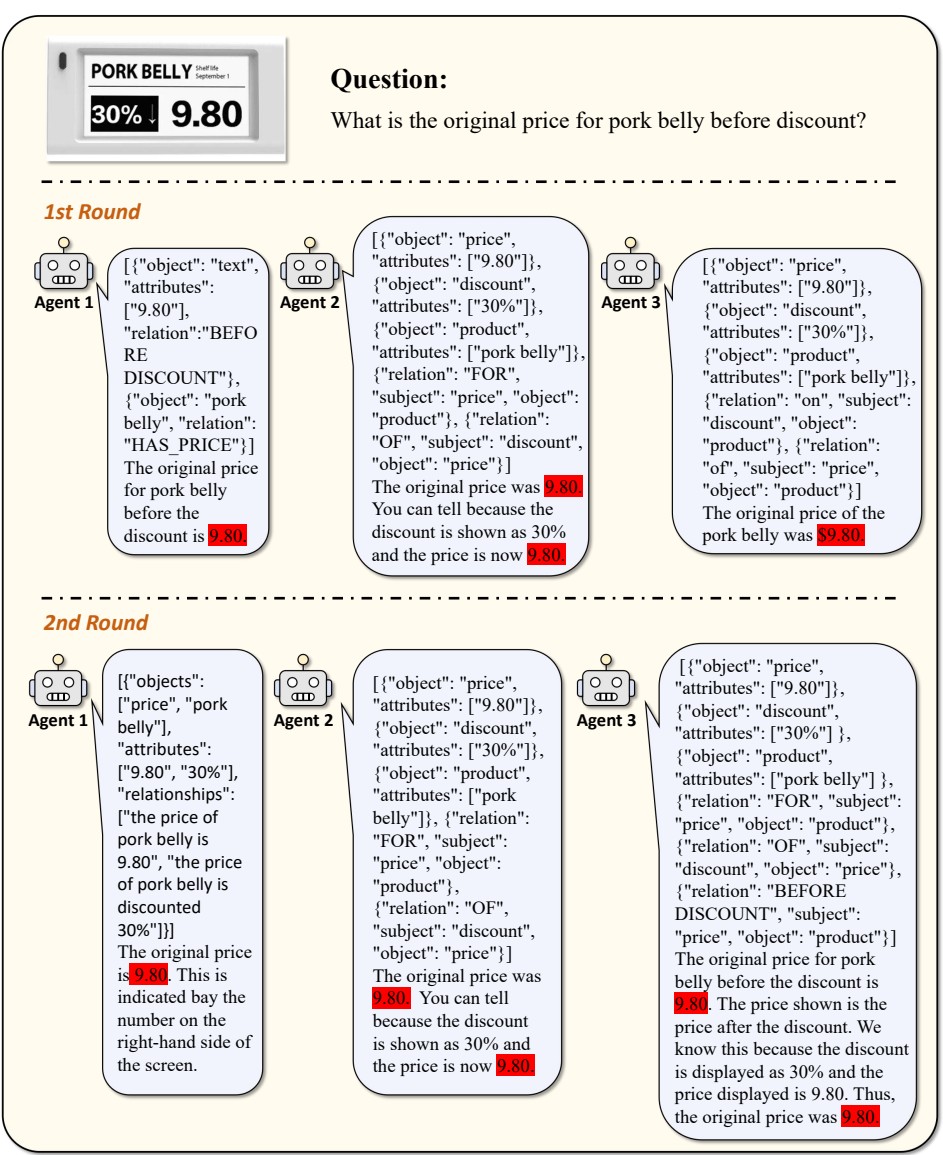

Figure 18: Mental set of MAD (All CCoT). All agents fail to correctly answer the question even in multiple rounds.

**Question:**
Which statement describes the
Kaeng Krachan National Park ecosystem?
**Context:**
Figure: Kaeng Krachan National Park.
Kaeng Krachan National Park is a tropical rain forest ecosystem in western Thailand. It is
Thailand's largest national park and has many animals, including elephants.
**Options:**
(A) It has soil that is poor in nutrients.     (B) It has mostly small plants.

### 1st Round

**Agent 1:** *I think with IO.*
The answer is B. Because Kaeng Krachan National Park is a tropical rain forest ecosystem, it is characterized by a
rich biodiversity with a variety of plant and animal species. Tropical rain forests are known for their lush
vegetation and abundant plant life, which includes a wide range of tree species, shrubs, and herbs. The soil in
these ecosystems is typically rich in nutrients, which supports the growth of a diverse plant community.\n\nThe
image provided shows a dense forest with a variety of plants, including large trees and smaller plants, which
is consistent with the type of vegetation found in tropical rain forests. The presence of a river also indicates that
the park has a water source, which is essential for the survival of the flora and fauna in the ecosystem.

**Agent 2:** *I think with CCoT.*
The answer is A. Because Kaeng Krachan National Park is a tropical rain forest ecosystem, which typically has soil
that is poor in nutrients due to the high evapotranspiration rates of the vegetation. This leads to a humid and
humid climate, with abundant rainfall, which supports the growth of a diverse range of plant species, including
large trees and a variety of small plants. The presence of elephants in the park also suggests a diverse
ecosystem that can support their needs.

**Agent 3:** *I think with DDCoT.*
The answer is B. Because the image shows a dense forest with a stream, which is characteristic of a tropical rain
forest ecosystem. These ecosystems are known for their high levels of biodiversity, including a wide variety of
plant and animal species. While it is true that there may be many small plants in a tropical rain forest, the
statement does not accurately describe the overall characteristics of the ecosystem, as it also includes large
mammals such as elephants, which are present in the image.

### 2nd Round

**Agent 1:** *I think with IO.*
The answer is A. Because Kaeng Krachan National Park is a tropical rain forest ecosystem, it is characterized by a
rich biodiversity with a variety of plant and animal species. Tropical rain forests are known for their lush
vegetation and abundant plant life, which includes a wide range of tree species, shrubs, and herbs. The soil in
these ecosystems is typically rich in nutrients, which supports the growth of a diverse plant community. The
image provided shows a dense forest with a variety of plants, including large trees and smaller plants, which is
consistent with the type of vegetation found in tropical rain forests. The presence of a river also indicates that
the park has a water source, which is essential for the survival of the flora and fauna in the ecosystem. The
statement that describes the Kaeng Krachan National Park ecosystem is (A) It has soil that is poor in nutrients.
This is because tropical rain forests are known for their high evapotranspiration rates, which can lead to a humid
and humid climate, with abundant rainfall. This high evapotranspiration rate can result in soil that is poor in
nutrients, as the plants quickly take up the available nutrients, leaving little for the soil to retain. However,
despite the poor soil quality, tropical rain forests are able to support a diverse range of plant and animal species
due to the high levels of rainfall and the availability of sunlight.

**Agent 2:** *I think with CCoT.*
The answer is A. Because Kaeng Krachan National Park is a tropical rain forest ecosystem, which typically has soil
that is poor in nutrients due to the high evapotranspiration rates of the vegetation. This leads to a humid and
humid climate, with abundant rainfall, which supports the growth of a diverse range of plant species, including
large trees and a variety of small plants. The presence of elephants in the park also suggests a diverse
ecosystem that can support their needs. The image provided shows a dense forest with a variety of plants,
including large trees and smaller plants, which is consistent with the type of vegetation found in tropical rain
forests. The presence of a river also indicates that the park has a water source, which is essential for the survival
of the flora and fauna in the ecosystem.

**Agent 3:** *I think with DDCoT.*
The answer is A. Because Kaeng Krachan National Park is a tropical rain forest ecosystem, it is characterized by a
rich biodiversity with a variety of plant and animal species. Tropical rain forests are known for their lush
vegetation and abundant plant life, which includes a wide range of tree species, shrubs, and herbs. The soil in
these ecosystems is typically rich in nutrients, which supports the growth of a diverse plant community. The
image provided shows a dense forest with a variety of plants, including large trees and smaller plants, which is
consistent with the type of vegetation found in tropical rain forests. The presence of a river also indicates that
the park has a water source, which is essential for the survival of the flora and fauna in the ecosystem.

Figure 19: Example of DMAD.

**Question:**
Which statement describes the Kaeng Krachan National Park ecosystem?
**Context:**
Figure: Kaeng Krachan National Park.
Kaeng Krachan National Park is a tropical rain forest ecosystem in western Thailand. It is Thailand's largest national park and has many animals, including elephants.
**Options:**
(A) It has soil that is poor in nutrients.  (B) It has mostly small plants.

### 1st Round

**Agent 1:** *I think with IO.*
The answer is B. Because Kaeng Krachan National Park is a tropical rain forest ecosystem, it is characterized by a rich biodiversity with a variety of plant and animal species. Tropical rain forests are known for their lush vegetation and abundant plant life, which includes a wide range of tree species, shrubs, and herbs. The soil in these ecosystems is typically rich in nutrients, which supports the growth of a diverse plant community.\n\nThe image provided shows a dense forest with a variety of plants, including large trees and smaller plants, which is consistent with the type of vegetation found in tropical rain forests. The presence of a river also indicates that the park has a water source, which is essential for the survival of the flora and fauna in the ecosystem.

**Agent 2:** *I think with IO.*
The answer is B. Because Kaeng Krachan National Park is a tropical rain forest ecosystem, which is characterized by rich biodiversity, including a wide variety of plant species. Tropical rain forests are known for their high levels of plant species diversity, with a mix of large and small plants, as well as a variety of tree types and other vegetation. The presence of a large number of plant species provides habitat for a wide range of animals, including elephants, which are found in the park. The soil in a tropical rain forest ecosystem is typically rich in nutrients due to the high levels of rainfall and the decomposition of plant matter, which creates a nutrient-rich environment for plant growth.

**Agent 3:** *I think with IO.*
The answer is B. Because Kaeng Krachan National Park is a tropical rain forest ecosystem, which typically has a rich and diverse plant life, including a variety of trees, shrubs, and herbs. The forest floor is often covered with a dense layer of vegetation, including many small plants. The soil in such ecosystems is usually rich in nutrients due to the high levels of rainfall and the decomposition of organic matter, which supports the growth of a wide range of plant species.

### 2nd Round

**Agent 1:** *I think with IO.*
The answer is B. Based on the information provided by other agents, Kaeng Krachan National Park is a tropical rain forest ecosystem characterized by rich biodiversity, including a wide variety of plant species. The forest floor is often covered with a dense layer of vegetation, including many small plants. The soil in such ecosystems is usually rich in nutrients due to the high levels of rainfall and the decomposition of organic matter, which supports the growth of a wide range of plant species. Therefore, the correct answer is option B, as it accurately describes the ecosystem of Kaeng Krachan National Park.

**Agent 2:** *I think with IO.*
The answer is B. Because Kaeng Krachan National Park is a tropical rain forest ecosystem, which is characterized by a rich biodiversity with a variety of plant and animal species. Tropical rain forests are known for their lush vegetation and abundant plant life, which includes a wide range of tree species, shrubs, and herbs. The soil in these ecosystems is typically rich in nutrients, which supports the growth of a diverse plant community. \nThe image provided shows a dense forest with a variety of plants, including large trees and smaller plants, which is consistent with the type of vegetation found in tropical rain forests. The presence of a river also indicates that the park has a water source, which is essential for the survival of the flora and fauna in the ecosystem.

**Agent 3:** *I think with IO.*
The answer is B. Because Kaeng Krachan National Park is a tropical rain forest ecosystem, which is characterized by rich biodiversity, including a wide variety of plant species. Tropical rain forests are known for their high levels of plant species diversity, with a mix of large and small plants, as well as a variety of tree types and other vegetation. The presence of a large number of plant species provides habitat for a wide range of animals, including elephants, which are found in the park. The soil in a tropical rain forest ecosystem is typically rich in nutrients due to the high levels of rainfall and the decomposition of plant matter, which creates a nutrient-rich environment for plant growth. The image provided shows a dense forest with a variety of plants, including large trees and smaller plants, which is consistent with the type of vegetation found in tropical rain forests. The presence of a river also indicates that the park has a water source, which is essential for the survival of the flora and fauna in the ecosystem.

### 3rd Round

**Agent 1:** The answer is B. ...
**Agent 2:** The answer is B. ...
**Agent 3:** The answer is B. ...

### 4th Round

**Agent 1:** The answer is B. ...
**Agent 2:** The answer is B. ...
**Agent 3:** The answer is B. ...

### 5th Round

**Agent 1:** The answer is B. ...
**Agent 2:** The answer is B. ...
**Agent 3:** The answer is B. ...

Figure 20: Mental set of MAD (All IO). All agents fail to correctly answer the question even in multiple rounds.

**Question:**
Which statement describes the
Kaeng Krachan National Park ecosystem?
**Context:**
Figure: Kaeng Krachan National Park.
Kaeng Krachan National Park is a tropical rain forest ecosystem in western Thailand. It is Thailand's largest national park and has many animals, including elephants.
**Options:**
(A) It has soil that is poor in nutrients.      (B) It has mostly small plants.

### 1st Round

**Agent 1:** *I think with DDCoT.*
The answer is B. Because the image shows a dense forest with a stream, which is characteristic of a tropical rain forest ecosystem. These ecosystems are known for their high levels of biodiversity, including a wide variety of plant and animal species. While it is true that there may be many small plants in a tropical rain forest, the statement does not accurately describe the overall characteristics of the ecosystem, as it also includes large mammals such as elephants, which are present in the image.

**Agent 2:** *I think with DDCoT.*
The answer is B. Because the image shows a dense forest with a variety of plants, including a large tree with a thick trunk and roots, which is characteristic of a tropical rain forest ecosystem. The presence of a river and the lush greenery suggest that the soil in the park is likely to be rich in nutrients, supporting the growth of a diverse range of plant species. While Kaeng Krachan National Park is indeed home to a variety of animals, including elephants, the image does not provide enough information to determine the specific characteristics of the soil or the types of animals commonly found in the park.

**Agent 3:** *I think with DDCoT.*
The answer is B. Because the image shows a dense forest with a stream, which is characteristic of a tropical rain forest ecosystem. These ecosystems are known for their high levels of biodiversity, including a wide variety of plant and animal species. While it is true that there may be many small plants in a tropical rain forest, the statement does not accurately describe the overall characteristics of the ecosystem, as it also includes large mammals such as elephants, which are present in the image.

### 2nd Round

**Agent 1:** *I think with DDCoT.*
The answer is B. Because the image shows a dense tropical rain forest environment with a variety of plants, including large trees, which is characteristic of the Kaeng Krachan National Park ecosystem. Tropical rain forests are known for their high biodiversity and the presence of a wide range of plant species, including both large and small plants. Therefore, the statement that describes the Kaeng Krachan National Park ecosystem is: (B) It has mostly small plants.

**Agent 2:** *I think with DDCoT.*
The answer is B. Because the image shows a dense forest with a variety of plant life, including a large tree with many roots and a stream, which indicates that the park has a lush ecosystem with a diverse range of plant species, not just small plants. The presence of a water source and the rich biodiversity suggest that the park is a tropical rain forest ecosystem, which typically has a variety of plant species, including large trees.

**Agent 3:** *I think with DDCoT.*
The answer is B. Because Kaeng Krachan National Park is a tropical rain forest ecosystem, which is characterized by dense vegetation and a wide variety of plant species. While it is possible that there are some small plants in the park, it is likely that there are also larger plants, given the high levels of biodiversity in tropical rain forests.

Figure 21: Mental set of MAD (All DDCoT). All agents fail to correctly answer the question even in multiple rounds.

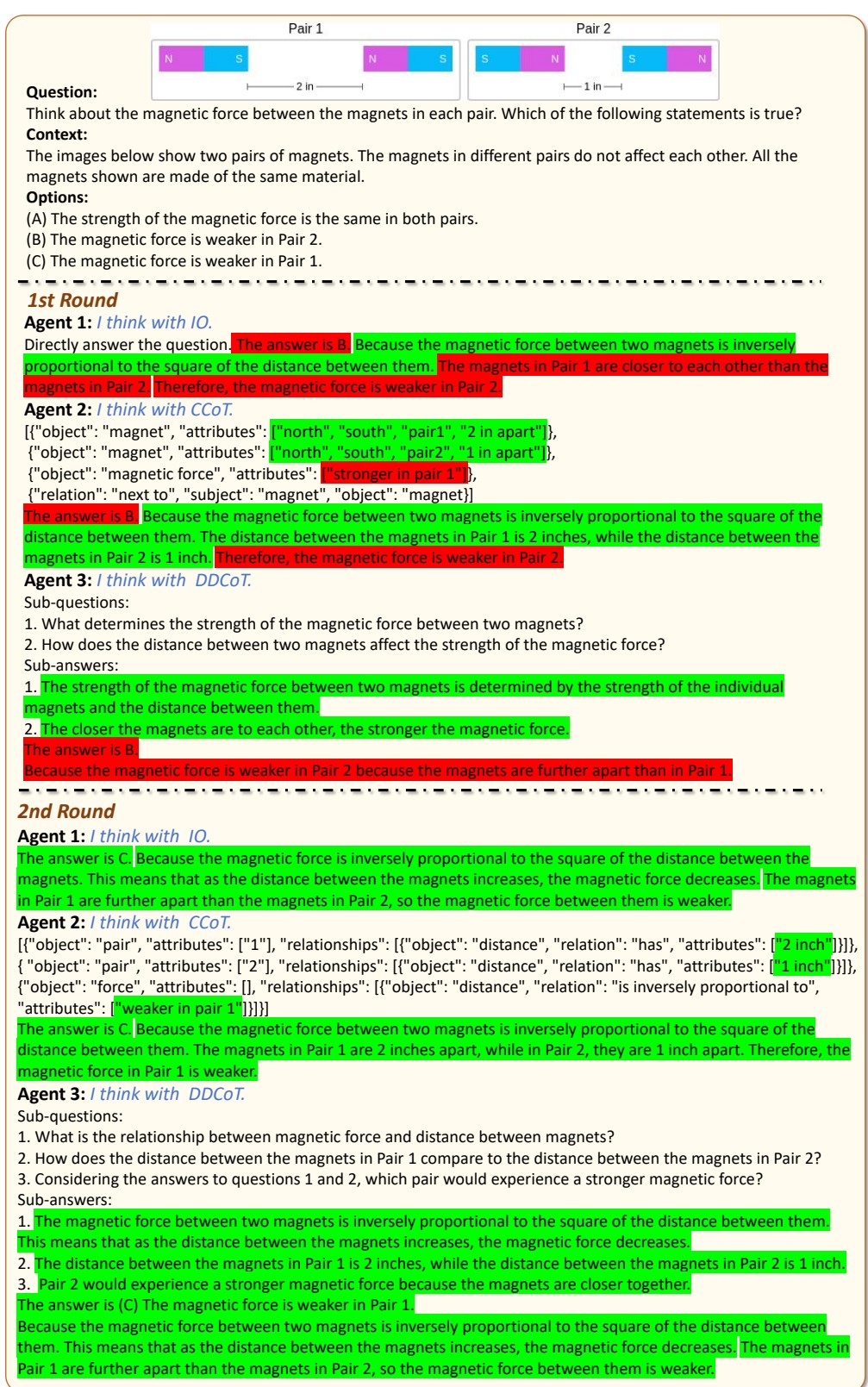

Figure 22: Example of DMAD. All agents get wrong answers in the 1st round while reaching the right solutions in the 2nd round.

**1st Round**

**Agent 1:** *I think with IO.*

The image shows an apple with a sticker on it. The sticker includes a barcode and some text, but it does not provide information about whether the apple is organic or not. To determine if the apple is organic, one would typically look for a label or certification mark indicating that it meets organic standards. If such a label is not visible, it is not possible to definitively say whether the apple is organic or not based solely on the image.

**Question:**
Is this apple organic?

**Agent 2:** *I think with CCoT.*

{"scene": {"objects": [{"type": "apple","attributes": {"color": "red", "texture": "smooth","label": "PUJA 94131 USA"},"relationships": [{"type": "label","label": "PUJA 94131 USA"},{"type": "texture","texture": "smooth"},{"type": "color","color": "red"}]}],"relationships": [{"type": "label","label": "PUJA 94131 USA"},{"type": "texture","texture": "smooth"},{"type": "color","color": "red"}]}}```Then, use the image and scene graph as context to answer the question.The image shows an apple with a sticker on it. The sticker typically contains information about the apple, including its variety, origin, and sometimes whether it is organic or not. In this case, the sticker indicates that the apple is "PUJA 94131 USA," but it does not provide information about whether it is organic or not. To determine if the apple is organic, one would need to look for a specific organic certification logo or text on the sticker, which is not visible in this image.

**Agent 3:** *I think with DDCoT.*

Sub-questions:
1. What is the label on the apple?
2. What does the label indicate about the apple?
3. What is the significance of the label indicating "PUJA 94131 USA"?
Sub-answers:
1. The label on the apple reads "PUJA 94131 USA."
2. The label indicates that the apple is a product of the United States.
3. The significance of the label "PUJA 94131 USA" is that it provides information about the origin of the apple. It suggests that the apple was grown, harvested, and packaged in the United States. However, it does not provide information about whether the apple is organic or not.
Then, according to the sub-questions and sub-answers to answer the question. No, the apple is not organic.

**2nd Round**

**Agent 1:** *I think with IO.*

Directly answer the question. Based on the image and the information provided, it is not possible to definitively determine if the apple is organic or not. The sticker on the apple indicates that it is a product of the United States, but it does not provide any information about whether it is organic or not. To determine if the apple is organic, one would need to look for a specific organic certification logo or text on the sticker, which is not visible in this image. Therefore, the answer to the question is I cannot tell if the apple is organic based on the image and the information provided.

**Agent 2:** *I think with CCoT.*

{"scene": {"objects": [{"type": "apple","attributes": {"color": "red","texture": "smooth","label": "PUJA 94131 USA"},"relationships": [{"type": "label","label": "PUJA 94131 USA"},{"type": "texture","texture": "smooth"},{"type": "color","color": "red"}]}],"relationships": [{"type": "label","label": "PUJA 94131 USA"},{"type": "texture","texture": "smooth"},{"type": "color","color": "red"}]}} Based on the image and the scene graph provided, we can see that the apple has a sticker on it with the text "PUJA 94131 USA." The sticker does not provide any information about the apple being organic. To determine if the apple is organic, one would need to look for a label or certification mark indicating that it meets organic standards. Since such a label is not visible in the image, it is not possible to definitively say whether the apple is organic or not based solely on the image.

**Agent 3:** *I think with DDCoT.*

Sub-questions:
1. What is the label on the apple?
2. What does the label indicate about the apple?
3. What is the significance of the label "PUJA 94131 USA"?
Sub-answers:
1. The label on the apple reads "PUJA 94131 USA."
2. The label indicates that the apple is a product of the United States.
3. The significance of the label "PUJA 94131 USA" is that it provides information about the origin of the apple. It suggests that the apple was grown, harvested, and packaged in the United States. However, it does not provide information about whether the apple is organic or not.
To determine if the apple is organic, one would typically look for a label or certification mark indicating that it meets organic standards. If such a label is not visible, it is not possible to definitively say whether the apple is organic or not based solely on the image.
Therefore, the answer to the original question is: No, the apple is not organic.

Figure 23: Failed case of DMAD.

# H PROMPTS

We display the prompts of all methods in this section, *i.e.*, CoT (Figure 24), SBP (Figure 25), PoT (Figure 26), IO (Figure 27), CCoT (Figure 28), DDCoT (Figure 29), Self-Refine (Figure 30), the contrast stage of Self-Contrast (Figure 31), the reflection stage of Self-Contrast (Figure 32), MRP for LLMs (Figure 33), MRP for MLLMs (Figure 34), MAD-persona (Figures 35 to 38), MAD (Figure 39) and DMAD (Figure 40). The part colored yellow represents prompts that need to be filled, and the part colored green represents the responses of the assistant.

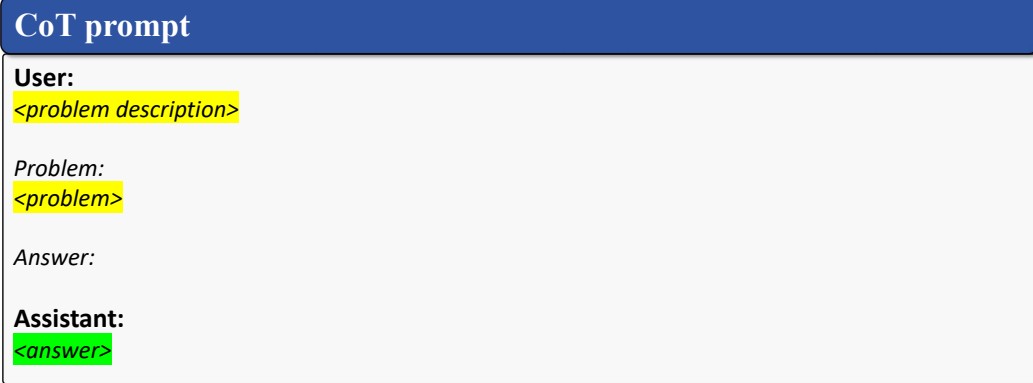

Figure 24: CoT prompt.

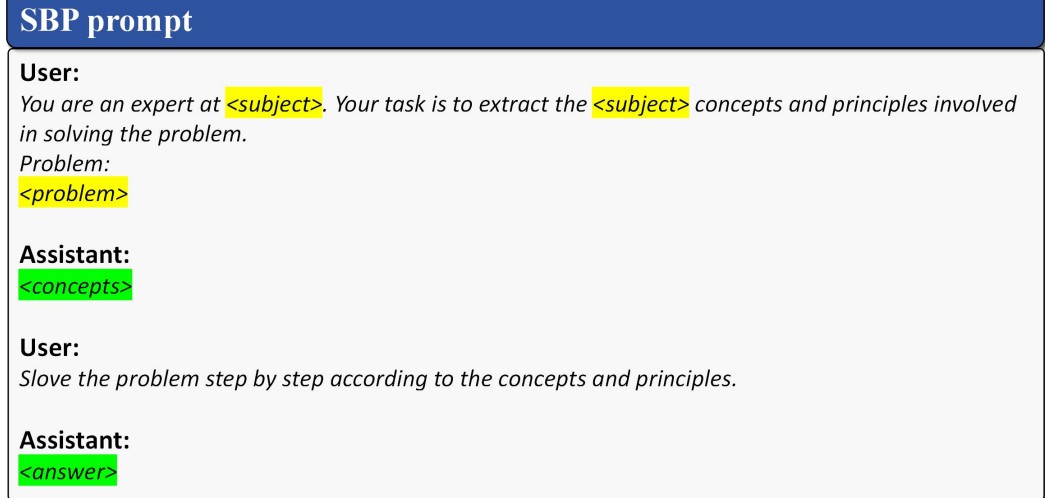

Figure 25: SBP prompt.

**PoT prompt**

**User:**
*Write Python Code to solve the following mathematics problem. Store your result as a variable named "ans". You can only output a Python program that can be executed by the Python function exec(). Do not say any other words.*
*Problem:*
*<problem>*

**Assistant:**
**

Figure 26: PoT prompt.

**IO prompt**

**User:**
*<image>*

*Question:*
*<question>*

**Assistant:**
*<answer>*

Figure 27: IO prompt.

**CCoT prompt**

**User:**
*<image>*

*Question:*
*<question>*

*For the provided image and its associated question, generate a scene graph in JSON format that includes the following:*
*1. Objects that are relevant to answering the question.*
*2. Object attributes that are relevant to answering the question.*
*3. Obect relationships that are relevant to answering the question.*

*Just generate the scene graph in JSON format. Do not say extra words.*

**Assistant:**
*<scene graph>*

**User:**
*Use the image and scene graph as context and answer the following question.*
*<question>*

**Assistant:**
*<answer>*

Figure 28: CCoT prompt.

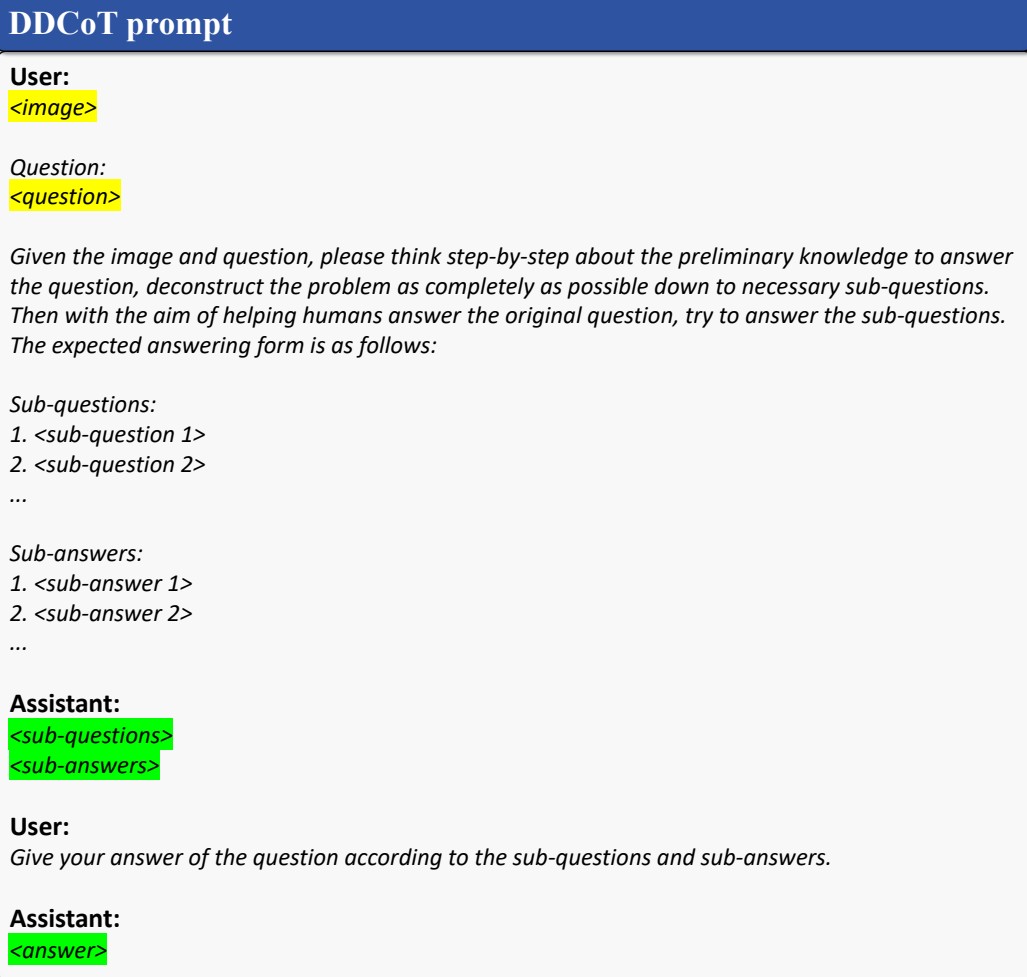

Figure 29: DDCoT prompt.

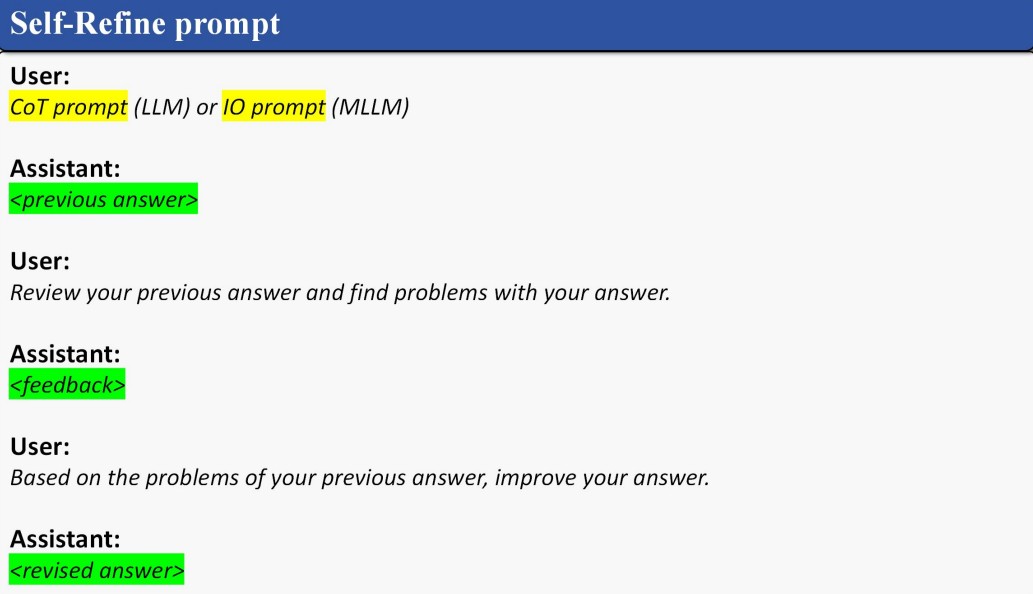

Figure 30: Self-Refine prompt.

---

**Self-Contrast prompt - Contrast**

**User:**
*You are a specialist who specializes in <subject>. Given some candidate solutions for a question, you should carefully compare the difference for each two solutions in their solving steps. When you compare, you need to consider the following questions:*
*1: Are the two solutions have different final answers and expressions?*
*2: Where are the differences in their solution steps and expressions?*
*3: Why are the answers of the two solutions different?*

*After contrasting, you should generate a checklist based on these differences between candidate solutions. You should carefully consider each discrepancy and the reasons behind it, summarizing them into a few checking instructions in the checklist. This checklist can guide others to re-examine the input question and these candidate solutions to eliminate these discrepancies.*

*Input Format:*
*The question is <question>*
*All solutions:*
*Solution1: {Solution1}*
*Solution2: {Solution2}*
*Solution3: {Solution3}*

*Output Format:*
*For Solution1 and Solution2: {Difference1}*
*For Solution1 and Solution3: {Difference2}*
*For Solution2 and Solution3: {Difference3}*
*Checklist: {Directive1, Directive2, ...}*

**Assistant:**
*<Differences>*
*<Checklist>*

Figure 31: Prompt of the contrast stage of Self-Contrast.

**Self-Contrast prompt - Reflection**

**User:**

*Given a question, multiple inconsistent solutions, their differences in their solving processes, and a checklist. You should revise the inconsistent solving step for each solution, eliminate the differences, and output a new solving process for each solution.*

*Guidance Rules for Reflection:*

*1. Please check carefully according to the requirements on the checklist. It helps you to resolve conflicts between different solutions.*

*2. When you finish revising inconsistent solutions, please ensure all revised solutions should have the same answer. If not, please revise again until all inconsistencies are removed, and all candidates are consistent.*

*3. Please output all revised solutions in JSON format as input, without any other text.*

*The question is {question}. The candidate solutions and their discrepancy are as follows:*

*{*
*"Candidate":*
*  {*
*    "result_1": {"solution": "{solution1}"},*
*    "result_2": {"solution": "{solution2}"},*
*    "result_3": {"solution": "{solution3}"}*
*  },*
*"Discrepancy":*
*  {*
*    "difference_1_2":*
*      {*
*        "source": "result_1",*
*        "target": "result_2",*
*        "relation": "{difference1_2}"*
*      },*
*    "difference_1_3":*
*      {*
*        "source": "result_1",*
*        "target": "result_3",*
*        "relation": "{difference1_3}"*
*      },*
*    "difference_2_3":*
*      {*
*        "source": "result_2",*
*        "target": "result_3",*
*        "relation": "{difference2_3}"*
*      },*
*  }*
*}*
*Checklist: {Checklist}*
*Please revise each inconsistent solution.*

**Assistant:**
*<answer>*

Figure 32: Prompt of the reflection stage of Self-Contrast.

**MRP prompt - LLM**

**User:**
*Instructions:*
*You are an adaptive reasoning method with meta-reasoning abilities, capable of selecting the most appropriate reasoning method based on the task provided by the user. Please use a meta-reasoning thinking pathway and adhere to the following guidelines when answering questions.*

*Reasoning method pool:*
*These reasoning methods are available for your tasks. Understand their applications for various tasks.*
*1. CoT:*
*Directly answer the question step by step.*
*2. Step_back:*
*First, extract the mathematics concepts and principles involved in solving the problem.*
*Then, solve the problem step by step according to the concepts and principles.*
*3. PoT:*
*Write Python Code to solve the mathematics problem. Store the result as a variable named "ans".*

*Analyzing giving task:*
*Evaluate the problem's difficulty carefully. Avoid underestimating the complexity and make a considered decision.*

*Anticipate the mistakes you might make:*
*Identify possible errors in method selection, such as underestimating the problem's difficulty.*

*Meta reasoning:*
*Apply meta-reasoning to choose the appropriate pathway.*

*Grading:*
*Rate the suitability of each solution on a scale of 1-3, selecting the method with the highest score as your preferred choice.*

*Choosing reasoning method:*
*Proceed methodically, taking a deep breath and thinking step-by-step. Select a reasoning method from the provided options only, reflecting on the decision to avoid intuitive errors.*

*Question:*
*<question>*

*The last sentence of your answer should be "So the best reasoning method is xxx", where "xxx" is the name of the best reasoning method.*

**Assistant:**
*<answer>*

Figure 33: MRP prompt for experiments on LLMs.

---

**MRP prompt - MLLM**

**User:**
*Instructions:*
*You are an adaptive reasoning method with meta-reasoning abilities, capable of selecting the most appropriate reasoning method based on the task provided by the user. Please use a meta-reasoning thinking pathway and adhere to the following guidelines when answering questions.*

*Reasoning method pool:*
*These reasoning methods are available for your tasks. Understand their applications for various tasks.*
*1. IO:*
*Directly answer the question.*
*2. CCoT:*
*First, get the scene graph of the image in JSON format.*
*Then, use the image and scene graph as context to answer the question.*
*3. DDCoT:*
*First, deconstruct the question down to sub-questions.*
*Then, accord to the sub-questions and sub-answers to answer the question.*

*Analyzing giving task:*
*Evaluate the problem's difficulty carefully. Avoid underestimating the complexity and make a considered decision.*

*Anticipate the mistakes you might make:*
*Identify possible errors in method selection, such as underestimating the problem's difficulty.*

*Meta reasoning:*
*Apply meta-reasoning to choose the appropriate pathway.*

*Grading:*
*Rate the suitability of each solution on a scale of 1-7, selecting the method with the highest score as your preferred choice.*

*Choosing reasoning method:*
*Proceed methodically, taking a deep breath and thinking step-by-step. Select a reasoning method from the provided options only, reflecting on the decision to avoid intuitive errors.*

*Question:*
*<question>*

*The last sentence of your answer should be "So the best reasoning method is xxx", where "xxx" is the name of the best reasoning method.*

**Assistant:**
*<answer>*

Figure 34: MRP prompt for experiments on MLLMs.

**MAD-persona prompt - Affirmative Side**

**System:**
*You are a debater. Hello and welcome to the debate. It's not necessary to fully agree with each other's perspectives, as our objective is to find the correct answer.*
*The debate topic is stated as follows:*
*<question>*

**User:**
*You think the correct answer is: <answer> Restate it and provide your reasons.*

**Assistant:**
*<answer>*

**User:**
*<negative answer>*

**Assistant:**
*<answer>*

**User:**
...

**Assistant:**
...

Figure 35: Prompt of the affirmative side of MAD-persona.

## MAD-persona prompt - Negative Side

**System:**
*You are a debater. Hello and welcome to the debate. It's not necessary to fully agree with each other's perspectives, as our objective is to find the correct answer.*
*The debate topic is stated as follows:*
*<question>*

**User:**
*<affirmative answer>*

*You disagree with my answer. Provide your answer and reasons.*

**Assistant:**
*<answer>*

**User:**
*<affirmative answer>*

*Do you agree with my perspective? Please provide your reasons and answer.*

**Assistant:**
*<answer>*

**User:**
*...*

**Assistant:**
*...*

Figure 36: Prompt of the negative side of MAD-persona.

## MAD-persona prompt - Discriminative Mode

**System:**
*You are a moderator. There will be two debaters involved in a debate. They will present their answers and discuss their perspectives on the following topic:*
*<question>*
*At the end of each round, you will evaluate answers and decide which is correct.*

**User:**
*Now the <number> round of debate for both sides has ended.*

*Affirmative side arguing:*
*<affirmative answer>*

*Negative side arguing:*
*<negative answer>*

*You, as the moderator, will evaluate both sides' answers and determine if there is a clear preference for an answer candidate. If so, please summarize your reasons for supporting affirmative/negative side and give the final answer that you think is correct, and the debate will conclude. If not, the debate will continue to the next round. Now please output your answer in json format, with the format as follows: {"Whether there is a preference": "Yes or No", "Supported Side": "Affirmative or Negative", "Reason": "", "debate_answer": ""}. "debate_answer" in the format should only be the right answer. Please strictly output in json format, do not output irrelevant content.*

**Assistant:**
*<answer>*

**User:**
...

**Assistant:**
...

Figure 37: Prompt of the discriminative mode of MAD-persona.

---

**MAD-persona prompt - Extractive Mode**

**System:**
*You are a moderator. There will be two debaters involved in a debate. They will present their answers and discuss their perspectives on the following topic:*
*<question>*
*At the end of each round, you will evaluate answers and decide which is correct.*

**User:**
*Affirmative side arguing:*
*<affirmative answer>*

*Negative side arguing:*
*<negative answer>*

*Now, what candidate answers do we have? Present them without reasons.*

**Assistant:**
*<answer>*

**User:**
*Therefore, what is the correct answer of the following question?*
*<question>*

*Please summarize your reasons and give the final answer that you think is correct. Now please output your answer in json format, with the format as follows: {"Reason": "", "debate_answer": ""}. "debate_answer" in the format should only be the right answer. Please strictly output in json format, do not output irrelevant content.*

**Assistant:**
*<answer>*

Figure 38: Prompt of the extractive mode of MAD-persona.

## MAD prompt

**User:**
*<prompt of one reasoning method>* (All agents think with the same method.)

**Assistant:**
*<solving process>*
*<answer>*

**User:**
*These are the answers to the question from other agents:*

*One agent answer:* ```{solving process 1, answer 1}```
*One agent answer:* ```{solving process 2, answer 2}```
*...*

*Using the answers from other agents as additional information, can you provide your answer to the question?*

*<question>*

**Assistant:**
*<answer>*

**User:**
*...*

**Assistant:**
*...*

Figure 39: MAD prompt.

**DMAD prompt**

**User:**
*<prompt of one reasoning method>* (All agents think with different methods.)

**Assistant:**
*<solving process>*
*<answer>*

**User:**
*These are other answers to the question using different reasoning methods:*

*One answer:* ```*{solving process 1, answer 1}*```
*One answer:* ```*{solving process 2, answer 2}*```
*...*

*Using the answers of different methods as additional information, can you provide your answer to the question?*

*<question>*

**Assistant:**
*<answer>*

**User:**
*...*

**Assistant:**
*...*

Figure 40: DMAD prompt.

