# OpenReview forum: "Breaking Mental Set to Improve Reasoning through Diverse Multi-Agent Debate"
_ICLR.cc/2025/Conference — ICLR 2025 Poster_

### Official Review · Reviewer_pspo · 2024-10-25

**Soundness:** 3
**Presentation:** 3
**Contribution:** 2
**Rating:** 6
**Confidence:** 3

**Summary:**

The paper presents a new method, Diverse Multi-Agent Debate (DMAD), for enhancing the reasoning capabilities of large language models (LLMs) by incorporating diverse approaches to problem-solving within a debate framework. The approach builds on the Multi-Agent Debate (MAD) method by requiring each agent—represented by an LLM—to employ a unique reasoning strategy in order to address a shared problem. Traditional techniques, such as self-reflection, are limited by the model's fixed patterns of reasoning, and MAD likewise suffers when agents adopt similar approaches due to inherent limitations in reasoning diversity. By (manually) instructing each agent to use a distinct reasoning approach, DMAD seeks to enhance solution accuracy across problem-solving scenarios. In each debate round, agents independently solve a problem and then iteratively refine their solutions using insights from other agents' approaches. DMAD is tested on multiple models, including GPT-4o-mini, LLaMA-3-70B, LLaVA-1.6-13B, Gemini-1.5-Flash, and GPT-4o-2024-05-13, demonstrating overall positive, though incremental, performance gains across benchmarks.

**Strengths:**

The paper contributes meaningfully to the ongoing exploration of multi-agent debate in LLMs by proposing a straight forward extension of the MAD approach that encourages diverse reasoning within the debate process. DMAD's design is simple and builds logically upon MAD, representing an incremental advance. The experiments are comprehensive and cover multiple relevant benchmarks, which gives a well-rounded view of DMAD’s performance. The inclusion of evaluations across various LLMs, both uni- and multimodal, is particularly nice, as it demonstrates the method's effectiveness across different architectures and contexts. The results show that reasoning techniques enforced by DMAD enhance performance on most tasks, confirming that diversity in reasoning approaches can be advantageous even if the cumulative improvement is moderate. Overall, the paper is clearly written and presents a straightforward extension to the multi-agent debate approaches in LLMs, supported by compelling experimental results.

**Weaknesses:**

While DMAD is a nice addition to the multi-agent debate line of work, its novelty is limited. The method represents an incremental improvement rather than a fundamentally new concept. The debate setup in DMAD may be perceived as a minor extension of the MAD framework, as it mainly focuses on modifying agents’ reasoning patterns. I would have liked to see additional investigations that more deeply alter the default setup of the debate, even if it was as “simple” as running an ablation study with different number of agents. Additionally, while DMAD shows an overall improvement, some gains are marginal or even negative, raising questions about the method’s broader applicability. Though the authors attempt to reason about cases where DMAD's performance lags behind other approaches, the study could benefit from a more thorough investigation of typical mistakes made during debating with DMAD.

**Questions:**

## General Questions
1. In Table 1, for the Alg. column and the LLaMA model, the value 72 achieved by DMAD is not the highest (c.f. MAD (All CoT) and CoT - SC); is this a typo or a correct results (in the latter case, please correct the text bolding)?
2. Do you believe there would be merit in attempting to set up another LLM to propose reasoning strategies, instead of having them fixed? I understand this is a feat warranting a paper on its own, though I'm quite curious about such applications, as it would allow one to close the loop — having reasoning strategies being proposed and directly “evaluated” by the agents;
3. Have you examined setups with more than 3 agents? Building on my previous question, I assume it's not as straight forward to find additional reasoning strategies that would be assigned to new debaters? Do you believe, given more reasoning strategies, that the performance of the method would scale up with the number of agents, or would it plateau, similar to what happens with the number of rounds?
4. I got confused when reading Eq. 3 and Eq. 4; namely, history $\mathbb{H}$ is defined as a set of tuples, one per agent, each of which containing information pertaining to a particular agent (question, solving process, answer). Perhaps I am missing something obvious, but in Eq. 4, wouldn't it be sufficient to simply define $h_i \equiv \mathbb{H}$ because $\mathbb{H}$ by definition contains the information tuple for the agent $i$, but also for all other agents? To quote the paper “*$h_i$ represents the history of messages for $M_i$, extended with messages from other agents*”, which to me looks exactly like $\mathbb{H}$?
5. Regarding the MATH benchmark, could you provide some intuition on why the performance degrades after 3 rounds (for GPT-4o-mini) compared to 2 rounds? Additionally, why does the LLaMA model's performance improve linearly with more rounds using the MAD method, but drops after the third round with DMAD? Do you have any insights into what kinds of mistakes are being made with additional rounds when using DMAD?

## Minor Comments
- Line 157: Did you mean to write $h_i$ instead of $A_{i,1}$? The variable $A_{i,1}$ is not referenced in Eq. 1;
- Line 314: Shouldn't this be "DMAD" instead of "MAD"?
- Appendix Figures (e.g., 9, 10, 11, etc.): The figures are difficult to interpret due to the formatting characters. You may want to include pre-formatted versions for clarity and consider adding a footnote explaining that the agents actually receive raw markdown text, if necessary.

---

> ### Author Response · Authors · 2024-11-19
> **Summary Response**
>
> Sincerely appreciate for your meticulous review of our paper! We thank for your valuable insights, which could be much helpful for improving and clarifying our paper. We would like to address your each concern in detail.
>
> We summarize our responses as follows:
>
> 1. We provide a specific and systematic definition for **mental set**, connected with the corresponding psychological phenomenon. (In Response 1/5)
> 2. We emphasize our novelty and differences compared with other methods leveraging diverse solutions. (In Response 1/5)
> 3. We discuss about typical mistakes of DMAD and MAD. (In Response 1/5)
> 4. We conduct abundant experiments to prove the broader applicability of our method. (In Response 2/5)
>
>    (1) Statistical significance. We re-conduct our experiments on MATH with GPT-4o-mini and LLaMA-3-70B-Instruct 3 times to get the average accuracy and standard deviation. Our method DMAD consistently outperforms other MAD methods.
>
>    (2) We test on a smaller model, LLaMA-3-8B-Instruct. DMAD also achieves the best performance.
>
>    (3) We conduct experiments on the subset of a more challenging benchmark, MMLU [1], where DMAD also outperforms other methods.
> 5. We elaborate our opinions on automatically generating prompting strategies (In Response 3/5) and setting more agents (In Response 4/5).
> 6. We answer the questions about equations. (In Response 4/5)
> 7. We explain the degraded performance of LLaMA in the 3rd round. (In Response 5/5)
> 8. We design 4 different stop criteria to determine when the debate ends. (In Response 5/5)
>
> We also respond to other minor questions, and have added these supplementary experiments in the revised paper. We hope our responses can address your concerns. Looking forward to further communication with you.
>
> [1] Measuring Massive Multitask Language Understanding. ICLR 2021.

---

> ### Author Response · Authors · 2024-11-19
> **Response 1/5**
>
> **What is mental set?**
>
> In our paper, we introduce a new concept of **mental set** according to the psychological theory. Here we provide a specific definition for it. Denote MAD (All CoT), MAD (All SBP), and MAD (All PoT) as $M_{1}, M_{2}$ and $M_{3}$ respectively. When using a kind of MAD method $M_{i}$ to solve a problem, if all agents consistently get wrong answers in all debate rounds, we assume that $M_{i}$ is unable to correctly solve the problem. Record all such problems for $M_{i}$ as the set $P_{i}$, and get $P=P_{1} \cap P_{2} \cap P_{3}$. For a problem $p \in P_{i}$, if it satisfies $p \notin P$, we define that the problem $p$ causes **mental set** of $M_{i}$, and define $p$ as the **mental set problem** of $M_{i}$. It means although $M_{i}$ constantly gets wrong solutions, the model can correctly solve the problem by changing to another method.
>
> We record the mental set problems of each method on MATH with GPT-4o-mini, and the problems among them which at least one agent of other methods correctly solves. All methods have 3 agents and debate for 2 rounds. **The results show that DMAD can more effectively solve other methods' mental set problems.**
>
> |         | MAD (All CoT) | MAD (All SBP) | MAD (All PoT) |
> |----------|---------|---------|------|
> | Number of mental set problems |    70    |   87    |   67    |
> ||
> | Problems that MAD (All CoT) correctly solves |  0    |   45 (51.72%)   |   46 (68.7%)   |
> | Problems that MAD (All SBP) correctly solves |   28 (40.0%)    |   0    |   31 (46.3%)   |
> | Problems that MAD (All PoT) correctly solves |   **49 (70.0%)**   |   51 (58.62%)    |   0    |
> | Problems that DMAD correctly solves |   48 (68.6%)   |   **60 (69.0%)**   |  **49 (73.1%)**    |
>
>
> - **Weakness 1: Limited novelty.**
>
> We would like to emphasize our novelty and differences compared with other methods leveraging diverse solutions.
>
> 1.   We introduce a new concept of mental set and provide a specific definition and systematic analysis as above, which is inspired by the psychological theory of mental set.
> 2.   Self-Contrast just contrasts diverse reasoning solutions to get better performance than simple IO solutions. However, they don't compare their method with each distinct reasoning strategy. So it's not sure if using diverse solving perspectives all outperforms completely using one perspective among them. Meta-Reasoning Prompting dynamically chooses the best reasoning method to solve problems, while it only achieves a balanced performance between various reasoning methods and is often inferior to a specific one in most cases. Different from them, we conduct a more comprehensive analysis and prove that leveraging diverse reasoning methods can outperform all single methods with the same setting.
> 3. Our experiment shows that DMAD has the potential to break the mental set of single reasoning methods. For example, for the same problem, all agents in MAD (All IO) constantly get wrong answers in all 5 rounds. While in DMAD, an agent thinking with IO correctly solves the problem after receiving messages from agents thinking from other perspectives in the 2nd round. This also holds for MAD (All CoT), MAD (All SBP), MAD (All PoT), MAD (All CCoT) and MAD (All DDCoT). We state this in Section 4.3.4 in the initial submission.
>
> - **Weakness 2:  Investigation of typical mistakes made during debating with DMAD.**
>
> Apart from the failed case we discuss in Appendix, another typical mistake is that sometimes the agents may be affected by other agents' wrong solutions, changing its original correct answer to a wrong one. This is not the special case of DMAD, but a common issue of MAD, attracting many scholars to try to study how to alleviate this problem. However, this is not the core of our paper. We just introduce DMAD to help solve problems that MAD struggles with, improving the upper bound of its reasoning capability.

---

> ### Author Response · Authors · 2024-11-19
> **Response 2/5**
>
> - **Weakness 3: Broader applicability.**
>
> 1. Statistical significance.
>
> We run experiments with GPT-4o-mini and LLaMA-3-70B-Instruct on MATH 3 times to calculate the average accuracy and standard deviation. **Statistical experiments demonstrate that DMAD outperforms other MAD settings. DMAD on LLaMA-3-70B-Instruct also gets better average accuracy than MAD (All PoT).**
>
> | Models | Methods       | Alg. | C&P  | Geom. | Int. Alg. | Num. Th. | PreAlg. | PreCalc. | Avg.  |
> |---------|---------------|-------|------|--------|------------|-----------|----------|-----------|-------|
> | GPT-4o-mini     | MAD (All CoT) | 91.3±2.16 | 78.7±0.82 | 55.3±2.16 | **55.0±2.45**  | **82.7±1.63** | **86.3±0.82** | 39.7±0.82 | 69.9±0.93 |
> |   GPT-4o-mini      | MAD (All SBP)| 88.3±0.82  | 77.7±1.63  | 49.3±2.16  | 44.0±3.74  | 81.3±2.94 | 83.7±0.82 | 38.7±0.82 | 66.1±0.35 |
> |   GPT-4o-mini      | MAD (All PoT)| 91.3±1.63  | 75.7±2.16 | 49.0±3.74  | 52.7±5.72  | 80.7±0.82 | 85.3±0.82 | 39.7±0.82 | 67.8±2.24|
> |    GPT-4o-mini     | DMAD          | **91.7±1.63**  | **81.0±1.41**  | **57.3±2.16**  | 53.7±0.82  | **82.7±0.82** | **86.3±0.82** | **40.0±1.41** | **70.4±0.95**|
> | LLaMA-3-70B-Instruct   | MAD (All CoT) | **72.7±2.94**  | 48.0±3.74  | 31.3±0.82  | 24.3±0.82  | 40.7±2.94 | 69.7±2.94 | 31.0±1.41 | 45.6±0.23|
> | LLaMA-3-70B-Instruct      | MAD (All SBP)| 69.3±4.32  | 51.0±2.83  | 29.3±4.32  | 25.0±0.00  | 42.0±3.74 | 70.0±1.41 | 27.0±1.41 | 44.8±1.52|
> |   LLaMA-3-70B-Instruct       | MAD (All PoT)| 66.7±5.89  | **52.0±2.83**  | 31.0±2.45  | 27.7±2.16  | **49.0±4.90** | 67.0±7.35 | **32.7±1.63** | 46.6±0.35|
> |   LLaMA-3-70B-Instruct       | DMAD          | **72.7±1.63**  | 49.0±4.24  | **32.7±1.63**  | **29.3±2.94**  | 44.3±4.08  | **72.3±2.16** | 27.3±3.56 | **46.8±1.43** |
>
> 2. On smaller models.
>
> We test {MAD (All CoT), MAD (All SBP), MAD (PoT), DMAD} on LLaMA-3-8B-Instruct on MATH. **Results demonstrate DMAD is also effective on smaller models.**
>
> | Models          | Methods            | Alg. | C&P | Geom. | Int. Alg. | Num. Th. | PreAlg. | PreCalc. | Avg. |
> |-----------------|--------------------|---------|--------------------------|-----------|-----------------------|---------------|------------|-------------|---------|
> | LLaMA-3-8B-Instruct | MAD (All CoT)     | **49.7±0.82** | 19.3±1.63            | 18.3±2.16 | 15.3±0.82           | 18.3±0.82    | **46.7±1.63**   | 15.0±1.41 | 26.1±0.42 |
> |  LLaMA-3-8B-Instruct               | MAD (All SBP)     | 46.3±2.94 | 20.7±4.97            | 17.3±1.63 | 15.3±0.82           | 20.0±6.16   | 44.3±5.35   | 11.3±0.82 | 25.3±1.94 |
> |  LLaMA-3-8B-Instruct               | MAD (All PoT)     | 41.3±5.72 | 20.0±2.83            | 16.0±1.41 | 13.7±4.55           | 21.0±5.66   | 39.7±4.97   | **19.0±4.24** | 24.4±1.62 |
> | LLaMA-3-8B-Instruct                | DMAD               | 47.7±0.82 | **21.3±0.82**            | **21.7±0.82** | **16.0±0.00**           | **21.3±5.72**   | 45.0±1.41   | 15.0±1.41 | **26.9±0.40** |
>
> 3. On more challenging reasoning dataset of MMLU [1].
>
> We supplement experiments on a more challenging benchmark, MMLU [1]. We test GPT-4o-mini on the subset "abstract_algebra" of MMLU and run 3 times. As this dataset consists of multi-choice questions and some options are not numbers, we replace PoT with Least-to-Most (L2M) [2]. **DMAD also outperforms other MAD settings on this challenging multi-hop reasoning task.**
>
> |         | MAD (All CoT) | MAD (All SBP) | MAD (All L2M) | DMAD  |
> |---------|---------|---------|---------|-------|
> | Accuracy| 72.3±0.82 | 79.0±1.41 | 74.3±0.82 | **79.7±1.63** |
>
> [1] Measuring Massive Multitask Language Understanding. ICLR 2021.
>
> [2] Least-to-Most Prompting Enables Complex Reasoning in Large Language Models. ICLR 2023.

---

> ### Author Response · Authors · 2024-11-19
> **Response 3/5**
>
> - **Q1: "In Table 1, for the Alg. column and the LLaMA model, the value 72 achieved by DMAD is not the highest (c.f. MAD (All CoT) and CoT - SC); is this a typo or a correct results (in the latter case, please correct the text bolding)?"**
>
> Thanks for your feedback! We have corrected the text bolding.
>
> - **Q2: "Do you believe there would be merit in attempting to set up another LLM to propose reasoning strategies, instead of having them fixed?"**
>
> This is an interesting idea. However, based on our experiments and experience, we're concerned that this approach may not lead to a better performance, especially on smaller and weaker models. The only advantage may be its automation. The reasons come from two aspects:
>
> 1. **Quality**: The quality of the automatically generated prompt is not guaranteed. Instead, we select published prompting strategies that have been widely used and their effectiveness has been widely proven.
>
> 2. **Diversity**: The diversity of the automatically generated prompt is not guaranteed. Instead, we can subjectively choose diverse strategies or use a designed metric to select.
>
> Assuming we have $K$ candidate basic reasoning strategies $\\{R_i\\}^K_{i=1}$ and want to select $k$ diverse ones, run each strategy $N$ times, and record the problems which $R_{i}$ correctly solve at least once as $P_{i}$ (Note that the definition of $P_{i}$ here is different from $P_{i}$ when introducing mental set). Note all problems on the measured dataset as $P_{all}$. We can define the diversity of the selected strategies $\lbrace R_{s_i}\rbrace ^{k}_{i=1}$ as
>
> $$diversity =\frac{\|\cup \\{{P_{s_i}\\}}^{k}_{i=1}\|}{\|P\_{all}\|}\in [0, 1],  s_i \in \\{1, 2, ... \,K\\}, s_i \neq s_j \, for \\, \\, i \neq j.$$
>
> This represents the proportion of total questions that the selected $k$ methods can answer correctly. **The more diverse these methods are, the larger this proportion should be.**
>
> We run $N=3$ times for each reasoning strategy in $\\{{R_i}  \\}^{4}_{i=1}=\\{CoT, L2M, SBP, PoT\\}$ and select $k=3$ strategies to calculate $diversity$, and test DMAD with different strategy groups on MATH with GPT-4o-mini. We can see $diversity$ of {CoT, SBP, PoT} is the highest. **Experiment results show that using the strategy group with larger $diversity$ can get better results.**
>
> |Model| ${R_{s}}_{1}$  | ${R_{s}}_{2}$  | ${R_{s}}_{3}$  |$diversity$ |
> |---|---|---|---|-------|
> |GPT-4o-mini| CoT | L2M | SBP |  0.8471|
> |GPT-4o-mini| CoT | L2M | PoT |  0.8643|
> |GPT-4o-mini| **CoT** | **SBP** | **PoT** |  **0.8657** |
> |GPT-4o-mini| L2M | SBP | PoT |  0.8557|
> ||
> |LLaMA-3-70B-Instruct| CoT | L2M | SBP | 0.6743|
> |LLaMA-3-70B-Instruct| CoT | L2M | PoT | 0.6600|
> |LLaMA-3-70B-Instruct| **CoT** | **SBP** | **PoT** | **0.7286**|
> |LLaMA-3-70B-Instruct| L2M | SBP | PoT | 0.6386|
>
> |  Method  | Alg. | C&P | Geom. | Int. Alg. | Num. Th. | PreAlg. | PreCalc. | Average |
> |--|---------|---------------------------|----------|-----------------------|---------------|------------|-------------|---------|
> | DMAD (CoT, L2M, SBP)     | 88.7±0.82 | 78.0±1.41                 | 54.7±4.55 | 49.0±5.10             | **82.7±0.82**      | 85.7±0.82   | 37.7±3.27   | 68.0±1.83 |
> | DMAD (CoT, L2M, PoT)     | **91.7±1.63** | **81.3±2.94**                 | 54.0±1.41 | **54.7±0.82**             | 82.3±0.82      | **87.7±2.16**   | 39.0±2.83   | 70.1±0.65 |
> | DMAD (CoT, SBP, PoT)     | **91.7±1.63** | 81.0±1.41                   | **57.3±2.16** | 53.7±0.82             | **82.7±0.82**      | 86.3±0.82   | **40.0±1.41**   | **70.4±0.95**|
> | DMAD (L2M, SBP, PoT)|	87.0±1.41 |	**81.3±5.72** |	54.7±3.27 |	51.3±1.63	| 80.7±4.55	| 85.0±1.41| 	38.0±3.74| 68.3±2.13|

---

> ### Author Response · Authors · 2024-11-19
> **Response 4/5**
>
> - **Q3: Examining setups with more than 3 agents, with automatically generated diverse methods and manually selected ones.**
>
> According to our answer to Q2, it may be difficult for the model to simply propose sufficiently high-quality and diverse strategies. So it is hard to guarantee the performance of DMAD with automatically generated strategies. Besides, manually selecting more high-quality strategies does not necessarily mean much higher diversity. Finding more sufficiently diverse reasoning methods is challenging. We calculate $diversity$ for $k=4$ strategies. The improvement is limited compared to $k=3$. For LLaMA, $diversity$ of $k=4$ is even equal to $k=3$ with $\\{CoT, SBP, PoT\\}$.
>
> |Model| ${R_{s}}_{1}$  | ${R_{s}}_{2}$  | ${R_{s}}_{3}$  | ${R_{s}}_{4}$| $diversity$ |
> |---|---|---|---|---|----|
> |GPT-4o-mini| CoT | L2M | SBP | - | 0.8471|
> |GPT-4o-mini| CoT | L2M | PoT | - | 0.8643|
> |GPT-4o-mini| CoT | SBP | PoT | - | 0.8657|
> |GPT-4o-mini| L2M | SBP | PoT | - | 0.8557|
> |GPT-4o-mini| CoT | L2M | SBP | PoT | 0.8743|
> |
> |LLaMA-3-70B-Instruct| CoT | L2M | SBP | - | 0.6743|
> |LLaMA-3-70B-Instruct| CoT | L2M | PoT | - | 0.6600|
> |LLaMA-3-70B-Instruct| CoT | SBP | PoT | - | 0.7286|
> |LLaMA-3-70B-Instruct| L2M | SBP | PoT | - | 0.6386|
> |LLaMA-3-70B-Instruct| CoT | L2M | SBP | PoT| 0.7286|
>
> What's more, assuming we do have sufficiently diverse high-quality strategies, we think the performance may first increase, then stabilize, or even decrease at last when increasing agents. This is because, when there are too many agents, the strategies are so diverse that it leads to confusion. It may be more difficult for different agents to reach consensus, and more easily to make typical mistakes discussed in the answer to Weakness 2.
>
> Nonetheless, we cannot rule out the possibility that this automatic setup is feasible and performs better, as we have not conducted systematic and comprehensive experiments to verify it. We greatly appreciate your insightful suggestion. It is worth leaving for future work exploration.
>
> - **Q4: Confusion about Eq. 3 and Eq. 4.**
>
> The messages stored in history $h$ are in order. The message of the agent itself is appended into $h$ at first, and others afterwards. This is consistent with the original prompt of MAD for a fair comparison, which places the agent's own solution in front while others at the back. We will emphasize this in the revised version to help readers have a better understanding.

---

> ### Author Response · Authors · 2024-11-19
> **Response 5/5**
>
> - **Q5: Explanation about degraded performance and mistakes made with additional rounds when using DMAD. Why does LLaMA model's performance drop after the third round with DMAD?**
>
> This is mainly because of the typical mistakes discussed in the answer to Weakness 2. For example, in the 2nd round, Agent 1 and Agent 2 get the right answer while Agent 3 gets a wrong one. Then the judge with Self-Consistency gets the right answer. However, in the 3rd round, Agent 1 and Agent 2 mistakenly obtain the wrong conclusion, while Agent 3 correctly modifies its wrong answer. Then the judge with Self-Consistency gets the wrong answer. (Situation 1)
>
> What we hope is, Agent 3 correctly revises its answer, and Agent 1 and Agent 2 retain their correct answers. Another ideal situation is, if two agents get wrong answers while the other one is right, all agents or at least two agents reach correct solutions in the next round. (Situation 2)
>
> DMAD will more frequently experience that agents get different answers due to its diversity compared to MAD. If there are more Situation 2 than Situation 1, the performance will upgrade, otherwise, it degrades, which is why llama's performance drops after the 3rd round.
>
>
> **When to stop the debate?**
>
> So it is important to determine when the debate ends. **Our method DMAD can be over in just 2 rounds and gets good performance, even surpassing MAD in 5 rounds.** Besides, we can also design different criteria to determine when the debate ends, such as:
>
> 1. Consistency-2: If there exist 2 agents getting the same answer, the debate ends. Otherwise, the debate continues unless reaching its maximum number of rounds.
>
> 2. Consistency-3: If all the 3 agents get the same answer, the debate ends. Otherwise, the debate continues unless reaching its maximum number of rounds.
>
> 3. Self-Determine: Set another model as a judge to determine whether the debate should be over. The judge can receive all agents' solutions in each round and save them in its history.
>
> 4. Hybrid: In each round, if all agents get the same answer, the debate ends. Otherwise, use Self-Determine to judge whether the debate should end or continue.
>
> We set the maximum number of the debate round to 5 with Gemini on ScienceQA. With the stop criteria, MAD and DMAD can get a relatively high accuracy with a low average round. Results show that the criteria of Consistency-3 is the best. However, DMAD with the stop criteria of Consistency-3 needs more overhead with nearly 5 debate rounds.
>
> | Stop Criteria   | Method | Average Round | Accuracy | Method | Average Round | Accuracy |
> |-----------------|---------|---------------|-----------|---------|---------------|-----------|
> | Consistency-2    | MAD     | 1.0977        | 85.09     | DMAD    | 1.0788        | 84.93     |
> | Consistency-3    | MAD     | 1.1076        | 85.14     | DMAD    | 4.9861        | 85.32     |
> | Self-Determine   | MAD     | 1.1091        | 84.89     | DMAD    | 1.6063        | 85.03     |
> | Hybrid          | MAD     | 1.0605        | 84.84     | DMAD    | 1.6063        | 85.03     |
> | Fixed Round      | MAD     | 1            | 84.24     | DMAD    | 1            | 84.58     |
> | Fixed Round      | MAD     | 2            | 84.84     | DMAD    | 2            | 85.57     |
> | Fixed Round      | MAD     | 3            | 85.34     | DMAD    | 3            | **85.62** |
> | Fixed Round      | MAD     | 4            | 85.34     | DMAD    | 4            | 85.52     |
> | Fixed Round      | MAD     | 5            | 85.44     | DMAD    | 5            | 85.52     |
>
>
> - **Minor Comments**
>
> Thanks for your careful review of our paper. We have revised the minor errors you mentioned in the first two items. We have also added an explanation that the agents actually receive raw markdown text.
>
> Thanks again for your valuable feedback. We hope our responses can address your concerns. Looking forward to further communication with you.

---

> ### Comment · Reviewer_pspo · 2024-11-23
> **Thank you for your response**
>
> I thank the authors for their comprehensive response. After consideration, I decided to keep my initial assessment — the work, while incremental, would in my opinion represent an interesting and valuable addition to the community.

---

> > ### Author Response · Authors · 2024-11-23
> > **Thanks for your reply for Reviewer pspo**
> >
> > Thanks for your reply! We deeply appreciate your insightful feedback and will continue to make progress.

---

### Official Review · Reviewer_QNMW · 2024-11-03

**Soundness:** 3
**Presentation:** 2
**Contribution:** 1
**Rating:** 6
**Confidence:** 4

**Summary:**

The pair introduces Diverse Multi-Agent Debate (DMAD), a new method that encourages AI agents to employ different reasoning approaches when debating to solve problems. Unlike traditional MAD, in which agents use the same reasoning method, DMAD assigns distinct reasoning strategies to each agent. DMAD consistently outperformed other methods across multiple benchmarks, demonstrating that diverse reasoning approaches can help break mental sets - the tendency to approach problems in fixed ways based on past experience. This work provides a promising direction for enhancing AI reasoning capabilities without requiring model retraining.

**Strengths:**

1: The solution is simple since it doesn't require model retraining or additional data collection, making it immediately applicable.
2: The research provides extensive experimental validation across multiple dimensions and shows consistent improvements.

**Weaknesses:**

1:Resource Efficiency Analysis

The paper lacks a critical discussion on computational efficiency and practical costs. A detailed analysis comparing token usage, API costs, and computational overhead between DMAD and simpler reasoning strategies would better demonstrate its practical value. This is particularly important for real-world applications where resource constraints must be balanced against performance gains.

2: Weak Theoretical Foundation and Empirical Evidence

While the paper draws motivation from psychology's "mental set" phenomenon, the connection appears tenuous. Traditional MAD with a single reasoning method (e.g., CoT) achieves comparable performance to DMAD (as shown in Table 1), suggesting that method diversity might not be the key driver of improvement (Or statistical significance should be provided). This weakens the paper's central thesis that diverse reasoning methods are necessary to break mental set, as even fixed methods can yield different problem-solving approaches through debate. This raises questions about whether the gains are truly from method diversity or simply from the debate process itself (which also encourages some sort of "divergent thinking").

3: Model Capability Scaling

The paper's focus on large, capable models leaves an important question unexplored: could smaller models using DMAD potentially outperform larger models using simpler reasoning methods? This comparative analysis would better demonstrate the method's value proposition, especially for scenarios where using large models isn't feasible. Understanding how DMAD's effectiveness scales with model size could provide valuable insights for practical applications.

4:Limited Implementation Guidelines

Despite DMAD's potential impact on engineering workflows, the paper lacks concrete guidance on implementation decisions. Critical questions remain unanswered; See Questions blow for more.

5: Incremental Rather Than Novel Contribution:

While the paper effectively implements diverse reasoning in a MAD framework, the core concept builds on existing ideas. Similar approaches of leveraging diverse perspectives for improved reasoning have been explored in recent work (Zhang et al., 2024). Although DMAD shows impressive results, its primary contribution appears to be an incremental improvement (applying diversity to MAD) rather than introducing a fundamentally new concept to the field.

Reference:
Zhang, W., Shen, Y., Wu, L., Peng, Q., Wang, J., Zhuang, Y., & Lu, W. (2024). Self-Contrast: Better Reflection Through Inconsistent Solving Perspectives. 2024 ACL.

**Questions:**

1: Is there a reason you chose the 3 reasoning strategies (CoT, SBP, PoT)? How about others like least-to-most prompting (Zhou et al., 2023)?

2: Could the system dynamically adjust its reasoning methods during the debate process? For instance, could agents switch reasoning strategies based on the progress of the debate or the nature of disagreements that emerge?

3: Why do you focus on mathematical reasoning and visual understanding? How about other tasks?

4: What criteria determine the ideal number of debate rounds?  Examining Figure 4 (Performance with increased rounds), there's an interesting pattern where DMAD sometimes achieves better performance in 2 rounds than MAD does in 5 rounds, however would a smarter stopping criteria can help mitigate the problem?


Reference:

Zhou, Denny, Nathanael Schärli, Le Hou, Jason Wei, Nathan Scales, Xuezhi Wang, Dale Schuurmans, Claire Cui, Ed Chi, Yun-Hsuan Sung, Daphne Ippolito, and Claire Cassidy. 2023. "Least-to-most Prompting Enables Complex Reasoning in Large Language Models." In International Conference on Learning Representations (ICLR).

---

> ### Author Response · Authors · 2024-11-19
> **Summary Response**
>
> We appreciate your comprehensive review and valuable feedback of our paper. We would like to address each of your questions.
>
>
> We summarize our responses as follows:
>
> 1. We elaborate the connection between our idea and the psychological theory of mental set, and provide a specific and systematic definition for it, proven by our experiment results. (In Response 1/4)
> 2. We re-conduct our experiments on MATH with GPT-4o-mini and LLaMA-3-70B-Instruct 3 times to get the average accuracy and standard deviation. Our method DMAD consistently outperforms other MAD methods. (In Response 1/4)
> 3. We report the tokens overhead and cost of all methods on MATH. DMAD achieves optimal performance with balanced overhead compared with other MAD methods. (In Response 2/4)
> 4. We explain why choosing CoT, SBP, and PoT, and also test other groups including Least-to-Most prompting. We design an objectively quantitative metric to measure the diversity of selected basic prompting strategies, and conduct several experiments to prove the effectiveness of this metric and the gains do come from diversity. (In Response 2/4)
> 5. We test on a smaller model, LLaMA-3-8B-Instruct. DMAD also achieves the best performance. (In Response 3/4)
> 6. We emphasize our novelty and differences compared with other methods leveraging diverse solutions. (In Response 3/4)
> 7. We conduct experiments on the subset of a more challenging benchmark, MMLU [1], where DMAD also outperforms other methods. (In Response 4/4)
> 8. We design 4 different stop criteria to determine when the debate ends to reduce excessive tokens. (In Response 4/4)
>
> We also respond to other minor questions, and have added these supplements in the revised paper. We hope our responses can address your concerns. Looking forward to further communication with you.
>
> [1] Measuring Massive Multitask Language Understanding. ICLR 2021.

---

> > ### Comment · Reviewer_QNMW · 2024-11-24
> > **response**
> >
> > I appreciate the authors for their comprehensive response, especially in addressing my concerns about some of the design choices and the lack of more experiments. After consideration, I decided to raise my score to 6 and keep my rating neural. I still have concerns about the novelty of the work. The psychological perspective, while interesting, would, in my point of view, still not help enough to distinguish it well from other comparable works especially given ICLR standard. I do commend the soundness of the current experiments and writing and will leave ACs judgment to make the final decision.

---

> > > ### Author Response · Authors · 2024-11-25
> > > **Thanks for your response for Reviewer QNMW**
> > >
> > > We are glad that our responses have addressed your concerns, and we are grateful for your commendation of the soundness of our experiments and writing. We sincerely appreciate and value your valuable opinions, which will help us to push the frontier in our future work.

---

> ### Author Response · Authors · 2024-11-19
> **Response 1/4**
>
> - **Weakness 2-1: Weak theoretical foundation and empirical evidence.**
>
> Our idea is inspired by the psychological theory of mental set, which refers to the cognitive tendency to approach problems in a particular way based on past experiences, learned behaviors, or established habits. This may hinder diverse thinking and make it difficult to solve a problem in the mental set way. However, if thinking in a different way, it may surprisingly find it is easy to address the problem. **We observe that LLMs have an analogous phenomenon. MAD with a fixed prompting strategy may always get wrong answers to a problem, while changing to another prompting strategy can correctly solve the problem.**
>
> Here we provide a specific definition for it. Denote MAD (All CoT), MAD (All SBP), and MAD (All PoT) as $M_{1}, M_{2}$ and $M_{3}$ respectively. When using a kind of MAD method $M_{i}$ to solve a problem, if all agents consistently get wrong answers in all debate rounds, we assume that $M_{i}$ is unable to correctly solve the problem. Record all such problems for $M_{i}$ as the set $P_{i}$, and get $P=P_{1} \cap P_{2} \cap P_{3}$. For a problem $p \in P_{i}$, if it satisfies $p \notin P$, we define that the problem $p$ causes **mental set** of $M_{i}$, and define $p$ as the **mental set problem** of $M_{i}$. It means although $M_{i}$ constantly gets wrong solutions to $p$, the model can correctly solve the problem by changing to another strategy.
>
> We record the mental set problems of each method on MATH with GPT-4o-mini, and the problems among them which at least one agent of other methods correctly solves. All methods have 3 agents and debate for 2 rounds. **The results prove the connection between LLM reasoning and the mental set phenomenon in psychology, and DMAD can more effectively solve other methods' mental set problems.** In the initial submission, we state that DMAD can solve mental set problems of other MAD methods in Section 4.3.4 and list many examples in Appendix D.
>
> |         | MAD (All CoT) | MAD (All SBP) | MAD (All PoT) |
> |----------|---------|---------|------|
> | Number of mental set problems |    70    |   87    |   67    |
> ||
> | Problems that MAD (All CoT) correctly solves |  0    |   45 (51.72%)   |   46 (68.7%)   |
> | Problems that MAD (All SBP) correctly solves |   28 (40.0%)    |   0    |   31 (46.3%)   |
> | Problems that MAD (All PoT) correctly solves |   **49 (70.0%)**   |   51 (58.62%)    |   0    |
> | Problems that DMAD correctly solves |   48 (68.6%)   |   **60 (69.0%)**   |  **49 (73.1%)**    |
>
> - **Weakness 2-2: Statistical significance.**
>
> We run experiments with GPT-4o-mini and LLaMA-3-70B-Instruct on MATH 3 times to calculate the average accuracy and standard deviation. **Statistical experiments demonstrate that DMAD outperforms other MAD settings. DMAD on LLaMA-3-70B-Instruct also gets better average accuracy than MAD (All PoT). This implies the gains come from method diversity rather than the debate process itself to some extent.**
>
> | Models | Methods       | Alg. | C&P  | Geom. | Int. Alg. | Num. Th. | PreAlg. | PreCalc. | Avg.  |
> |---------|---------------|-------|------|--------|------------|-----------|----------|-----------|-------|
> | GPT-4o-mini     | MAD (All CoT) | 91.3±2.16 | 78.7±0.82 | 55.3±2.16 | **55.0±2.45**  | **82.7±1.63** | **86.3±0.82** | 39.7±0.82 | 69.9±0.93 |
> |   GPT-4o-mini      | MAD (All SBP)| 88.3±0.82  | 77.7±1.63  | 49.3±2.16  | 44.0±3.74  | 81.3±2.94 | 83.7±0.82 | 38.7±0.82 | 66.1±0.35 |
> |   GPT-4o-mini      | MAD (All PoT)| 91.3±1.63  | 75.7±2.16 | 49.0±3.74  | 52.7±5.72  | 80.7±0.82 | 85.3±0.82 | 39.7±0.82 | 67.8±2.24|
> |    GPT-4o-mini     | DMAD (Ours)        | **91.7±1.63**  | **81.0±1.41**  | **57.3±2.16**  | 53.7±0.82  | **82.7±0.82** | **86.3±0.82** | **40.0±1.41** | **70.4±0.95**|
> |
> | LLaMA-3-70B-Instruct   | MAD (All CoT) | **72.7±2.94**  | 48.0±3.74  | 31.3±0.82  | 24.3±0.82  | 40.7±2.94 | 69.7±2.94 | 31.0±1.41 | 45.6±0.23|
> | LLaMA-3-70B-Instruct      | MAD (All SBP)| 69.3±4.32  | 51.0±2.83  | 29.3±4.32  | 25.0±0.00  | 42.0±3.74 | 70.0±1.41 | 27.0±1.41 | 44.8±1.52|
> |   LLaMA-3-70B-Instruct       | MAD (All PoT)| 66.7±5.89  | **52.0±2.83**  | 31.0±2.45  | 27.7±2.16  | **49.0±4.90** | 67.0±7.35 | **32.7±1.63** | 46.6±0.35|
> |   LLaMA-3-70B-Instruct       | DMAD (Ours)        | **72.7±1.63**  | 49.0±4.24  | **32.7±1.63**  | **29.3±2.94**  | 44.3±4.08  | **72.3±2.16** | 27.3±3.56 | **46.8±1.43** |

---

> ### Author Response · Authors · 2024-11-19
> **Response 2/4**
>
> - **Weakness 1: Resource efficiency analysis.**
> Thanks for your advice. We report the token overhead and cost for each method in the experiment on MATH with GPT-4o-mini. **DMAD balances the overhead of MAD (All CoT), MAD (All SBP), and MAD (All PoT), and can achieve the best performance than other methods.**
>
> | Method            | Tokens   | Cost ($) | Accuracy |
> |-------------------|----------|---------|  --- |
> | CoT-SC            | 1,494,692  | 0.7801  | 68.57|
> | SBP-SC            | 3,933,134  | 1.8876  | 66.43|
> | PoT-SC            | 1,015,705  | 0.3441  | 56.14|
> | Self-Refine       | 2,871,764  | 1.0271  | 67.71|
> | Self-Contrast     | 6,159,049  | 2.4389  | 62.14|
> | MRP               | 4,298,926  | 2.0293  | 65.00|
> | MAD-persona-D     | 5,156,017  | 1.2743  | 62.43|
> | MAD-persona-E     | 2,680,871  | 0.6824  | 62.43|
> | MAD (All CoT)           | 4,445,066  | 1.5077  | 69.86|
> | MAD (All SBP)           | **11,076,792** | **3.5419**  | 66.14|
> | MAD (All PoT)           | 3,113,716  | 1.0215  | 67.76|
> | DMAD (Ours)             | 6,331,316  | 2.0449  | **70.38**|
>
>
> - **Weakness 2-3: Whether the gains are truly from method diversity or simply from the debate process itself?**
> - **Weakness 4: Limited implementation guidelines.**
>
> -  **Question 1: Why choosing CoT, SBP, PoT? How about Least-to-Most prompting?**
>
> We initially chose divergent prompting strategies by intuition. CoT solves the problem step by step, but may struggle with problems involving complex theorems. SBP first extracts relevant principles and then solves the problem according to them, but may weaken the ability to solve problems step by step. PoT uses Python programs, but this is mostly suitable for problems involving numbers. Therefore, these methods are diverse and can complement each other. Instead, Least-to-Most (L2M) breaks down the problem into progressive sub-questions and answers them to get the final answer. It looks somewhat similar to CoT which gradually solves the problem in steps.
>
> To offer a more quantitative analysis, we design a more objectively quantitative metric. Assuming we have $K$ candidate basic reasoning strategies $\\{R_i\\}^K_{i=1}$ and want to select $k$ diverse ones, run each strategy $N$ times, and record the problems which $R_{i}$ correctly solve at least once as $P_{i}$ (Note that the definition of $P_{i}$ here is different from $P_{i}$ when introducing mental set). Note all problems on the measured dataset as $P_{all}$. We can define the diversity of the selected strategies $\lbrace R_{s_i}\rbrace ^{k}_{i=1}$ as
>
> $$diversity =\frac{\|\cup \\{{P_{s_i}\\}}^{k}_{i=1}\|}{\|P\_{all}\|}\in [0, 1],  s_i \in \\{1, 2, ... \,K\\}, s_i \neq s_j \, for \\, \\, i \neq j.$$
>
>  This represents the proportion of total questions that the selected $k$ methods can answer correctly. **The more diverse these methods are, the larger this proportion should be.**
>
> We run $N=3$ times for each reasoning strategy in  $\\{{R_i}  \\}^{4}_{i=1}=\\{CoT, L2M, SBP, PoT\\}$ and select $k=3$ strategies to calculate $diversity$, and test DMAD with different strategy groups on MATH with GPT-4o-mini. We can see $diversity$ of {CoT, SBP, PoT} is the highest. **Experiment results show that using the method group with larger $diversity$ can get better results. This further proves gains are from method diversity, not the debate process.**
>
> |Model| ${R_{s}}_{1}$  | ${R_{s}}_{2}$  | ${R_{s}}_{3}$  |$diversity$ |
> |---|---|---|---|-------|
> |GPT-4o-mini| CoT | L2M | SBP |  0.8471|
> |GPT-4o-mini| CoT | L2M | PoT |  0.8643|
> |GPT-4o-mini| **CoT** | **SBP** | **PoT** |  **0.8657** |
> |GPT-4o-mini| L2M | SBP | PoT |  0.8557|
> ||
> |LLaMA-3-70B-Instruct| CoT | L2M | SBP | 0.6743|
> |LLaMA-3-70B-Instruct| CoT | L2M | PoT | 0.6600|
> |LLaMA-3-70B-Instruct| **CoT** | **SBP** | **PoT** | **0.7286**|
> |LLaMA-3-70B-Instruct| L2M | SBP | PoT | 0.6386|
>
> |  Method  | Alg. | C&P | Geom. | Int. Alg. | Num. Th. | PreAlg. | PreCalc. | Average |
> |--|---------|---------------------------|----------|-----------------------|---------------|------------|-------------|---------|
> | DMAD (CoT, L2M, SBP)     | 88.7±0.82 | 78.0±1.41                 | 54.7±4.55 | 49.0±5.10             | **82.7±0.82**      | 85.7±0.82   | 37.7±3.27   | 68.0±1.83 |
> | DMAD (CoT, L2M, PoT)     | **91.7±1.63** | **81.3±2.94**                 | 54.0±1.41 | **54.7±0.82**             | 82.3±0.82      | **87.7±2.16**   | 39.0±2.83   | 70.1±0.65 |
> | DMAD (CoT, SBP, PoT)     | **91.7±1.63** | 81.0±1.41                   | **57.3±2.16** | 53.7±0.82             | **82.7±0.82**      | 86.3±0.82   | **40.0±1.41**   | **70.4±0.95**|
> | DMAD (L2M, SBP, PoT)|	87.0±1.41 |	**81.3±5.72** |	54.7±3.27 |	51.3±1.63	| 80.7±4.55	| 85.0±1.41| 	38.0±3.74| 68.3±2.13|

---

> ### Author Response · Authors · 2024-11-19
> **Response 3/4**
>
> - **Weakness 3: Model capability of scaling.**
>
> We test MAD (All CoT), MAD (All SBP), MAD (All PoT), and DMAD with LLaMA-3-8B-Instruct on MATH. **Results demonstrate DMAD is also effective on smaller models.**
>
> | Models          | Methods            | Alg. | C&P | Geom. | Int. Alg. | Num. Th. | PreAlg. | PreCalc. | Avg. |
> |-----------------|--------------------|---------|--------------------------|-----------|-----------------------|---------------|------------|-------------|---------|
> | LLaMA-3-8B-Instruct | MAD (All CoT)     | **49.7±0.82** | 19.3±1.63            | 18.3±2.16 | 15.3±0.82           | 18.3±0.82    | **46.7±1.63**   | 15.0±1.41 | 26.1±0.42 |
> |  LLaMA-3-8B-Instruct               | MAD (All SBP)     | 46.3±2.94 | 20.7±4.97            | 17.3±1.63 | 15.3±0.82           | 20.0±6.16   | 44.3±5.35   | 11.3±0.82 | 25.3±1.94 |
> |  LLaMA-3-8B-Instruct               | MAD (All PoT)     | 41.3±5.72 | 20.0±2.83            | 16.0±1.41 | 13.7±4.55           | 21.0±5.66   | 39.7±4.97   | **19.0±4.24** | 24.4±1.62 |
> | LLaMA-3-8B-Instruct                | DMAD               | 47.7±0.82 | **21.3±0.82**            | **21.7±0.82** | **16.0±0.00**           | **21.3±5.72**   | 45.0±1.41   | 15.0±1.41 | **26.9±0.40** |
>
> However, LLaMA-3-8B-Instruct using DMAD still gets lower accuracy (26.9%) than LLaMA-3-70B-Instruct using CoT (41.43%). We analyze that the upper bound of LLaMA-3-8B-Instruct using DMAD is still lower than 41.43%. We calculate $diversity$ for {CoT, SBP, PoT} and get 0.3829. This means {CoT, SBP, PoT} can only correctly solve 38.29% of the problems, which is lower than 41.43%.
>
> - **Weakness 5: Incremental rather than novel contribution.**
>
> We would like to emphasize our novelty and differences compared with other methods leveraging diverse solutions.
>
> 1.   We introduce a new concept of mental set and provide a specific definition and systematic analysis as above, which is inspired by the psychological theory of mental set.
> 2.   Self-Contrast just contrasts diverse reasoning solutions to get better performance than simple IO solutions. However, they don't compare their method with each distinct reasoning strategy. So it's not sure if using diverse solving perspectives all outperforms completely using one perspective among them. Meta-Reasoning Prompting dynamically chooses the best reasoning method to solve problems, while it only achieves a balanced performance between various reasoning methods and is often inferior to a specific one in most cases. Different from them, we conduct a more comprehensive analysis and prove that leveraging diverse reasoning methods can outperform all single methods with the same setting.
> 3. Our experiment shows that DMAD has the potential to break the mental set of single reasoning methods. For example, for the same problem, all agents in MAD (All IO) constantly get wrong answers in all 5 rounds. While in DMAD, an agent thinking with IO correctly solves the problem after receiving messages from agents thinking from other perspectives in the 2nd round. This also holds for MAD (All CoT), MAD (All SBP), MAD (All PoT), MAD (All CCoT) and MAD (All DDCoT). We state this in Section 4.3.4 in the initial submission.
>
> - **Question 2: Could the system dynamically adjust its reasoning methods during the debate process?**
>
> We assign each agent to think with a fixed reasoning method in DMAD. It's interesting to teach each agent to dynamically adjust its reasoning method. We mention that we leave it for future work in Appendix A *"LIMITATIONS AND FUTURE WORK"*, *"We can also dynamically guide agents to choose the method they deem appropriate independently. We leave this to future work and believe this may further improve the models’ reasoning performance.*" However, from another perspective, our experiments have proven that an agent thinking with a fixed method can break its mental set by getting insights from other diverse solutions. So it may not be necessary to design a dynamic adjustment strategy.

---

> ### Author Response · Authors · 2024-11-19
> **Response 4/4**
>
> - **Question3: Why focusing on mathematical reasoning and visual understanding?**
>
> We conduct abundant experiments in multiple aspects, not limited to mathematical reasoning and vision understanding. In our paper, we conduct experiments on mathematics, biology, chemistry, and physics reasoning. We also test on Scientific Vision-Question Answering (VQA) including natural, social, and language science, and MM-Vet involving various visual capabilities such as object recognition, optical character recognition, knowledge answering, math VQA, language generation, and spatial awareness. These benchmarks have been widely used by researchers.
>
> **What's more, we supplement experiments on a more challenging benchmark, MMLU [1].** We use GPT-4o-mini to conduct experiments on the subset "abstract_algebra" of MMLU and run 3 times to get the average accuracy and standard deviation for each method. As this dataset consists of multi-choice questions and some options are not numbers, we replace PoT with L2M. **DMAD also outperforms other MAD settings on this challenging multi-hop reasoning task.**
>
> |         | MAD (All CoT) | MAD (All SBP) | MAD (All L2M) | DMAD  |
> |---------|---------|---------|---------|-------|
> | Accuracy| 72.3±0.82 | 79.0±1.41 | 74.3±0.82 | **79.7±1.63** |
>
> - **Question 4: What criteria determine the ideal number of debate rounds? Is there a smarter stopping criteria?**
>
> To fairly compare with MAD, we adopt all the same settings in their paper, which executes MAD in the fixed round. Without considering this, we can design different criteria to determine when the debate ends, such as:
>
> 1. Consistency-2: If there exist 2 agents getting the same answer, the debate ends. Otherwise, the debate continues unless reaching its maximum number of rounds.
>
> 2. Consistency-3: If all the 3 agents get the same answer, the debate ends. Otherwise, the debate continues unless reaching its maximum number of rounds.
>
> 3. Self-Determine: Set another model as a judge to determine whether the debate should be over. The judge can receive all agents' solutions in each round and save them in its history.
>
> 4. Hybrid: In each round, if all agents get the same answer, the debate ends. Otherwise, use Self-Determine to judge whether the debate should end or continue.
>
> We set the maximum number of the debate round to 5 with Gemini on ScienceQA. With the stop criteria, MAD and DMAD can get a relatively high accuracy with a low average round. Results show that the criteria of Consistency-3 is the best. However, DMAD with the stop criteria of Consistency-3 needs more overhead with nearly 5 debate rounds.
>
> | Stop Criteria   | Method | Average Round | Accuracy | Method | Average Round | Accuracy |
> |-----------------|---------|---------------|-----------|---------|---------------|-----------|
> | Consistency-2    | MAD     | 1.0977        | 85.09     | DMAD    | 1.0788        | 84.93     |
> | Consistency-3    | MAD     | 1.1076        | 85.14     | DMAD    | 4.9861        | 85.32     |
> | Self-Determine   | MAD     | 1.1091        | 84.89     | DMAD    | 1.6063        | 85.03     |
> | Hybrid          | MAD     | 1.0605        | 84.84     | DMAD    | 1.6063        | 85.03     |
> | Fixed Round      | MAD     | 1            | 84.24     | DMAD    | 1            | 84.58     |
> | Fixed Round      | MAD     | 2            | 84.84     | DMAD    | 2            | 85.57     |
> | Fixed Round      | MAD     | 3            | 85.34     | DMAD    | 3            | **85.62**     |
> | Fixed Round      | MAD     | 4            | 85.34     | DMAD    | 4            | 85.52     |
> | Fixed Round      | MAD     | 5            | 85.44     | DMAD    | 5            | 85.52     |
>
>
> Thanks again for your valuable feedback. We hope our responses can address your concerns. Looking forward to further communication with you.

---

### Official Review · Reviewer_PpgF · 2024-11-04

**Soundness:** 3
**Presentation:** 3
**Contribution:** 3
**Rating:** 6
**Confidence:** 3

**Summary:**

This paper proposes Diverse Multi-Agent Debate (DMAD), an improved version of Multi-Agent Debate (MAD) that leverages agents using different reasoning methods to break "mental set" and enhance reasoning performance. In DMAD, each agent adopts a distinct prompting strategy (e.g. chain-of-thought, step-back prompting, program-of-thoughts) to generate solutions. The agents then review each other's solutions, extract insights from the different reasoning approaches, and iteratively refine their own responses. Experiments on both language models and multimodal language models across various benchmarks demonstrate that DMAD outperforms MAD with homogeneous reasoning as well as other baselines like self-reflection methods. Key results include DMAD in 2 rounds surpassing MAD in 5 rounds, and improvements of 7-8% on metrics compared to 5% for standard MAD. The paper provides a systematic analysis of DMAD, showing performance improves with more debate rounds and agents using a greater diversity of reasoning methods.

**Strengths:**

1) The key idea of using diversity in reasoning methods to break mental set and improve multi-agent debate is novel and creative. This can leverage different type of strneghts and activation patternsa of LLMs at inference time to produce diverse outputs. This leverages the ideas from cognitive science - the idea of mental set is interestijng.
2) The approach is evaluated across a variety of models (multiple open and closed-source language and multimodal models), datasets/tasks (4 language and multimodal benchmarks), and against strong baselines (standard MAD, self-reflection methods, etc). The authors demonstrate that DMAD approach works well.
3) The ablation studies on number of agents, debate rounds, importance of thought process, etc provide ways to undersstand the robustness of the papers results.

**Weaknesses:**

While the experiments cover a good range of models and tasks, it would have been nice to see results on more challenging reasoning datasets like MMLU to further validate the approach.

While the paper shows that using agents with diverse reasoning strategies is helpful, its not clear on what kind of diversity is most effective. The authors use three pre-defined reasoning methods (chain-of-thought, step-back, program-of-thoughts), but do not explore other possibilities or study the properties of these methods that make them complementary. A more principled analysis of reasoning diversity could be helpful.

The paper shows the benefits of DMAD with a fixed number of agents (3) and debate rounds (2-5). However, there is no systematic analysis of how the performance and computational cost scale with these parameters. In real-world applications, it may be necessary to use a much larger number of agents and rounds to tackle complex reasoning tasks. Understanding the scalability properties of DMAD is important.
Also, for the multimodal experiments, it's unclear if the visual information is playing an important role. Adding experiments on more vision-centric tasks/datasets could strengthen this aspect.
Its also not well discussed why self-reflection methods struggle on multimodal tasks compared to language-only tasks. Is this an inherent limitation or due to the specific self-reflection approaches used?

The paper is a good start but stops short on addressing these points.

**Questions:**

Some questions came up during the analysis:

How does DMAD perform on tasks that require multi-hop reasoning over longer contexts? Do the gains over baselines hold there as well?
Expanding on the above,  it would be great to see MMLU results, even if just on a subset of subjects most relevant to reasoning.
Can the authors comment on the computational overhead of DMAD compared to standard MAD, both at inference time and during the debate rounds?
Since multimodal models are more computationally intensive in general, were the number of trials/random seeds adjusted for those experiments? Clarifying those details would be helpful.
Figure 1 is quite interesting - if space permits, visualizing even more of the ablations (number of rounds, removing step output, etc) in this format could provide further insight.

---

> ### Author Response · Authors · 2024-11-19
> **Summary Response and Experiments on MMLU**
>
> We thank the reviewer for considering our idea of mental set is interesting and commending our sufficiency of experiments. We also appreciate the valuable and insightful feedback provided, which could be much helpful to better prove the effectiveness of our method. We would like to address each of your concerns.
>
> We summarize our responses as follows:
>
> 1. We conduct experiments on the subset of MMLU, where our method DMAD also outperforms other methods. (In Summary Response)
> 2. We provide a specific and systematic definition for **mental set**, connected with the corresponding psychological phenomenon. (In Respone 1/3)
> 3. We design an objectively quantitative metric to measure the diversity of selected basic prompting strategies, and conduct several experiments to prove the effectiveness of this metric. (In Response 1/3)
> 4. We report the computational overhead including tokens and inference time of DMAD and MAD during the debate rounds. DMAD achieves optimal performance with balanced overhead compared with other MAD methods. (In Response 2/3)
> 5. We design 4 different stop criteria to determine when the debate ends to reduce excessive tokens. (In Response 2/3)
> 6. We test different MLLMs and LLMs using 2 different *self-reflection* prompts, proving and analyzing the poor performance of *self-reflection* on MLLMs. (In Response 3/3)
>
> We also respond to other minor questions and have added these supplements in the revised paper. We hope our responses can address your concerns. Looking forward to further communication with you.
>
> - **Experiments on more challenging multi-hop reasoning tasks such as MMLU.**
>
> Thanks for your suggestion. We use GPT-4o-mini to conduct experiments on the subset "abstract_algebra" of MMLU and run 3 times to get the average accuracy and standard deviation for each method. As this dataset consist of multi-choice questions and some options are not numbers, we replace PoT with Least-to-Most (L2M) [1]. **DMAD also outperforms other MAD settings on this challenging multi-hop reasoning task.**
>
> |         | MAD (All CoT) | MAD (All SBP) | MAD (All L2M) | DMAD (Ours) |
> |---------|---------|---------|---------|-------|
> | Accuracy| 72.3±0.82 | 79.0±1.41 | 74.3±0.82 | **79.7±1.63** |
>
> [1] Least-to-Most Prompting Enables Complex Reasoning in Large Language Models. ICLR 2023.

---

> ### Author Response · Authors · 2024-11-19
> **Response 1/3**
>
> - **What is mental set?**
>
> In our paper, we introduce a new concept of **mental set** according to the psychological theory. Here we supplement a specific definition for it. Denote MAD (All CoT), MAD (All SBP), and MAD (All PoT) as $M_{1}, M_{2}$ and $M_{3}$ respectively. When using a kind of MAD method $M_{i}$ to solve a problem, if all agents consistently get wrong answers in all debate rounds, we assume that $M_{i}$ is unable to solve the problem correctly. Record all such problems for $M_{i}$ as the set $P_{i}$, and get $P=P_{1} \cap P_{2} \cap P_{3}$. For a problem $p \in P_{i}$, if it satisfies $p \notin P$, we define that the problem $p$ causes **mental set** of $M_{i}$, and define $p$ as the **mental set problem** of $M_{i}$. It means although $M_{i}$ constantly gets wrong solutions to $p$, the model can correctly solve the problem by changing to another method.
>
> We record the mental set problems of each method on MATH with GPT-4o-mini, and the problems among them which at least one agent of other methods correctly solves. All methods have 3 agents and debate for 2 rounds. **The results show that DMAD can more effectively solve other methods' mental set problems.**
>
> |         | MAD (All CoT) | MAD (All SBP) | MAD (All PoT) |
> |----------|---------|---------|------|
> | Number of mental set problems |    70    |   87    |   67    |
> ||
> | Problems that MAD (All CoT) correctly solves |  0    |   45 (51.72%)   |   46 (68.7%)   |
> | Problems that MAD (All SBP) correctly solves |   28 (40.0%)    |   0    |   31 (46.3%)   |
> | Problems that MAD (All PoT) correctly solves |   **49 (70.0%)**   |   51 (58.62%)    |   0    |
> | Problems that DMAD correctly solves |   48 (68.6%)   |   **60 (69.0%)**   |  **49 (73.1%)**    |
>
> - **What kind of diversity is most effective?**
>
> We initially chose divergent prompting strategies by intuition. CoT solves the problem step by step, but may struggle with problems involving complex theorems. SBP first extracts relevant principles and then solves the problem according to them, but may weaken the ability to solve problems step by step. PoT uses Python programs, but this is mostly suitable for problems involving numbers. **Therefore, these strategies are diverse and can complement each other.**
>
> To offer a more quantitative analysis, we design a more objectively quantitative metric. Assuming we have $K$ candidate basic reasoning strategies $\\{R_i\\}^K_{i=1}$ and want to select $k$ diverse ones, run each strategy $N$ times, and record the problems which $R_{i}$ correctly solve at least once as $P_{i}$ (Note that the definition of $P_{i}$ here is different from $P_{i}$ when introducing mental set). Note all problems on the measured dataset as $P_{all}$. We can define the diversity of the selected strategies $\lbrace R_{s_i}\rbrace ^{k}_{i=1}$ as
>
> $$diversity =\frac{\|\cup \\{{P_{s_i}\\}}^{k}_{i=1}\|}{\|P\_{all}\|}\in [0, 1],  s_i \in \\{1, 2, ... \,K\\}, s_i \neq s_j \, for \\, \\, i \neq j.$$
>
> This represents the proportion of total questions that the selected $k$ methods can answer correctly. **The more diverse these methods are, the larger this proportion should be.**
>
> We run $N=3$ times for each reasoning strategy in $\\{{R_i}  \\}^{4}_{i=1}=\\{CoT, L2M, SBP, PoT\\}$ and select $k=3$ strategies to calculate $diversity$, and test DMAD with different strategy groups on MATH with GPT-4o-mini. **We can see that using the method group with larger** $diversity$ **can get better results.**
>
> |Model| ${R_{s}}_{1}$  | ${R_{s}}_{2}$  | ${R_{s}}_{3}$  |$diversity$ |
> |---|---|---|---|-------|
> |GPT-4o-mini| CoT | L2M | SBP |  0.8471|
> |GPT-4o-mini| CoT | L2M | PoT |  0.8643|
> |GPT-4o-mini| **CoT** | **SBP** | **PoT** |  **0.8657** |
> |GPT-4o-mini| L2M | SBP | PoT |  0.8557|
> ||
> |LLaMA-3-70B-Instruct| CoT | L2M | SBP | 0.6743|
> |LLaMA-3-70B-Instruct| CoT | L2M | PoT | 0.6600|
> |LLaMA-3-70B-Instruct| **CoT** | **SBP** | **PoT** | **0.7286**|
> |LLaMA-3-70B-Instruct| L2M | SBP | PoT | 0.6386|
>
> |  Method  | Alg. | C&P | Geom. | Int. Alg. | Num. Th. | PreAlg. | PreCalc. | Average |
> |--|---------|---------------------------|----------|-----------------------|---------------|------------|-------------|---------|
> | DMAD (CoT, L2M, SBP)     | 88.7±0.82 | 78.0±1.41                 | 54.7±4.55 | 49.0±5.10             | **82.7±0.82**      | 85.7±0.82   | 37.7±3.27   | 68.0±1.83 |
> | DMAD (CoT, L2M, PoT)     | **91.7±1.63** | **81.3±2.94**                 | 54.0±1.41 | **54.7±0.82**             | 82.3±0.82      | **87.7±2.16**   | 39.0±2.83   | 70.1±0.65 |
> | DMAD (CoT, SBP, PoT)     | **91.7±1.63** | 81.0±1.41                   | **57.3±2.16** | 53.7±0.82             | **82.7±0.82**      | 86.3±0.82   | **40.0±1.41**   | **70.4±0.95**|
> | DMAD (L2M, SBP, PoT)|	87.0±1.41 |	**81.3±5.72** |	54.7±3.27 |	51.3±1.63	| 80.7±4.55	| 85.0±1.41| 	38.0±3.74| 68.3±2.13|

---

> ### Author Response · Authors · 2024-11-19
> **Response 2/3**
>
> - **Computational overhead including tokens and inference time of DMAD and MAD during the debate rounds.**
>
> Thanks for your advice. This is important for real-world applications. Based on our record of experiment results on Gemini on ScienceQA, we use a word as a token to calculate the tokens overhead of traditional MAD, *i.e.*, MAD (All IO), and DMAD during the debate rounds.
>
> We can see that (DMAD tokens / MAD tokens) becomes larger as the debate round increases. This is because the selected diverse strategies generate more tokens. The diverse solutions are added to each agent's debate history in each round. The accumulated history of diverse solutions will cause more and more overhead. **Nonetheless, DMAD in 2 rounds gets better performance and needs lower overhead and calls than MAD when MAD achieves its best performance in 5 rounds.**
>
> | Round | DMAD Accuracy| MAD Accuracy| DMAD tokens | MAD tokens | DMAD tokens / MAD tokens |
> |-------|----|---|------|------------|----------------------------|
> | 1     | 84.58| 84.24| 1,169,209     | 721,800     | 1.61985                    |
> | 2     | 85.57| 84.84| 4,114,780     | 2,059,791    | 1.99767                    |
> | 3     | **85.62**| 85.34| 7,753,647     | 3,585,669    | 2.16240                    |
> | 4     | 85.52| 85.34| 11,517,726    | 5,215,068    | 2.20855                    |
> | 5     | 85.52| 85.44| 15,364,827    | 6,905,985    | 2.22486                    |
>
> **When to stop the debate?**
>
> Besides, it is important to determine when the debate ends to reduce additional overhead. Although we run all experiments in the fixed round for a fair comparison, which is consistent with the settings of the paper of MAD, we can also design different criteria to determine when the debate ends, such as:
>
> 1. Consistency-2: If there exists 2 agents getting the same answer, the debate ends. Otherwise, the debate continues unless reaching its maximum number of rounds.
>
> 2. Consistency-3: If all the 3 agents get the same answer, the debate ends. Otherwise, the debate continues unless reaching its maximum number of rounds.
>
> 3. Self-Determine: Set another model as a judge to determine whether the debate should be over. The judge can receive all agents' solutions in each round and save them in its history.
>
> 4. Hybrid: In each round, if all agents get the same answer, the debate ends. Otherwise, use Self-Determine to judge the debate should end or continue.
>
> We set the maximum number of the debate round to 5. With the stop criteria, MAD and DMAD can get a relatively high accuracy with low average round. Results show that the criteria of Consistency-3 is the best. However, DMAD with the stop criteria of Consistency-3 needs more overhead with nearly 5 debate rounds.
>
> | Stop Criteria   | Method | Average Round | Accuracy | Method | Average Round | Accuracy |
> |-----------------|---------|---------------|-----------|---------|---------------|-----------|
> | Consistency-2    | MAD     | 1.0977        | 85.09     | DMAD    | 1.0788        | 84.93     |
> | Consistency-3    | MAD     | 1.1076        | 85.14     | DMAD    | 4.9861        | 85.32     |
> | Self-Determine   | MAD     | 1.1091        | 84.89     | DMAD    | 1.6063        | 85.03     |
> | Hybrid          | MAD     | 1.0605        | 84.84     | DMAD    | 1.6063        | 85.03     |
> | Fixed Round      | MAD     | 1            | 84.24     | DMAD    | 1            | 84.58     |
> | Fixed Round      | MAD     | 2            | 84.84     | DMAD    | 2            | 85.57     |
> | Fixed Round      | MAD     | 3            | 85.34     | DMAD    | 3            | **85.62** |
> | Fixed Round      | MAD     | 4            | 85.34     | DMAD    | 4            | 85.52     |
> | Fixed Round      | MAD     | 5            | 85.44     | DMAD    | 5            | 85.52     |
>
>
> The inference time of DMAD and MAD has the same trend as tokens. We re-conduct experiment on MATH with gpt-4o-mini and record the tokens overhead and inference time of DMAD and MAD. We run 3 times to calculate the average accuracy and standard deviation. **Our DMAD approach effectively strikes a balance among different approaches that require varying amount of inference computation while achieving optimal performance.** We hope the reported results can be helpful.
>
> |Method             | Tokens   |  Cost ($) |Inference Time (s)| Accuracy |
> |-------------------|----------|---------|-------|--|
> | MAD (All CoT)           | 4,445,066  | 1.5077  |  34,376 | 69.9±0.93|
> | MAD (All SBP)           | 11,076,792 | 3.5419  | 76,425 |66.1±0.35|
> | MAD (All PoT)           | 3,113,716  | 1.0215  | 25,189 |67.8±2.24|
> | DMAD              | 6,331,316  | 2.0449  | 45,514 | **70.4±0.95**|

---

> ### Author Response · Authors · 2024-11-19
> **Response 3/3**
>
> - **Adding experiments on more vision-centric tasks/datasets.**
>
> In our submitted paper, we conduct experiments on both LLMs and MLLMs to further verify the effectiveness of DMAD on different modalities. **We test 3 MLLMs on two vision reasoning benchmarks including multiple aspects such as object recognition, optical character recognition, knowledge answering, math vision-question answering, language generation, and spatial awareness, which requires MLLMs to solve vision-centric problems according to the given image.**
>
> We believe our evaluated datasets and tasks provide evidence to support the conclusions drawn. However, we greatly value the reviewer’s insights and would be happy to explore additional tasks or datasets that require different types of reasoning or demonstrate other properties, if the reviewer has specific recommendations.
>
> - **Why self-reflection methods struggle on multimodal tasks compared to language-only tasks?**
>
> We discuss it in Appendix C.3. This may be due to the underconfidence of MLLMs, which tend to believe that their initial answer is incorrect and modify it, even if most of their answers are right. Here we compared Self-Refine results of MLLMs on ScienceQA and LLMs on MATH. **We can find MLLMs are more likely to change their right answers to wrong ones.**
>
> | Type | Model     | Maintain | Right -> Wrong | Wrong -> Right | Wrong -> Wrong | Accuracy Variation |
> |--------|------------|-----------|----------------|-----------------|-----------------|-------------------|
> | MLLM   | LLaVA      | 36.24      | 38.77           | 14.28           | 10.71           | -24.49           |
> | MLLM   | Gemini     | 36.49      | 48.44           | 8.28            | 6.79            | -40.16            |
> | MLLM   | GPT-4o     | 66.00       | 25.00              | 4.00               | 5.00               | -21.00              |
> | LLM    | LLaMA      | 38.86        | 14.43              | 8.71      | 38.00      | -5.71            |
> | LLM    | GPT-4o-mini| 79.14        | 4.29     | 4.71      | 11.86      | +0.43             |
>
> *(Maintain: The answer remains unchanged. Right -> Wrong: A right answer is changed to wrong. Wrong -> Right: A wrong answer is changed to right. Wrong -> Wrong: A wrong answer is changed but remains incorrect.)*
>
> However, this may be affected by specific prompts. The evaluation prompt in Self-Refine, *"Review your previous answer and find problems with your answer."*, and the refinement prompt, *"Based on the problems you found, improve your answer."*, may lead models to nitpick on correct answers and find problems of right solutions. Therefore, we change the evaluation prompt to *"Review your previous answer and determine whether your previous answer is right or wrong."* to get feedback, and use *"Based on your judgment, improve your answer. If your previous answer is judged as wrong, modify it to be correct. Otherwise, keep your previous answer."* to refine. **On LLaVA and Gemini, this prompt setting performs better than Self-Refine, while still getting worse results than before modification.**
>
> | Type | Model         | Maintain | Right -> Wrong | Wrong -> Right | Wrong -> Wrong | Accuracy Variation |
> |--------|----------------|-----------|----------------|-----------------|-----------------|---------------|
> | MLLM   | LLaVA         | 66.83     | 17.95           | 9.02            | 6.20           | -8.93          |
> | MLLM   | Gemini        | 67.01     | 20.38           | 7.27            | 5.34           | -13.14          |
> | LLM    | LLaMA         | 30.71     | 20.00           | 9.14            | 40.14          | -10.86         |
> | LLM    | GPT-4o-mini   | 86.57     | 1.57            | 4.00              | 7.86           | +2.43           |
>
>
>
>
> - **Were the number of trials/random seeds adjusted for experiments on MLLMs?**
>
> No. On the the same benchmark and model, we strictly test the performance of different methods using completely identical settings, only the prompts are different. Although MLLMs are computationally intensive, we test LLaVA, Gemini, and GPT-4o on the whole MM-Vet benchmark. We also test LLaVA and Gemini on all ScienceQA samples, and test GPT-4o with 100 random samples with the sample seed 0 for reproduction. We clarify these in Section 4.2.2.
>
> - **Visualizing more of the ablations in the format of Figure 1.**
>
> Thanks for your advice. We have added such a figure in Appendix.
>
>
> Thanks again for your valuable feedback. We hope our responses can address your concerns. Looking forward to further communication with you.

---

### Meta-Review · Area_Chair_EHXZ · 2024-12-20

**Metareview:**

This paper is a contribution to the literature on multi-agent debate. Reviewers were pleased that the paper was clearly written and showed clear improvements over appropriately chosen baselines. There were concerns about the novelty of the approach but no one thought they were insurmountable.

**Additional Comments On Reviewer Discussion:**

There was reasonably good engagement in the conversation. One reviewer increased their score after reading the author response.

---

### Decision · Program_Chairs · 2025-01-22

Accept (Poster)